# Alkali cation-induced cathodic corrosion in Cu electrocatalysts

Shikai Liu[1,8], Yuheng Li [1,8], Di Wang[2,8], Shibo Xi[3] ✉, Haoming Xu[4], Yulin Wang[4], Xinzhe Li [1], Wenjie Zang [1], Weidong Liu[1], Mengyao Su[1], Katherine Yan [5], Adam C. Nielander [5], Andrew B. Wong [1,2,6], Jiong Lu [4,6], Thomas F. Jaramillo [5,7], Lei Wang [2,6] ✉, Pieremanuele Canepa [1,2] ✉ & Qian He [1,6] ✉

The reconstruction of Cu catalysts during electrochemical reduction of $CO_2$ is a widely known but poorly understood phenomenon. Herein, we examine the structural evolution of Cu nanocubes under $CO_2$ reduction reaction and its relevant reaction conditions using identical location transmission electron microscopy, cyclic voltammetry, in situ X-ray absorption fine structure spectroscopy and ab initio molecular dynamics simulation. Our results suggest that Cu catalysts reconstruct via a hitherto unexplored yet critical pathway - alkali cation-induced cathodic corrosion, when the electrode potential is more negative than an onset value (e.g., −0.4 $V_{RHE}$ when using 0.1 M $KHCO_3$). Having alkali cations in the electrolyte is critical for such a process. Consequently, Cu catalysts will inevitably undergo surface reconstructions during a typical process of $CO_2$ reduction reaction, resulting in dynamic catalyst morphologies. While having these reconstructions does not necessarily preclude stable electrocatalytic reactions, they will indeed prohibit long-term selectivity and activity enhancement by controlling the morphology of Cu pre-catalysts. Alternatively, by operating Cu catalysts at less negative potentials in the CO electrochemical reduction, we show that Cu nanocubes can provide a much more stable selectivity advantage over spherical Cu nanoparticles.

The electrochemical reduction reaction of $CO_2$ ($CO_2RR$) to higher-value chemicals holds great promise for mitigating climate change and transitioning the chemical industry towards sustainability[1,2]. Among various heterogeneous electrocatalysts, copper (Cu) was found uniquely capable of producing multicarbon products, such as ethylene, ethanol, and propanol, at significant rates[3,4]. While previous research[5–8] mainly aimed at improving the activity and selectivity of Cu catalysts, their stability under reaction conditions is at least equally important and has drawn increasing research attention[9,10]. Some studies demonstrate the stable operation of Cu catalysts for tens or hundreds of hours[11–13]. In

[1]Department of Material Science and Engineering, College of Design and Engineering, National University of Singapore, 9 Engineering Drive 1, EA #03-09, Singapore 117575, Singapore. [2]Department of Chemical and Biomolecular Engineering, College of Design and Engineering, National University of Singapore, 4 Engineering Drive 4, E5 #02-29, Singapore 117585, Singapore. [3]Institute of Sustainability for Chemicals, Energy and Environment (ISCE2), Agency for Science, Technology and Research (A*STAR), 1 Pesek Road Jurong Island, Singapore 627833, Singapore. [4]Department of Chemistry, National University of Singapore, 12 Science Drive 3, Singapore 117543, Singapore. [5]SUNCAT Center for Interface Science and Catalysis, Department of Chemical Engineering, Stanford University, Stanford, CA 94305, USA. [6]Centre for Hydrogen Innovations, National University of Singapore, E8, 1 Engineering Drive 3, Singapore 117580, Singapore. [7]SUNCAT Center for Interface Science and Catalysis, SLAC National Accelerator Laboratory, Menlo Park, CA 94025, USA. [8]These authors contributed equally: Shikai Liu, Yuheng Li, Di Wang. ✉e-mail: xi_shibo@isce2.a-star.edu.sg; wanglei8@nus.edu.sg; pcanepa@nus.edu.sg; mseheq@nus.edu.sg

contrast, others show substantial performance fluctuations within an hour or less[14,15]. The existence of such a discrepancy in literature is puzzling since essentially the same material, copper, was being tested. In addition to the possible causes associated with electrochemical reactors (e.g., flooding)[16], it is widely acknowledged that Cu catalysts undergo significant reconstructions during the electrocatalytic reactions that may also affect its stability[17-19]. Buonsanti and colleagues[20], as well as other researchers[21-23] showed that this potential-induced reconstruction is the primary deactivation mechanism for shape-controlled Cu catalysts. However, in contrast, other studies by Jung et al.[24], by Choi et al.[25], and more recently by Yang et al.[26] suggested that the reconstruction can create active sites in Cu catalysts.

Experimentally studying the reconstruction of Cu under $CO_2RR$ conditions has been challenging at least partially due to the susceptibility of Cu to be oxidized[27]. Recently, many in situ characterization methods, such as in situ electron microscopy[23,26,28,29], in situ scanning probe microscopy[17,18,30], in situ X-ray scattering[31] were developed and applied to study the reconstruction of Cu catalysts. However, the underlying causes and the impact of such a phenomenon on catalyst stability remain still elusive. For instance, some researchers[13,24] attributed it to the reduction of copper oxides, while others showed that $CO$[32,33] or $H_2$[34,35] played important roles. The limited understanding of this reconstruction behavior of Cu catalysts in the cathode made it difficult to clarify the structural-property relationship, possibly also fueling debates regarding issues such as the identity of the active sites in Cu catalysts[36] and whether oxide-derived Cu is intrinsically better than just Cu[8,37,38].

In this work, we track the structural evolution of Cu nanocubes under relevant $CO_2RR$ reaction conditions using a combination of identical location TEM (IL-TEM), cyclic voltammetry (CV), in situ X-ray Absorption Fine Structure Spectroscopy (XAFS), Scanning Transmission Electron Microscopy (STEM), Density Functional Theory (DFT) and grand-potential Ab Initial Molecular Dynamics (AIMD) simulations. Our results suggest that Cu catalysts reconstruct via a hitherto unexplored yet critical pathway - alkali cation-induced cathodic corrosion[39], when the electrode potential is more negative than an onset value (e.g., $-0.4\,V_{RHE}$ when using $0.1\,M$ $KHCO_3$). Having alkali cations in the electrolyte is found critical for this phenomenon to take place. Our experiments and modeling suggest that Cu can be etched out by possibly forming intermediate ternary hydride[40,41] with alkali cations, which can then be redeposited as metallic Cu under cathodic conditions, leading to experimentally observed reconstruction of the Cu catalyst and the formation of smaller Cu particles during the $CO_2RR$ reaction.

Since $CO_2RR$ usually requires highly cathodic conditions and having alkali cations in the electrolyte[42-45], Cu catalysts will inevitably undergo surface reconstructions that lead to dynamic morphologies. We will argue that having this dynamic morphologies does not necessarily preclude stable operation using Cu catalysts as the catalyst morphology may potentially reach an equilibrium. However, because of the alkali cation-induced cathodic corrosion, many approaches towards engineering the morphology of Cu pre-catalyst may unlikely bring long-term selectivity/activity benefits[46-48]. Alternatively, this observation suggests that strategies can be designed to mitigate selectivity changes by reducing the extent to which the cathodic corrosion of Cu can occur. To this end, by operating Cu catalysts at less negative potentials than the threshold (e.g., $-0.4\,V_{RHE}$) in the CO electrochemical reduction reaction (CORR)[49-51], we demonstrate that the cathodic corrosion can be switched off, allowing Cu nanocubes to maintain a stable selectivity advantage over spherical Cu nanoparticles.

## Results

### A hitherto unexplored yet critical reconstruction pathway for Cu catalysts

We use Cu nanocubes (NCs) as our model catalysts due to their well-defined initial structure, enabling us to closely track the structural

evolution of the catalysts. Cu nanocubes were prepared following a previously reported protocol (see "Methods" section)[7]. As shown in Fig. 1a, the as-synthesized NCs are largely uniform with an average size of approximately 40 nm, although occasionally NCs of different sizes and spherical Cu particles can also be found. The metallic nature of the NCs is confirmed by electron diffraction (inset of Fig. 1a and Fig. S1). Despite our efforts to limit air exposure (see "Methods" section), we still anticipate the presence of thin surface oxide shells on the Cu NCs. Through scanning transmission electron microscopy (STEM) imaging and electron energy loss spectroscopy (EELS), we show that the surface oxide layers on Cu NCs that were subjected to our typical sample handling procedures are about 2 nm in thickness (Fig. S2). As part of our control experiments, we have also considered extreme cases wherein Cu catalysts were deliberately stored or exposed to air for extended periods (Figs. S3 and S4), where the Cu oxide layers were found no thicker than 5 nm. Based on these observations, it is reasonable to assume that the oxide layer on the Cu NCs that will be discussed in this manuscript will typically be around 2 nm and unlikely to exceed around 5 nm.

As-synthesized Cu NCs were tested for $CO_2RR$ at $-1.1\,V_{RHE}$ using a typical H-cell setup and a $CO_2$-saturated $0.1\,M$ $KHCO_3$ electrolyte for 10 h (Table S1, entry 1). Significant changes in reaction selectivity among gas-phase products were observed (Fig. S5), as were the morphological changes of the Cu catalysts and the formation of smaller Cu nanoparticles (Figs. S5 and S6). Identical location TEM (IL-TEM) was applied (Fig. 1b) to follow individual NCs before and after treating the catalysts at desired electrode potentials, using a method modified from the work by Mayrhofer et al.[52] (see "Methods" section). A pair of representative IL-TEM bright field images of the catalyst before and after treating the catalysts at $-1.1\,V_{RHE}$ in a $CO_2$-saturated $0.1\,M$ $KHCO_3$ electrolyte (Table S1, entry 2) for 20 min were shown in Fig. 1c. Evidently, almost all Cu particles within this view exhibit clear signs of morphological changes after the treatment, irrespective of their size and shape. Moreover, as shown in Fig. 1d–f, it is apparent that the morphological changes of the Cu NCs extended deep into the core of the particle, significantly surpassing the estimated thickness of the earlier discussed oxide crusts. This suggests that the reduction or dissolution of surface oxides is unlikely the sole contributor to the morphological changes of the Cu catalysts.

Intriguingly, when we removed $CO_2$ gas from the reaction and repeated the IL-TEM experiments at $-1.1\,V_{RHE}$ using Ar-saturated $0.1\,M$ $KHCO_3$ electrolyte (Table S1, entry 3), a very similar catalyst reconstruction was observed (Fig. 2a, b). In contrast, the morphological changes observed on nanocubes subjected to electrochemical treatment under $0\,V_{RHE}$ (Table S1, entry 4) or slightly more negative potential of $-0.36\,V_{RHE}$ (Table S1, entry 25) in Ar-saturated $0.1\,M$ $KHCO_3$ electrolyte, are primarily confined to less than 5–6 nm (Figs. S8–S10), which matches the estimated thickness of oxide layers. When we replace the electrolyte with Ar-saturated $0.1\,M$ $K_2SO_4$ (Table S1, entry 5) or $0.1\,M$ $K_2CO_3$ (Table S1, entry 6) and applied $-1.1\,V_{RHE}$, significant catalyst reconstruction will again be observed (Fig. S11). These results suggested that CO-related mechanisms[32] are likely not the only possible mechanism responsible for observed morphological changes in Cu catalysts.

To address the limited sampling capacity of TEM, we implemented cyclic voltammetry (CV) as a complementary technique to provide an overall assessment of the structural evolution of Cu NCs[53]. CV is done in an electrochemical cell with a three-electrode configuration as in the IL-TEM setup, except that more catalysts were used to enhance OH- adsorption signals (see "Methods" section). Figure 2e show two voltammograms, recorded in $0.1\,M$ NaOH after the catalysts were treated at about $-1.05\,V_{RHE}$ in Ar-saturated $0.1\,M$ $KHCO_3$ (Table S1, entry 3) for 20 min and 65 min, respectively. This two-stage comparison helps to exclude the influence of surface oxide crusts, which should be reduced quickly (e.g., within 10 min for 40 nm thick

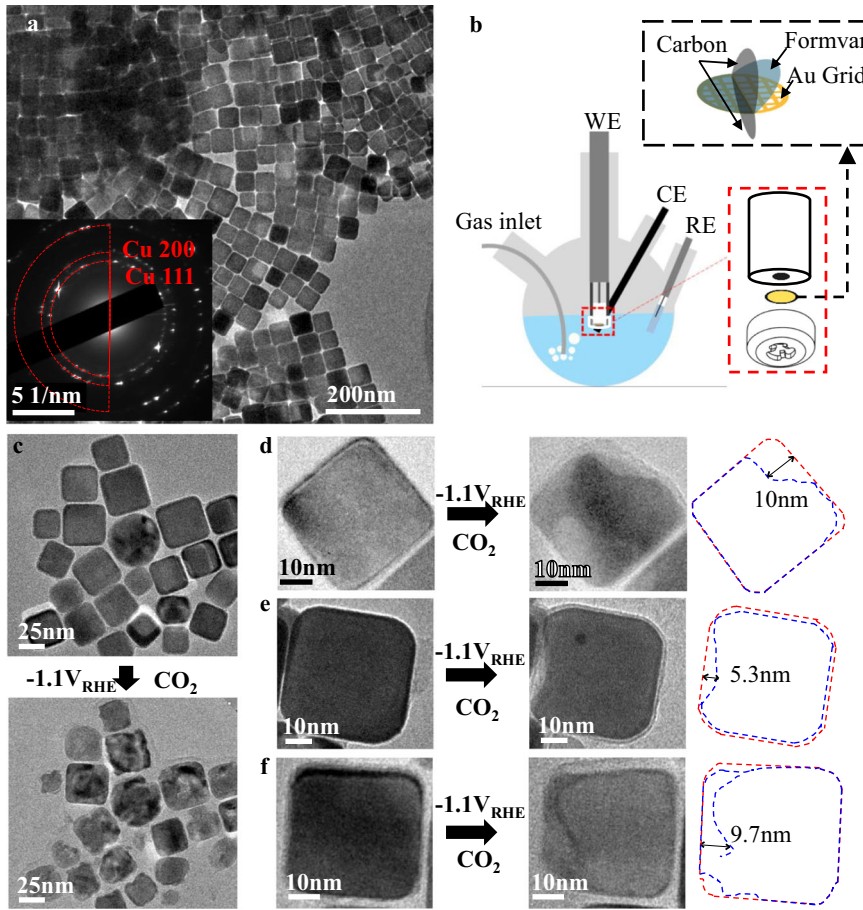

**Fig. 1 | IL-TEM characterization of Cu nanocubes under CO₂RR at −1.1 V_RHE.**
**a** Representative TEM bright field images of Cu nanocubes about 40 nm in size. The inset shows a selected area electron diffraction pattern, confirming the metallic Cu crystal structure. **b** Schematic demonstration of the Identical location TEM (IL-TEM) to study Cu catalysts at cathodic conditions, using a standard RDE electrode with a custom-made PTFE cap and Au TEM grid with Formvar, coated with carbon on both sides. (see "Methods" section). **c–f** Identical location TEM bright field images before and after treating the catalysts at −1.1 V_RHE (iR-corrected) using in a CO₂-saturated 0.1 M KHCO₃ electrolyte (Table S1, entry 2) for 20 min. The morphology changes before (red) and after (blue) CO₂RR are highlighted using outlines, which are significantly beyond the thickness of the surface oxide layer. The current recorded during the IL-TEM experiment can be found in Fig. S7.

$Cu_2O^{11}$) under such a highly cathodic condition. It is therefore reasonable to assume the surface oxides are completely reduced within 20 min under our experimental conditions. The voltammogram of the 65-min treated catalyst differs notably from its 20-min treated analogs in two main aspects: it exhibits a reduced feature associated with the {100}-specific OH-adsorption at around −0.1 V, and an enhanced feature related to surface defects Cu at above +0.3 V. These disparities observed are notably more substantial than those arising from the experimental uncertainties inherent between individual CV experiments (Fig. S12). This reinforces the suggestion that the reduction of surface oxides cannot be the sole reason for the reconstruction of Cu catalysts. It is also important to highlight that morphological changes occur on {100} facets and not just on the particle corners. As shown in Fig. 1c, we also observe significant morphological transformations occurring on spherical particles. These findings suggest that the shape of the particles and the potential field concentration are at least not the determining factors influencing the reconstruction process.

Considering the potential contribution of the hydrogen evolution reaction (HER) to the reconstruction of Cu under CO₂RR relevant conditions[34,35], we carried out additional CV experiments to study the catalysts being treated at about −1.05 V_RHE using 0.05 M H₂SO₄ as the electrolyte (Table S1, entry 7). In this case, we observed only minor changes between voltammograms (Fig. 2f) associated with catalysts treated by 20 min and by 65 min. This suggests that in scenarios where only the HER occurs without the presence of alkali cations, the catalyst

reconstruction is considerably less pronounced. Interestingly, changes in the voltammogram that correspond to significant reconstructions of NCs could be observed again once 0.3 M K₂SO₄ was added into the electrolyte, indicating that the presence of alkali cations in the electrolyte may promote the catalyst reconstruction. IL-TEM results of the catalysts being treated at about −1.1 V_RHE in acidic electrolyte without (Table S1, entry 7) and with K⁺ (Table S1, entry 8) are shown in Figs. S13a and S12b, respectively, which are consistent with the CV observations. It's important to emphasize that in our experiments, the electrolyte was thoroughly de-aerated to eliminate oxygen, which, if present, could lead to further dissolution of Cu, as reported in previous studies[54]. To experimentally demonstrate this, we intentionally reintroduced oxygen into the H₂SO₄ electrolyte by blowing air (Table S1, entry 9), which then resulted in significant copper dissolution observable through both IL-TEM and CV methods (Figs. S13c and S12d). These findings indicate that the further reconstruction of Cu catalysts, occurring after the initial surface oxides are removed, is unlikely due to additional reduction or dissolution of oxides. Increasing the K⁺ concentrations to 0.5 M and 1.5 M (Table S1, entries 10–11) led to apparently more obvious structural changes (Fig. S14). Significant reconstruction was observed when the catalysts were treated in electrolyte with other alkali cations including Li⁺, Na⁺, and Cs⁺ (Table S1, entries 12–19) both at constant voltage (Fig. S15) and at constant current (Fig. S16). This observation is consistent with the cathodic corrosion behavior reported for noble metals[40].

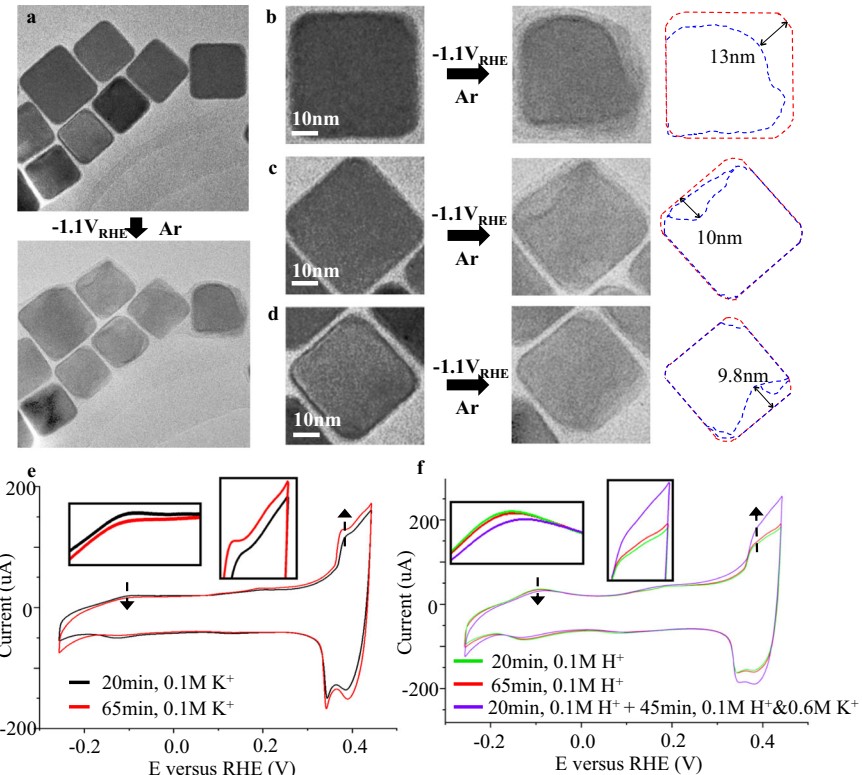

**Fig. 2 | IL-TEM and CV characterization of Cu nanocubes with different reaction conditions. a–d** Representative IL-TEM bright field images of Cu nanocubes before and after 20 min of reaction at about −1.1 $V_{RHE}$ (iR-corrected) in Ar-saturated 0.1 M KHCO₃ (Table S1, entry 3). The morphology changes of individual nanocubes before (red) and after (blue) the reaction are highlighted using outlines, which are significantly beyond the surface oxide layer. The current recorded during the IL-TEM experiment can be found in Fig. S7. **e** Voltammetric profiles of the Cu nanocubes recorded after 20 min (black curve) and 65 min (red curve) of reactions at about −1.05 $V_{RHE}$ in Ar-saturated 0.1 M KHCO₃ (Table S1, entry 3). **f** Voltammetric profiles of the Cu nanocubes recorded after 20 min (green curve) and 65 min (red curve) of reaction at about −1.05 $V_{RHE}$ in Ar-saturated 0.05 M H₂SO₄ (Table S1, entry 7). The purple curve represents the case where the Cu nanocubes first undergo 20 min of treatment in Ar-saturated 0.05 M H₂SO₄, followed by subsequent 45 min of reaction at about −1.05 $V_{RHE}$ (purple curve) after adding additional Ar-saturated K₂SO₄ (0.3 M) (Table S1, entry 8) to the electrolyte while withholding the electrode potential. 0.05 M H₂SO₄ was replaced by 0.05 M H₂SO₄ & 0.3 M K₂SO₄ through successive electrolyte replacement several times with half the volume of the electrolyte replaced each time. The arrows highlighted the changes at the region representing the {100} terraces and the regions representing surface defects on Cu, respectively.

These experimental findings indicate the existence of a previously unexplored yet critical pathway of reconstruction for Cu catalysts under CO₂RR-related reaction conditions. The promoting effect of alkali cations makes the process closely resembling to the phenomenon of alkali cation-induced cathodic corrosion, which is known to cause the reconstructions of many metal electrodes, such as platinum, rhodium, and gold[39]. Koper and colleagues[40,55] proposed that the presence of alkali cations can lead to the formation of intermediate ternary hydride with the cathode metal. Such species leave the electrode but are unstable and will be quickly redeposited onto the cathode, leading to a reconstruction. Although the detailed mechanism of cathodic corrosion is still not fully understood, likely due to the difficulties of identifying these transient species, researchers have already harnessed this phenomenon for applications, such as nanoparticle synthesis[56]. It is important to note that, to the best of our knowledge, alkali cation-induced cathodic corrosion has not been previously reported for Cu catalysts under CO₂RR reaction conditions[27].

### Identifying an onset potential for the cathodic corrosion of Cu catalysts

Another important characteristic of cathodic corrosion is that it occurs only when the electrode potential is more negative than a certain onset potential[55]. This was found to also be the case for Cu catalysts. Figure S17 shows that the changes in CV features associated with {100} facets and defects become suppressed as the electrode potential was adjusted to more positive values of −0.82 $V_{RHE}$, −0.62 $V_{RHE}$, and −0.42 $V_{RHE}$ (Table S1, entries 21–23). We then repeated the experiment to ensure that the total charge transferred is approximately equivalent while varying the electrode potentials. As shown in Fig. 3a, a brief 3-min treatment at about −1.02 $V_{RHE}$ caused much more significant changes in the voltammogram compared to a 120 min additional treatment at a more positive potential of −0.36 $V_{RHE}$ (Table S1, entries 25–26). This indicates a more significant reconstruction in the case of −1.02 $V_{RHE}$ treated catalysts compared to the −0.36 $V_{RHE}$ treated, notwithstanding that the total amount of charge transferred in both cases was nearly identical (i.e., about 0.4 C, Fig. S18). The repeated CV experiments show consistent results as shown in Fig. S20. These results from the CV experiment also agree well with the ex-situ TEM (Fig. S19) and the IL−TEM results (Fig. 3b and Fig. S21) which show that the Cu NCs morphology remains largely intact after 150 min at −0.36 $V_{RHE}$. Note that the images of the Cu nanocubes before the reaction shown in Fig. 3b have thin layers of surface oxides, making them differ slightly from those generating the signal of reference CV (30 min) in Fig. 3a since the 30 min of treatment under −0.36 $V_{RHE}$ or more will remove the surface oxides. We then repeated the experiment, this time comparing the CV of Cu nanocubes experienced a 60-min treatment at −0.36 V and the CV of Cu nanocubes with a 25-min treatment at −0.36 $V_{RHE}$ plus a 30-s treatment at about −1 $V_{RHE}$ using 1 M KOH as the electrolyte (Table S1, entries 23–24), and a similar trend was observed (Fig. S22). In sum, we found that there exists an onset potential of cathodic corrosion for Cu catalysts of approximately −0.4 $V_{RHE}$ under our experimental conditions.

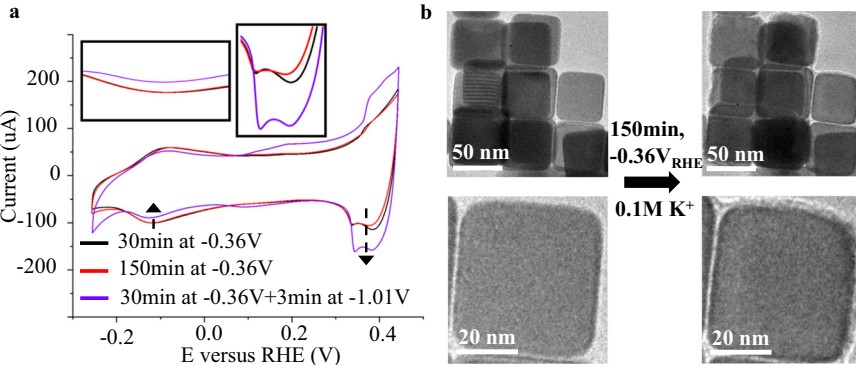

**Fig. 3 | CV and IL-TEM characterization of Cu Nanocubes reacted at −0.36 V$_{RHE}$.** **a** Voltammograms of Cu nanocubes after 30 min (black curve) and 150 min (red curve) of reactions at around −0.36 V$_{RHE}$ (iR-corrected) in 0.1 M KHCO$_3$ are presented (Table S1, entries 20 and 25). The purple curve depicts the profile after an initial 30-min reaction at around −0.36 V$_{RHE}$, followed by a 3-min reaction at approximately −1.01 V$_{RHE}$ (Table S1, entry 26). Highlighted regions show changes related to defects and the {100} terraces. The suppressed {100}-related charge of catalysts treated at −1.01 V$_{RHE}$ versus −0.36 V$_{RHE}$ suggests a decreased population of {100} facets. The defects-related peak at about +0.37 V shows a marked enhancement compared to the results after 30 and 150 min at −0.36 V$_{RHE}$. **b** Representative IL-TEM bright field images of Cu nanocubes before and after 150 min of reaction at −0.36 V$_{RHE}$ in 0.1 M KHCO$_3$.

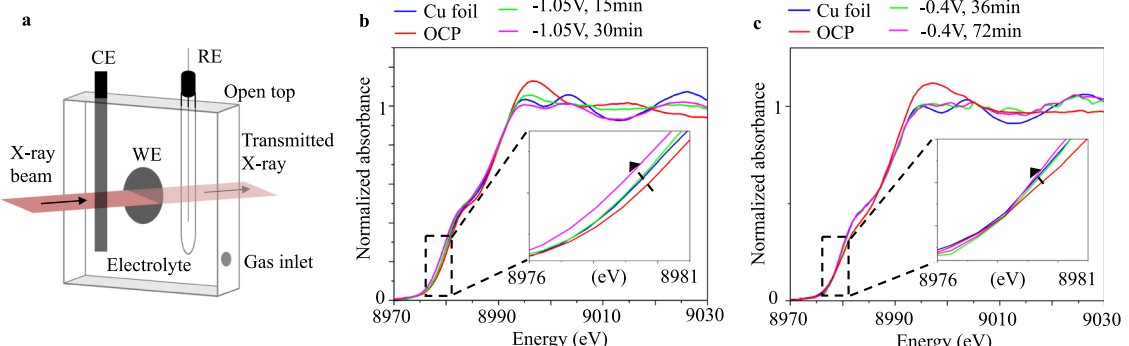

**Fig. 4 | In situ XAFS of 7 nm Cu nanoparticles during CO$_2$RR at different potentials.** **a** Schematic of in situ XAFS cell. In situ, Cu K-edge XANES of spherical Cu nanoparticles about 7 nm in size operated at **b** −0.4 V$_{RHE}$ (iR-corrected), **c** −1.05 V$_{RHE}$ (iR-corrected) in CO$_2$ saturated 0.1 M KHCO$_3$, respectively (Table S1, entries 29–30). The insets show the region around 8980 eV, where a difference from Cu foil (highlighted with arrows) can be observed when the catalysts were operated at deeply cathodic conditions, but not at −0.4 V$_{RHE}$. The current recorded during the in situ XAFS experiment can be found in Fig. S23.

In situ X-ray absorption fine structure (XAFS) spectroscopy studies were carried out in order to monitor the oxidation states of Cu during the CO$_2$RR process. To eliminate the possible self-absorption effect in fluorescence mode[57–59] and thus to have proper comparisons with a Cu foil standard, the transmission mode was adopted for the study. A schematic view of the in situ XAFS cell is shown in Fig. 4a. The electrochemical cell has a three-electrode configuration, with a total electrolyte thickness of 1.5 mm on one side of the electrode. This design ensures adequate transmission of X-ray photons. (see "Methods" section) To ensure that XAS captures the maximum amount of information about the sample surface, we employed the smallest attainable particle size of 7 nm[7]. This size represents the lower limit of controllable synthesis in our laboratory while maintaining a sufficiently large specific surface area. Figures 4b, c show Cu K-edge X-ray absorption near edge fine structures (XANES) recorded in 0.1 M KHCO$_3$ as the Cu nanoparticles being treated at about −1.05 V$_{RHE}$ and −0.4 V$_{RHE}$, respectively (Table S1, entries 29–30). At open circuit potential (OCP), Cu was found in a partially-oxidized state. After 15 min of cathodization at −1.05 V$_{RHE}$, the spectra from the catalyst largely resembled the Cu foil reference, indicating oxide reduction to metallic Cu. Intriguingly, further cathodization at −1.05 V$_{RHE}$ led to a redshift for the shoulder feature relative to the Cu foil reference (Fig. 4b). Similar redshifts on the Pt L edge[60] or Pd K-edge[61] have been previously reported in alloys where the precious metals become electron-rich.

Since this trend was not observed when catalysts were cathodized only at −0.4 V$_{RHE}$ for 72 min (Fig. 4c), we suspect that this feature could be related to the electron-rich form of Cu in the corrosion intermediates. More work is certainly needed to further identify the nature of such intermediates. The XAFS results align with our IL-TEM and CV observations shown in Fig. 3, suggesting an onset potential for alkali cation-induced cathodic corrosion exists for Cu at around −0.4 V$_{RHE}$ under our experimental conditions.

The computational investigation provides detailed thermodynamic information and theoretical insights into the corrosion-reaction kinetics at the atomic scale, which is important to understand the copper cathodic corrosion phenomena observed in our experiments. Inspired by a previous work of Pt cathodic corrosion by Alexandrov and colleagues[41], our goal is to calculate the trend of free energy and its gradient during a hypothetical cathodic corrosion reaction. The trend at different potentials indicates whether the reaction is energetically favored. We performed AIMD simulations of copper dissolution at an interface between a protonated copper (001) surface and a solvated potassium ion. As shown in Fig. S24, the interface model contains a three-layer copper slab with 24 copper atoms (gray), and one potassium ion (blue) is equilibrated to be 3.4 Å from the copper surface. Water molecules (red and white) form an interface with copper, and an implicit water region extending up to 25 Å along the *c* direction - along which the reaction of copper dissolution is

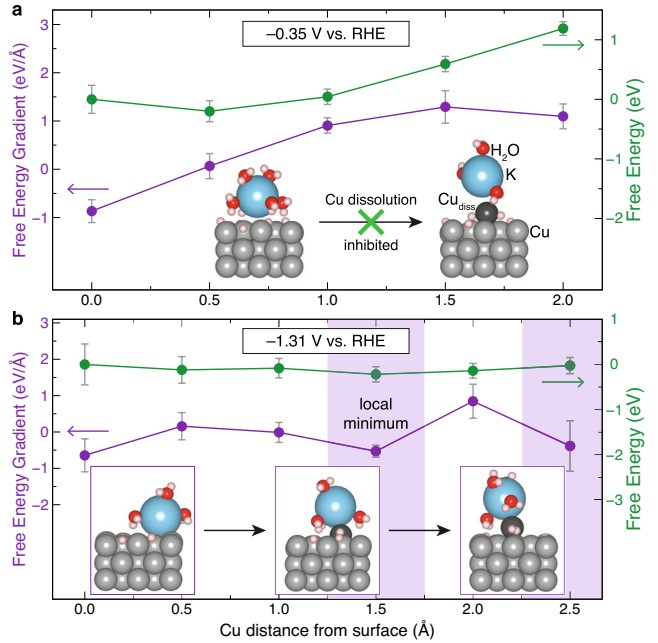

**Fig. 5 | AIMD simulation with constant-potential thermodynamic integration for Cu dissolution at different potentials.** Free energy gradient (purple) and free energy (green) as functions of Cu distance from the surface along the coordinate of the dissolution reaction. The gradient shows an overall monotonically increasing trend at $-0.35\,V_{RHE}$ (**a**) while it shows negative local minima at $-1.31\,V_{RHE}$ (**b**). Inset schematic in (**a**) shows that copper dissolution is inhibited at the less negative voltage of $-0.35\,V_{RHE}$ due to the increasing gradient and high free energy barrier. Inset in (**b**) shows the atomic structures (with Cu distance from the surface of 0 Å, 1.5 Å, and 2.5 Å) of the copper dissolution reaction that happens at the more negative voltage of $-1.31\,V_{RHE}$.

simulated. The bottom plot in Fig. S24 shows the electrostatic potential of the model along the $c$-axis (black curve), and the solvent potential $V_{Sol} = 2.02\,eV$ is calculated from the implicit water region. With the constant $V_{Sol}$, the electrode potential versus standard hydrogen electrode (SHE) can be calculated by $U_{SHE} = V_{Sol}\text{-}E_{Fermi}\text{-}\mu_e$, where $\mu_e = 4.4\,V$ is the absolute potential of SHE, and $E_{Fermi}$ (blue line in the bottom of Fig. S24) is a variable determined by the number of electrons injected into the system.

We performed a temperature ramping followed by a period of the equilibration (see "Methods" section) of the interface model at two different potentials, namely $-0.35\,V_{RHE}$ and $-1.31\,V_{RHE}$. These potentials are achieved by adding into the system ~0.2 and ~3.0 electrons, thereby simulating a grand-canonical ensemble where the system can exchange electrons with a reservoir. Due to limitations in our modeling that prevent the precise adjustment of potential to align with the original experimental conditions at $-1.1\,V_{RHE}$, we also conducted additional cyclovoltammetry (CV) experiments at approximately $-1.3\,V_{RHE}$ (Table S1, entries 24) to better match with the simulation parameters. The results, presented in Fig. S17d, reveal that the catalyst undergoes significant reconstruction at $-1.3\,V_{RHE}$, mirroring the behavior observed at $-1.1\,V_{RHE}$. This consistency validates the choice of parameters used in our modeling. We then conducted constant-potential Blue-Moon AIMD[41,62–66] on snapshots at 300 K taken from a copper dissolution trajectory generated by slow-growth AIMD (see "Methods" section). Here, the Blue-Moon AIMD simulation samples microstates of each window along the dissolution-reaction trajectory. Figures 5a, b show the statistical free energy gradient (purple) and numerically integrated free energy (green) as functions of the reaction coordinate of the copper dissolution. The reaction coordinate represents an approximate distance along with the copper atom gets away (in other words, dissolves) from the metal surface.

At $-0.35\,V_{RHE}$ (Fig. 5a), the free energy gradient shows a monotonic increase up to a distance 1.5 Å far from the basal plane of the copper surface reaching a gradient ~1.3 eV/Å, and the gradients stay over 1 eV/Å at 2.0 Å. This trend leads to an increase in free energy, which reaches ~1.2 eV (115.8 kJ/mol) at 2.0 Å. The increasing gradient and high energy indicate that copper dissolution is energetically unfavored and inhibited. Notably, the structure at 2.5 Å from prior slow-growth AIMD (shown in the inset of Fig. 5a on the right) becomes unstable and cannot be converged by the constrained constant-potential AIMD. This is an indication that the copper atom leaving the surface is unstable at positions far away from the surface (deep into the solvent) and at more positive potentials, e.g., $-0.35\,V_{RHE}$. On the other hand, at more negative potentials $-1.31\,V_{RHE}$ (Fig. 5b), the free energy gradient does not show an increasing trend, and multiple local minima below 0 eV/Å are observed. The trend results in a rather flat free energy curve around 0 eV and indicates energetically favored copper dissolution. The inset in Fig. 5b shows atomic structures along the dissolution trajectory (at 0 Å, 1.5 Å, and 2.5 Å Cu distance from the surface) generated by slow-growth AIMD. At 0 Å, the potassium cation appears partially desolvated to three water molecules and adsorbed at the copper surface. The potassium and copper species remain coordinated during the dissolution process. It is observed that copper species forms hydride with two protons (Cu-H bond lengths of ~1.64 and ~1.57 Å) when it dissolves at a distance of ~2.5 Å from the surface (right-most inset structure in Fig. 5b).

These modeling results show that the potential is indeed an important factor for the dissolution of copper. From these results, we can infer a plausible mechanism[41] of the cathodic corrosion of Cu: at an electrode potential more negative than the onset potential (i.e., about $-0.4\,V_{RHE}$ at our conditions) and with the presence of alkali cations in the solution, Cu forms soluble intermediate ternary hydride species. They may react quickly with adsorbed hydrogen species or water from the solution and get deposited back onto the electrode, leading to the observed reconstruction of the Cu catalyst and the formation of smaller Cu particles, and the reconstruction of Cu catalysts.

## The impact of cathodic corrosion of Cu on its catalytic performance

Previous studies have highlighted the critical roles alkali cations played in CO2RR, as they can stabilize key reaction intermediates[43,67] and suppress the hydrogen evolution reaction (HER)[42,45,68]. In addition, Cu catalysts are commonly operated at about $-1\,V_{RHE}$ or more cathodic conditions to enhance selectivity towards multicarbon products[4]. Therefore, if the alkali cation-induced cathodic corrosion is in play, Cu catalysts will inevitably undergo reconstructions under a typical CO2RR condition and the selectivity advantages brought by morphology control in the pre-catalyst stage will unlikely to be stable. This is exactly what we found. As shown in Figs. S5, 6a, and Fig. S25, although Cu NCs initially offer high selectivity towards $C_2H_4$, the catalyst morphology and catalytic performance evolve over time, producing a lot more $CH_4$ after 10 h (Table S1, entries 1). We then conducted another 10-h CO2RR experiment, this time employing 25 nm spherical Cu nanoparticles (Table S1, entries 31). These spherical Cu nanoparticles generate comparable amounts of C2 products (mainly $C_2H_4$) towards the end of the 10 h reaction, mirroring the performance of aged Cu nanocubes.

Given that controlling catalyst morphology is a major method to adjust the intrinsic activities of electrocatalysts[5,6,69,70], we hope to see if we can switch off the alkali cation-induced cathodic corrosion so that we maintain the morphology and selectivity advantages of Cu NCs over spherical Cu nanoparticles. We choose to do the electrochemical reduction of CO (CORR) since it can give a range of more useful products at more positive electrode potentials compared to CO2RR[50]. CORR experiments were carried out using an electrolyzer flow cell and a gas diffusion electrode (GDE) (see "Methods" section), with 1 M KOH

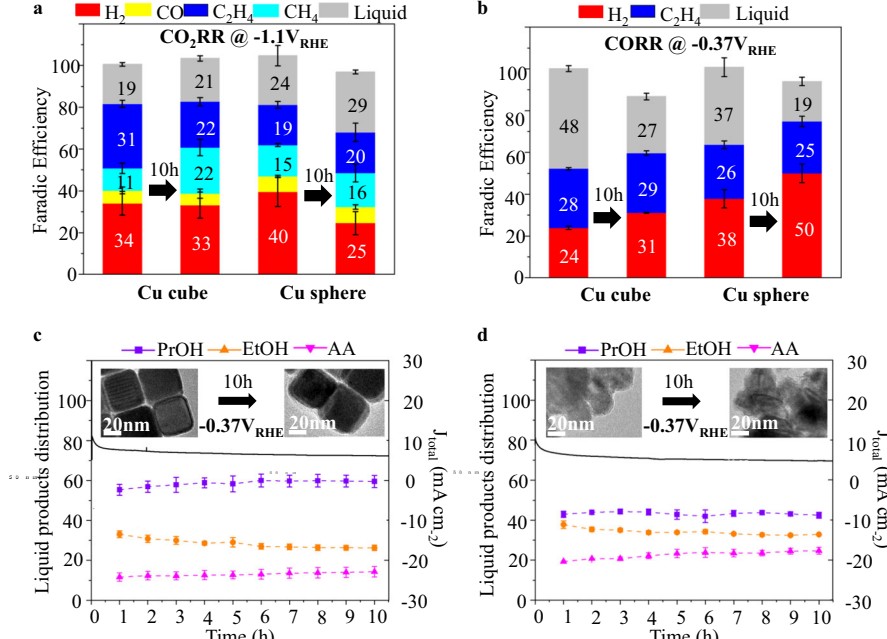

**Fig. 6 | The catalytic performance of Cu nanocubes and 25 nm spherical Cu nanoparticles with or without alkali cation-induced cathodic potential in play.** The comparison of Cu nanocubes and 25 nm spherical Cu nanoparticles when used for 10-h tests of **a** $CO_2RR$ at $-1.1\,V_{RHE}$ (iR-corrected) in 0.1 M $KHCO_3$ (Table S1, entries 1 and 31) and **b** CORR test at $-0.37\,V_{RHE}$ (iR-corrected) in 1 M KOH (Table S1, entries 36 and 40). Liquid product distributions of the CORR experiments as a function of time for **c** Cu nanocubes and **d** 25 nm spherical Cu nanoparticles. The error bars show the standard error of three independent measurements. Insets show ex-situ TEM bright field images of the corresponding sample in each case before and after the CORR experiments.

being used as the electrolyte. Figure S26a–d shows the cell current as a function of time on a Cu nanocube-containing electrode at various CORR potentials. The electrode was treated sequentially for 10 min each at $-0.27\,V_{RHE}$, $-0.36\,V_{RHE}$, $-0.42\,V_{RHE}$, and then $-0.46\,V_{RHE}$ (Table S1, entries 32–35). We initially observed stable or decreasing currents at potentials more positive than $-0.4\,V_{RHE}$, but then noted a significant current increase at more negative potentials, indicating Cu surface roughening and Cu nanoparticle formation. Since the catalyst has already been treated with negative potentials close to $-0.4\,V_{RHE}$ for 20 min, there should be no oxides left in the catalysts. Further experiments, including a 10-h test at $-0.37\,V_{RHE}$ and a separate 80-min test at $-0.58\,V_{RHE}$ (Figs. S26e, f, Table S1, entries 36–37), show consistent results. These findings strongly suggest that the cathodic corrosion is in play when the electrode potential is more negative than an onset potential around $-0.4\,V_{RHE}$ under our experimental conditions. This explanation is supported by the ex-situ TEM results, via which we observed significant reconstruction of Cu NCs that underwent CORR at more negative potential than $-0.4\,V_{RHE}$ (i.e., $-0.7\,V_{RHE}$, $-0.9\,V_{RHE}$) (Fig. S27, Table S1, entries 38–39). The results of these preliminary experiments indicate that the corrosion behavior of Cu catalysts during the CORR closely aligns with our earlier observations in $CO_2RR$. More importantly, they suggest that this corrosion can be switched off by operating the CORR at a potential more positive than approximately $-0.4\,V_{RHE}$ under our experimental conditions.

We subsequently conducted 10-h CORR experiments at $-0.37\,V_{RHE}$ using 25 nm spherical Cu nanoparticles (Table S1, entries 40) and compared the results with that of a similar experiment conducted using 40 nm Cu nanocubes (Table S1, entries 36). The Faraday Efficiencies (FE) of products from these reactions, at both the start and end of the 10-h period, are illustrated in Fig. 6b. Ethylene ($C_2H_4$) was the sole gas-phase product detected via gas chromatography, with FE remaining relatively stable for both Cu NCs (28%–29%) and spherical Cu nanoparticles (26%–25%). An observed increase in hydrogen evolution reaction (HER) for both catalysts is attributed to gradual electrode flooding rather than to changes in catalyst morphology, as

confirmed by TEM images before and after the reaction (insets in Fig. 6c, d). Liquid products were quantified by extracting about 0.6 ml of electrolyte every hour from the initial 15 ml catholyte and analyzing it via NMR (refer to Methods). A progressive decrease in total FE was noted (Fig. S28), likely due to some liquid products crossing over through the ion exchange membrane[71]. The low current and extended duration of the CORR experiments make them prone to such systematic errors. However, the total FE remained acceptably around or above 90%. Importantly, when focusing on the fractions of different liquid products, which were accurately determined via NMR, a consistent distribution was observed for both catalysts. Notably, Cu NCs produced a significantly higher fraction of propanol (about 60%) among all liquid products during the 10-h test (Fig. 6c), a result not replicated when spherical Cu nanoparticles were used (Fig. 6d). Thus, Cu NCs maintained a morphology-induced selectivity advantage in the CORR experiment at $-0.37\,V_{RHE}$, in contrast to the earlier $CO_2RR$ results at $-1.1\,V_{RHE}$. The CORR experiments were also carried out with 7 nm Cu spheres and the findings are basically identical (Fig. S29, Table S1, entries 41–42). This suggests that the corrosion behavior could indeed be switched off at relatively positive potentials, corroborating our hypothesis and indicating a solid understanding of such behaviors.

Given these observations, a legitimate question arises: If the morphology of Cu catalysts is going to be disrupted by the alkali cation-induced cathodic corrosion under typical reaction conditions, why stable operations of $CO_2RR$ using Cu can still be observed? We propose that the dissolution and redeposition of Cu species may reach an equilibrium state, allowing a stable performance to be observed. Rather than a uniform process, we posit that the alkali cation cathodic corrosion may preferentially remove Cu atoms with high surface energies (such as those with low coordination numbers)[55,72]. Similarly, vacancy sites with a high number of neighbors could be preferentially filled during the redeposition process. To visualize this process, we conducted a conceptual simulation using a two-dimensional square lattice model. As shown in Fig. S30, an equilibrium state can indeed be

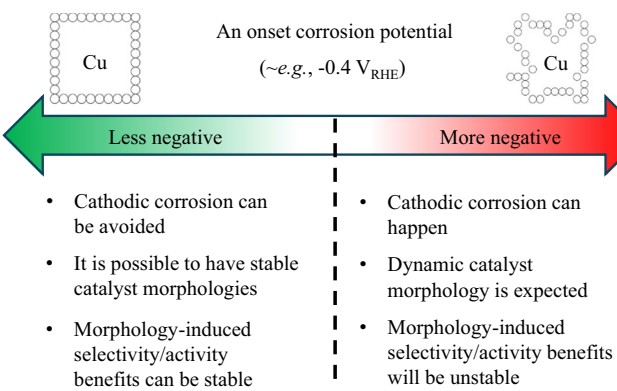

**Fig. 7 | A schematic summary.** With alkali cations used in the electrolyte, at deeply cathodic conditions typically needed for CO$_2$RR, alkali cation-induced cathodic corrosion and therefore a dynamic catalyst morphology can be expected. Long-term morphology-induced selectivity/activity benefits from the catalysts will be limited. On the other hand, if cathodic corrosion can be prohibited (e.g., working at a less negative potential compared to the onset), then catalyst morphology control can potentially bring stable benefits to the catalyst performance.

achieved through continuous dissolution and redeposition processes, provided an appropriate bias exists among surface sites and vacancies based on their coordination numbers. This simulation suggests that the initial morphology of Cu catalysts and their operating conditions would influence how quickly a catalyst could reach this "steady" state. This proposal agrees well with our observation that Cu NCs and spherical Cu nanoparticles eventually yield a similar selectivity profile in CO$_2$RR, as depicted in Fig. 6a.

## Discussion

In this work, we examined the structural evolution of Cu NCs under CO$_2$ electrochemical reduction (CO$_2$RR) and its relevant reaction conditions using a combined approach of identical location TEM (IL-TEM), cyclic voltammetry (CV), and additional experimental and theoretical approaches. Our experimental evidence strongly suggested the existence of a previously unexplored yet critical pathway reconstruction pathway for Cu catalysts – alkali cation-induced cathodic corrosion. Because of that, during a typical CO$_2$RR process, Cu catalysts will undergo surface reconstructions, resulting in dynamic catalyst morphologies. As schematically summarized in Fig. 7, we illustrate that this corrosion behavior in Cu does not necessarily preclude stable operation of CO$_2$RR, but will limit long-term selectivity and activity enhancement from morphology-controlled Cu pre-catalysts. Alternatively, by operating Cu catalysts at less negative potentials than the onset potential (around −0.4 V$_{RHE}$ in our conditions), we show that Cu NCs can maintain their morphology and provide a much more stable selectivity advantage towards propanol over spherical Cu nanoparticles.

Since the alkali cation-induced cathodic corrosion was found in many other metals[39], it should not be too surprising that it also takes place in Cu. Our in situ XAFS results provided some hints on the existence of an electron-rich form of Cu in the corrosion intermediates, which are in line with earlier proposals about the possible involvement of intermediate ternary hydride[40,41] in the cathodic corrosion of other metals. The pursuit to experimentally identify such transient species is still on. Notably, the continuous dissolution of Cu has indeed been observed in a recent ongoing work by Buonsanti and colleagues[73]. As such, cathodic corrosion is likely to be a missing puzzle piece for a mechanistic understanding of the performance of Cu catalysts in CO$_2$RR. For instance, this or other types of possible intermediates need to be taken into consideration when constructing the reaction mechanisms for CO$_2$RR on Cu.

It was predicted[74] that multivalent cations can have a significant effect on CO$_2$RR performance and some experimental exploration was reported[75]. Hence we went on to investigate whether multivalent cations (i.e., Mg$^{2+}$ and Al$^{3+}$) (Table S1, entries 43–45) can also lead to cathodic corrosion of Cu. The results (Fig. S31a) show that, despite we deliberately using an acidic environment (adding H$_2$SO$_4$ until the pH reaches 3), the local basic environment near the Cu electrode likely still result in the precipitation of hydroxides[76,77] which contaminated the Cu electrode. As a result, the CV results cannot be used to detect Cu surface reconstruction or to judge whether the cathodic corrosion occurs. No significant structural changes were observed using TEM (Fig. S31b) after the reaction with multivalent cations. Hence it remains unclear whether multivalent cations can lead to or help to prevent cathodic corrosion.

The occurrence of alkali cation-induced cathodic corrosion in Cu does not negate the possibility of other mechanisms contributing to catalyst reconstruction. During the preparation and peer review of our paper, Magnussen and colleagues[33] reported strong experimental evidence towards a CO-induced mechanism. Buonsanti and colleagues[73] argued that CO-mediated corrosion involves a cationic Cu intermediate. To understand the role of CO in our proposed mechanism of alkali cation-induced cathodic corrosion, we conducted additional AIMD and structural optimization calculations. These examined the process of Cu extraction at the Cu-water interface with both K$^+$ and CO present. Figure S32 reveals that CO tends to out-compete K$^+$ for adsorption sites, and therefore likely inhibiting the corrosion mechanism outlined in Fig. 5. Nevertheless, our simulations also suggest that when CO molecules are distant, K$^+$ can approach the surface to initiate the corrosion process. Given the significant difference in solubilities of K$^+$ and CO in water, the impeding effect of CO should be limited. Recent works by Xu and colleagues[78,79] indicate that CO coverage during CORR reactions is quite low and will decrease further as the electrode potential becomes more negative (e.g., approximately 0.05 monolayer at −0.6 V and 0.005 monolayer at −0.9 V). Moreover, our observations that the threshold potential for the alkali cation-induced cathodic corrosion to take place sit consistently at around −0.4 V$_{RHE}$ with (Fig. 6) or without CO present in the reaction suggest that variations in CO concentration or coverage do not significantly influence the process. Meanwhile, the H$_2$-induced[34,35] mechanisms for changing Cu morphologies could also be in play during a typical CO$_2$RR or CORR experiment. Our findings, as depicted in Fig. 2f, demonstrate that under identical conditions at −1.1 V$_{RHE}$, the morphological changes in Cu caused solely by the presence of H$_2$SO$_4$ in the electrolyte (pH = 1.16, Table S1, entry 7) were notably less pronounced than those observed when alkali cations were present in the electrolyte (pH = 1.28, Table S1, entry 8). An additional experiment conducted with K$_2$SO$_4$, wherein we adjusted the pH to 0.98 using H$_2$SO$_4$ (Table S1, entry 46), yielded significant morphological alterations again (Fig. S33). These findings underscore the pivotal role of alkali cations over the electrolyte's pH in influencing the morphology of Cu. Consequently, it is reasonable to infer that several corrosion mechanisms might coexist, with alkali cation-cathodic corrosion possibly being the dominating process, especially considering the relatively low coverage of CO on the Cu catalyst in the reaction.

The identification of the alkali cation-induced cathodic corrosion in Cu catalysts marks a crucial first step toward devising effective mitigation strategies. In this study, we focused on identifying the occurrence of alkali cation-induced cathodic corrosion. However, our current methodologies, such as IL-TEM and CV, lack the temporal resolution necessary to analyze the kinetics of this corrosion process or to pinpoint its initiation time. It is reasonable to speculate that this type of corrosion will start as soon as the metallic copper is exposed to the electrolyte following the removal of surface oxides, though the oxide thickness and hence the timing may vary slightly from particle to

particle. Our in situ XAFS results (Fig. 4) suggested that oxide layers should be removed within 15 min under typical reaction conditions, consistent with literature[11] that reports completion of reduction within 10 min for a 40 nm $Cu_2O$. Furthermore, Núria López and colleagues recently reported[80] that, at weak reduction potentials (i.e., $0\,V_{SHE}$), it takes approximately 208 s for an oxygen atom to diffuse through a 5 nm copper oxide layer. This simulation result largely agrees with our experimental observations.

It is possible that we have not seen the full potential of nanostructured Cu catalysts because of this corrosion behavior. In this work, we show that under conditions that do not prompt the alkali cation-induced cathodic corrosion, Cu nanocubes could provide stable selectivity advantages over Cu nanoparticles. It could be worthwhile to examine a broader range of nanostructured catalysts under conditions without the disruption of cathodic corrosion, which might get overlooked as they do not necessarily offer optimum performance in the beginning. Future research should also address many remaining questions about cathodic corrosion in Cu, including: Can the onset potential be tuned via changing reaction conditions? Is there any other way to control or even take advantage of the cathodic corrosion in Cu?

# Methods

## Chemicals

Copper (II) acetylacetonate ($Cu(acac)_2$, 99.9%), copper bromide (CuBr, 99.999% trace metals basis), Oleylamine (OAm, 70%), Trioctylphosphine oxide (TOPO, 99%), sodium hydroxide (NaOH, reagent grade, ≥98%), Potassium bicarbonate (ACS reagent, 99.7%), Potassium bicarbonate (Aladdin, 99.99% trace metals basis), Potassium sulfate (Sigma, 99.99% trace metals basis), Lithium sulfate (Sigma, 99.99% trace metals basis), Cesium sulfate (Sigma, 99.99% trace metals basis), Sodium sulfate (ACS reagent, ≥99.0%, anhydrous), Magnesium sulfate (anhydrous, ReagentPlus®, ≥99.5%), Aluminum sulfate hydrate (99.99% trace metals basis). Sulfuric acid ($H_2SO_4$) was purchased from Sigma–Aldrich. Hexane and ethanol were technical grades and used without further purification.

## Synthesis of Cu spherical nanoparticles Cu spheres (~7 nm)

The nanoparticles were synthesized according to a modified procedure[7]. 1.5 mmol of $Cu(acac)_2$, 5.7 mmol TOPO, and 35 mL OAm were loaded into a 100 mL three-necked flask under stirring. The mixture was heated at 80 °C for 30 min under Ar atmosphere. Then the mixture was further heated to 220 °C and reacted for 60 min, generating a reddish solution. The precipitate was cooled down to room temperature, washed five times with excessive ethanol, and dispersed in hexane. 25 nm commercial Cu spherical nanoparticles were purchased from Sigma–Aldrich, 774081, nanopowder, 25 nm particle size.

## Synthesis of Cu nanocubes (~40 nm)

Cu nanocubes were synthesized by adding 0.4 mmol of CuBr, 2 mmol TOPO, and 20 mL OAm into a 100 mL three-necked flask under stirring. The mixture was heated at 80 °C for 30 min under Ar atmosphere. Then, the mixture was further heated to 250 °C and reacted for 60 min, generating a reddish solution. The precipitate was cooled down to room temperature, washed five times with excessive ethanol, and dispersed in hexane.

## Sample storage and transfer to limit air exposure

After being taken out from the inert synthesis environment, the as-synthesized Cu NCs is normally washed for five times, during which the sample might be exposed to air for less than 5 min. The particles were then stored in hexane overnight for the next day's experiments. In the next day's experiment, Cu NCs were dropped onto the electrode or TEM grid and sent to electrochemical reaction within 5 min; otherwise, the electrodes were put into a glovebox and waited for use.

Despite our efforts to limit air exposure (see "Methods" section), we still anticipate a thin surface oxide shell on the Cu NC. Through scanning transmission electron microscopy (STEM) imaging and electron energy loss spectroscopy (EELS), we show that the surface oxide layers on Cu NCs that were subjected to our typical sample handling procedures are approximately 2 nm in thickness (Fig. S2). We have also considered extreme cases wherein Cu catalysts were deliberately stored or exposed to air for extended periods (Figs. S3 and S4) as part of our control experiments, where the Cu oxide layer was found no thicker than 5 nm. Based on these findings, it is reasonable to assume that the oxide layer on the Cu NCs that will be discussed in detail in this manuscript will typically be around 2 nm and unlikely to exceed 5 nm."

## Cyclic voltammetry

Copper surface was treated under cathodic potential before voltammetric characterization. For the cathodization, the working electrode was prepared by dropping cast ethanol dispersion of 20–50 ug copper nanocubes onto the $0.2\,cm^2$ glassy carbon on the rotation disk electrode. The catalyst loadings were kept identical across the samples in the batch used for any individual set of cyclic voltammetry (CV) comparisons. Before the electrochemical test, working electrode was treated by Ar plasma at 30 W for 60 s to remove surface ligands on the copper surface. TEM experiments confirmed that the morphology of the Cu nanocubes remain largely unchanged after the plasma treatment (Fig. S34). A PTFE electrochemical cell with a three-electrode configuration was employed, using a carbon rod through a Luggin capillary as the counter electrode to isolate the oxygen gas bubbles. For cathodization in acid, $Hg/HgSO_4$ was used as a reference electrode. Hg/HgO was used as a reference electrode in alkaline electrolytes. Before cathodization, electrodes were connected to a potentiostat, and potentiostatic was run so that protective cathodic potential was applied to the copper surface immediately when it contacts with the electrolyte to avoid significant oxidation/dissolution of the copper surface at OCP. After cathodic treatment, copper surfaces were transferred and electrochemically characterized at room temperature by cyclic voltammetry in argon-saturated 0.1 M NaOH solutions at a sweep rate of 50 mVs$^{-1}$. Transfer process takes several seconds from the cathodization cell to the characterization cell, during which the copper surface was mostly kept within the cell flowed with Ar to decrease the air exposure. For voltammetric, a PTFE electrochemical cell with a three-electrode configuration was employed, using a carbon rod as the counter electrode and an Hg/HgO electrode as the reference electrode. The voltammetries were normalized to yield identical double-layer (DL) thicknesses within a potential region where the voltammetries exhibited capacitive behavior ($+0.2\,V < E < +0.28\,V$, $+0.02\,V < E < +0.08\,V$)[81]. All electrolytes were pre-electrolyzed for 3–5 h on the clean carbon paper to remove electrolyte impurities. All electrolytes were bubbled with Argon for 1 h before measurement and kept being bubbled during measurement. In this work, all potentials are referred to RHE. The electrode potential was controlled using a CHI760E potentiostat.

## Identical location transmission electron microscopy (IL-TEM)

Identical location TEM was achieved by positioning copper particles on an electrically conductive and corrosion-resistant Au grid that can be used both as a working electrode in an electrochemical cell and as a sample holder during TEM (Fig. 1b). A custom-made PTFE cap with a hole in the center was used to fix the TEM grid onto the surface of glassy carbon on rotation disk electrode. Both sides of the gold grid were covered by carbon deposition to block gold surface from electrochemical reaction and copper particles supported on the carbon film were observed under TEM. By taking overview images at

progressively increasing magnification and using tracking markers, locations of interest in the untreated particles are selected and recorded. This step was repeated after an ex-situ electrochemical test, allowing the evaluation of the identical particles. After the electrochemical test, the grid was taken out and dried on a blotting paper for seconds before it was put into a glovebox and wait for the characterization. The electrochemical reaction was performed in the electrochemical cell with a three-electrode configuration, with a carbon rod through a Luggin capillary as the counter electrode and Hg/HgSO$_4$ for the acidic electrolyte, and Hg/HgO for the alkaline electrolyte as reference electrode. The electrolyte was bubbled with Argon or CO$_2$ for 1 h before measurement. Before immersion of the working electrode into the electrolyte, electrodes were connected to potentiostat, and potentiostatic was run so that protective cathodic potential was applied to the copper surface immediately when it contacts with the electrolyte to avoid further oxidation/dissolution of copper surface in the electrolyte at OCP. After electrolysis for the targeted time, the copper surface was extracted from the electrolyte with holding potential. For Ex-situ TEM imaging, a TEM grid was pressed onto the glassy carbon with a drop of hexane atop to collect the reacted nanoparticles after electrolysis.

## Electron microscopy characterization

Scanning Transmission Electron Microscopy High angle annular dark field (HAADF) imaging, X-ray Energy Dispersive Spectroscopy (X-EDS), and Electron Energy Loss Spectroscopy (EELS) studies were conducted on a JEOL JEM ARM 200CF in the National University of Singapore, equipped with an Oxford Instrument X-ray Energy Dispersive Spectrometer and a Gatan Image Filter (GIF) system. In a typical STEM-EELS measurement, a probe current of around 100pA was utilized, with a dwell time of approximately 20 ms per pixel. The convergence half angle is about 29 mrad and the collection half angle is about 36 mrad for the STEM-EELS measurement. To generate the elemental map, background removal, signal integration, and multiple linear least square fitting were performed using Gatan Digitalmicrograph software.

## X-ray absorption fine structure spectroscopy

XAFS spectra around the Cu K-edge (8979 eV) were collected at the XAFCA beamline of Singapore Synchrotron Light Source (SSLS) in transmission mode[82]. The double-crystal Si (111) monochromator was used for measurements. The size of the beam at the sample position was ca. 2 mm (h) × 1 mm. All XAFS data analyses were performed with the Athena software package to extract XANES[83]. The electrochemical setup with three electrodes and a Polyetheretherketone electrochemical cell was used for the reaction. The electrochemical setup consisted of an Ag/AgCl reference electrode, a carbon counter electrode, and a catalyst-modified carbon cloth (ca. 10 mm × 10 mm geometric area) working electrode. The catalyst ink was dropping cast onto carbon cloth. The electrolyte was first bubbled with CO$_2$ gas for 30 min and then a constant potential was applied. The XAFS measurements were recorded simultaneously while performing the chronoamperometric measurements.

## CO$_2$RR measurements

The experiments were implemented at an electrochemical workstation (CHI760E) under ambient pressure and temperature. A gastight two-compartment cell configuration was employed with a counter electrode of Pt mesh and reference electrode of Ag/AgCl electrode and electrolyte of 0.1 M KHCO$_3$. The electrolyte was prepared by dissolving KHCO$_3$ in DI water and stored in a plastic volumetric flask. A piece of anion-exchange membrane with 2.5*2.5 cm$^2$ area and 70um thickness was served to separate the counter and working electrodes in the two compartments. Working electrodes were prepared by evaporating hexane solution containing around

50ug Cu NCs within a squared area of 1cm$^2$ on the glassy carbon substrates. The loading was calculated by multiplying the Cu concentration of the ink with the volume of the ink dropped onto the substrate. Here, the loading amounts were chosen to yield a decent Faradaic efficiency towards multicarbon products. The working electrodes were then held at a constant bias of −1.1 V vs. RHE using chronoamperometry for a set time of up to 10 h. During electrolysis, CO$_2$ was constantly bubbled through the electrolyte at a flow rate of 20 sccm. The flow rate of CO$_2$ was controlled with a mass flow controller. All potentials were corrected for iR-loss compensation by electrochemical impedance spectroscopy (EIS) and converted to potentials versus the reversible hydrogen electrode (RHE) using the following equation (1): $E(vs.RHE) = E(vs.Ag/AgCl) - iR + 0.197 V + 0.0591 V \times pH$. The gaseous products were monitored by an online gas chromatograph, equipped with a thermal conductivity detector (TCD) detector for H$_2$ and a flame ionization detector (FID) detector for hydrocarbon quantification. High-purity He gas (99.999%) was used as the carrier gas for all compartments of the GC. A GC run repeats every 29.5 minutes. The GC was calibrated using calibration curves of analytical gas standards. The calculation of Faradaic efficiency (FE) for gaseous products based on the following equation (2): $FE_i(\%) = j_i/j_{total} \times 100 = zF \times (QP_0 V_i/RT)/j_{total}$. Here, the total current was measured by the potentiostat, and the partial currents were obtained from the areas of GC chromatogram peaks where Vi is the volume concentration of gas product i based on the calibration of GC. Q is the flow rate of CO$_2$, z is the number of electrons transferred for reduction to product i, F is the Faradaic constant, $P_0$ is atmosphere pressure, T is room temperature, and R is the ideal gas constant, 8.314 J·mol/K. 1 ml electrolyte was collected from both cathode and anode chamber at 1 h and 10 h of reaction and the containing liquid products were analyzed by 1H Nuclear Magnetic Resonance (NMR) spectrum (Bruker 400 MHZ system). The Faradic Efficiency (FE) of each product was calculated by following equation (3): $FE(\%) = amount\ of\ the\ product \times n \times F/C \times 100$, where n represents the number of electrons transferred, F represents the Faradaic constant, and C represents the total Coulomb number.

## CORR measurements

The measurements were conducted in a flow cell, which was assembled from the sequential stacking of a gas chamber, a catalyst-loaded GDE cathode, a catholyte chamber (where an Ag/AgCl reference electrode was located), an anion-exchange membrane, an anolyte chamber, and an IrO$_2$-loaded Ti plate anode. The working electrodes of Cu cube and Cu sphere were prepared as follows: 1 mg Cu cube/Cu sphere power was dispersed in a mixed solution with 1 mL of ethanol and 4 μL of 5 wt% Nafion solution to form a catalyst ink. The ink was dispersed in an ultrasonic machine for more than 10 minutes. And then the ink was sprayed onto carbon fiber (YSL-30T) paper with 1 mg cm$^{-2}$ to form a smooth catalyst film.

During COR measurements, 21 sccm of CO was continuously fed to the working electrode through the mass flow controller. And the electrolyte (1 M KOH) was pumped to circulate through the flow cell at the rate of 15 mL min$^{-1}$ by a peristaltic pump. The electrolyte was prepared by dissolving KOH in DI water and stored in a plastic bottle. The Bio-Logic VMP3 multichannel potentiostat/galvanostat was applied for the electrochemical measurements. The reference electrode was Ag/AgCl and the potential was converted to RHE with necessary iR compensation by equation (1). The ohmic-drop correction of the potentials applied was carried out manually using the resistance measured by the electrochemical impedance spectroscopy under open circuit potentials.

Gaseous reduction products were injected into an online gas chromatograph (GC, Shimadzu 2014) to analyze the concentration of each gas product. And liquid products were collected in the centrifuge

tube and were identified by 1H Nuclear Magnetic Resonance (NMR) spectrum (Bruker 400 MHZ system). 0.6 ml electrolyte was taken every 1 h during the 10 h CORR from the initial 15 ml circulated catholyte. The Faradic Efficiency (FE) of each product was calculated by equation (3).

## Computational investigation

All calculations were performed using Kohn-Sham DFT as implemented in VASP[84]. The unknown exchange and correlation functional was approximated by the revised Perdew−Burke−Erznzerhof (RPBE)[85]. Van der Waals (vdW) interactions were treated with the D3 method with zero damping[86]. The wavefunctions of the valence electrons were expanded as plane waves up to a cut-off of 400 eV[87,88]. Projector-augmented wave (PAW) potentials were used for core electrons[89]. A smearing width of 0.3 eV together with a second-order Methfessel−Paxton scheme[90].

To model the dissolution reaction, a three-layer copper slab containing 24 atoms modeling the Cu(001)/water interface was put in direct contact with 26 explicit water molecules followed by a region where the effect of water is described by an implicit electrostatic model. The 26 water molecules fill a 15 Å-long region to maintain the water density ~1 g/cm$^3$. A K ion was initially added 2.14 Å from the Cu(001) surface and then equilibrated. The Cu surface was pre-relaxed with DFT and 26 water molecules were pre-organized with classic molecular dynamics using LAMMPs. The implicit water was applied using VASPsol[91,92].

Ab initio molecular dynamics (AIMD) simulations were performed at the Γ point with a time step of 1.0 fs and hydrogen mass of 3 amu. Simulation of a negative voltage to match the experimental electrode potential is realized by adding extra electrons into the cell and determined from the electrostatic potential of implicit water and the system's Fermi level. Two different voltages were simulated, namely −1.31 V$_{RHE}$ by adding 3 electrons and −0.35 V$_{RHE}$ by adding 0.2 electrons. Temperature was ramped to 300 K during the first 3 ps using the Nose−Hoover thermostat[93,94], and then the system was equilibrated at 300 K for 5 ps. The slow-growth method[95] was employed to generate Cu extraction trajectory by controlling the distance between the Cu atom and bottom Cu layer, between the K atom and bottom Cu layer, and between the K and Cu atom. A trajectory of 2.5 Å in a period of 5 ps was generated for each system of different voltages. Five windows were taken from the trajectory and their free energy gradient was calculated using constrained constant-Fermi-level Blue-Moon AIMD[41,62−66]. The reaction coordinate - the Cu distance from the surface - was fixed for each window, and the Fermi level was kept within ±0.05 eV of the target value from the 5 ps equilibration by adding/removing electrons when it gets out of range.

## Data availability

The computational data associated with this study is available on zenodo  https://doi.org/10.5281/zenodo.8059745. The experiment-related source data were also uploaded to zenodo. https://doi.org/10.5281/zenodo.10259206. Source data are provided with this paper.

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

## Acknowledgements

We acknowledge the funding support by the National Research Foun-dation (NRF) Singapore, under its NRF Fellowship (NRF-NRFF11-2019-0002). S.L. would like to acknowledge the support of the Chinese Scholarship Council. A.B.W. would like to acknowledge support via the Presidential Young Professorship (PYP) start-up funds at the National University of Singapore (A-0009245-05-00). L.W. acknowledges finan-cial support from A*STAR (Agency for Science, Technology and Research) under its LCERF1 program (Award No U2102d2002), Centre for Hydrogen Innovations at NUS (CHI-P2022-06), and NRF Fellowship (NRF-NRFF13-2021-0007). P.C. and Y.L. acknowledge funding from the National Research Foundation under the NRF Fellowship NRF-NRFF12-2020-0012. J.L. acknowledges the support from the Agency for Science, Technology, and Research (A*STAR) under its AME IRG Grant (Project No. M22K2c0082). K.Y., A.C.N., and T.F.J. acknowledge support for evaluat-ing Cu corrosion using mass spectrometry from the Liquid Sunlight Alli-ance, which is supported by the Basic Energy Sciences program of the US Department of Energy Office of Science through the Fuels from Sunlight Hub under award number DE-SC0021266. We want to express our special thanks to Dr Qiang Gao and Prof. Huiyuan Zhu from the University of Virginia for preparing some Cu NCs for online mass spectroscopy ana-lysis. We also wish to thank Mr Han Xu and Mr Yukun Zhao from Professor Jiong Lu's group at NUS, Singapore for their kind and timely help on getting the NMR results during the paper-revision process.

## Author contributions

S. Liu and Q. He conceived the idea and led the project. S. Liu carried out the sample preparation, IL-TEM experiments, $CO_2RR$, and CV measure-ment under the supervision of Q. He and J. Lu, with many advices from L. Wang and A. Wong, and the technical help from H. Xu, X. Li, and W. Zang. M. Su and W. Liu also helped with Cu NCs synthesis and some TEM screening. Y. Li implemented the grand-potential AMID methodology carried out the simulation part of the study under the guidance of P.

Canepa. D. Wang, S. Liu carried CORR measurement under the guidance of L. Wang, helped by Y.L. Wang; S. Liu and S. Xi carried out the in situ XAFS study. K. Yan, A. Nielander carried out preliminary online mass spectroscopy study under the guidance of T. Jaramillo. S. Liu, Y. Li, and Q. He drafted the paper and all authors participated in the manuscript discussion and polishing.

## Competing interests

The authors declare no competing interests.
