## [Peer Review File · Nature Communications]

Alkali Cation Induced Cathodic Corrosion in Cu ElectrocatalystsREVIEWER COMMENTS

Reviewer #1 (Remarks to the Author):

Liu et al describe in their manuscript Cu nanoparticle morphological transformations at two different values of cathodic potential that should undergo CO₂ reduction reaction (CO₂RR). This system has been studied extensively in the literature and it receives much attention. Their experimental design includes identical location TEM and operando XAFS while the interpretation is assisted with AIMD simulations. The authors call the transformations cathodic corrosion at the CO₂RR cathodic potential, they argue that this has not been reported in the past, and they suggest to not use Cu nanocubes for CO₂RR and use them instead for CORR. This is indeed puzzling; this comparison takes place for the two reactions in dissimilar potentials, dissimilar pH values, and dissimilar ageing conditions. I would suggest the authors to revise their manuscript and focus on their main findings. Unfortunately, the novelty of this work is difficult to comprehend and the results do not support the conclusions. In detail:

- The authors call the morphological change “cathodic corrosion” and they back their argument with metal-based corrosion articles. None of these articles refer to corrosion of metallic Cu. Cu has a very low oxidative potential and Cu surfaces are already oxidized from the beginning of the process, as the authors also show for their nanoparticles. Under CO₂ reduction reaction conditions of Cu, it is known that first the oxide reduces and then the metallic copper participates in the reaction by dissolution and reprecipitation. Hence, it seems that what the authors name corrosion is the reduction of the oxide and the subsequent dissolution of Cu, a reconstruction process that is well known, so there is no “puzzle” here. Cathodic corrosion is a wide term that refers to electrochemical degradation of materials that interact with their environment; it does not bring insights into the mechanisms of degradation.
- The authors refer to an “onset” potential for the “cathodic corrosion”, i.e., the reconstruction. However, there are only two potential values, two data points. Data for -0.6 and -0.8 V vs RHE should be provided to argue towards this “onset” value (which is relevant for the CORR and the selectivity as well).
- In general, there is a huge confusion across the manuscript concerning the degradation pathways of the Cu nanoparticles. For example, in Page 4 and also Page 15, the authors refer to the fragmentation pathway. This is completely different from dissolution and redeposition of Cu species. Fragmented particles are composed of primary particle fragments, while redeposited particles are nucleated from dissolved species. Thus, the statement that “etching out/ redeposition leads to the fragmentation” is erroneous. There are no in-situ data in the manuscript and therefore such statements are fully unsupported, and wrong in this case.
- A major issue is that each of the methods applied for the study (i.e., IL TEM, XAFS, and computational simulations) were used at dissimilar conditions. For example, the XAFS (Fig.4) was performed for 7 nm Cu nanoparticles – most likely nanospheres –, and this is somehow compared to IL-TEM of Cu nanocubes of 40 nm in size. The molecular dynamic simulations (Fig.5) were performed at different potentials than the measurements, with a big discrepancy at the very negative potentials. The authors’ use of -1.31 V vs RHE for their simulations is particularly problematic. Even the selectivity results between CO₂RR and CORR was performed for different ageing conditions (10 hours in the first case vs 7 hours in the second case). Please show the results for 10 hours for CORR. Hence, all of these results cannot be used to support each other.
- The effect of alkali metal cations on the processes is not adequately supported in the manuscript. The CVs do not show distinct differences from metal cation containing electrolytes. There have been reports on Cu dissolution under acidic environment at cathodic potentials (<https://doi.org/10.1023/B:PROM.0000013107.65745.b0>). Thus, when using only proton containing electrolyte it is expected to result in different CV features than K⁺ containing electrolyte. However, the CV for the 0.05 M H₂SO₄ and 0.3M K₂SO₄ do not show much of a difference in terms of features related to the surface defects in Cu. Please provide IL TEM (or ex situ TEM) images of particles before/after treatment in acidic electrolyte to prove the hypothesis.
- Importantly, the current value is not normalized on the CV data and thus comparison of the features cannot be performed. Even if the loading is the same (which should be reported), the electrochemically active surface area (ECSA) of the catalysts will fluctuate from the initial value upon degradation of the particles.
- Also, in the methodology, the authors report that for the preparation of the catalysts for the CV studies, Ar plasma was used to remove the ligands (something that does not appear to have been

performed for all other experimental studies. Is there any effect of the Ar plasma treatment on the intactness of Cu NCs and their degradation behavior with respect to the Cu NCs not treated with plasma? Previous reports have indicated faster degradation process as opposed to ligand capped Cu NCs.

- In Supplementary Figure S5, the trend for the partial current density of HER and CO₂RR products differs from the trend of Faradaic efficiency (directly proportional to partial current density) reported by other researchers on similar catalysts of the same size and at the same potential (<https://doi.org/10.1038/s41467-018-05544-3>). Please elaborate on the possible reason of this deviation from the results earlier published.
- Faradaic efficiency of more than 100 % is unphysical (Fig.6b, initial Cu nanospheres). In the same Fig.6b, what does the "other" refer to and how come there is no "liquid" as in Fig.6a)?
- The last paragraph of the introduction would account to (unsupported) conclusions and therefore it is misplaced and should be removed.

Reviewer #2 (Remarks to the Author):

I eagerly read the work by Liu et al. as, during my research career, I've been myself dealing with the same research question very often. Besides, I found it pretty reassuring that the proposed insights overlap significantly with very recent reports (ChemRxiv. Prepr. 2022, DOI: 10.26434/chemrxiv-2022-3cr9k; Nat. Catal. 2023, DOI: 10.1038/s41929-023-01009-z. <https://doi.org/10.1038/s41929-023-01009-z>.) Such overlap definitely confirms that there is indeed new science here!

Specifically, Liu et al. employed identical location TEM, cyclic voltammetry, in situ XAFS and finally DFT/AIMD to show that, beyond a certain potential threshold (quantified as -0.4 V vs RHE), Cu reconstructs through a mechanism very much aligned with cathodic corrosion. They proved such phenomenon to be independent from the presence of oxidic phases and that it occurs preferentially in presence of alkali cations. Remarkably, via X-ray absorption fine structure, the authors proposed such reconstruction to take place through the formation of Cu anionic species. Such proposed mechanism was validated through density functional theory ab initio molecular dynamic simulations on a protonated Cu surface with a neighboring water cation and a > 0.1 nm solvation layer consisting of explicit water molecules.

Although I find the work very valuable and robust, I think the title and the structure is a bit misleading. Both the previously mentioned papers did observe Cu dissolution (or continuous Cu oxidation-reduction) phenomena which, to the best of my understanding, is conceptually very similar to what the authors here define cathodic corrosion. Thus, in my opinion their work does not really unveil cathodic corrosion, instead they rather reconcile previous results by proposing a unique mechanism, here called cathodic corrosion. Therefore, I personally suggest the authors to change the scope of their manuscript and my feedback below points toward this direction, i.e. to further benchmark the proposed mechanism of an already observed phenomenon.

(1) The recent work by Magnussen et al. (Nat. Catal. 2023, DOI: 10.1038/s41929-023-01009-z. <https://doi.org/10.1038/s41929-023-01009-z>.) is not mentioned anywhere. Although it is pretty recent so the authors may not have read it during the preparation of the manuscript, it should now be considered. Besides, I personally think the work by Buonsanti et al. also needs additional attention, rather than just a small note close in the conclusion.

(2) Few different electrolytes have been included and, consequently, different anions have been considered. Could the authors investigate further the role of anions in driving the reconstruction? Besides H and CO species, also bicarbonate, carbonate, hydroxide, and SO₄ can bind to the surface, so these species may actually be co-promoter (or main responsible) of corrosion phenomena.

(3) In Fig. S11 the authors show that larger size cations lead to an increased extent of reconstruction. Besides, Fig. S10 reports that higher K⁺ concentration leads to higher corrosion rates. Larger size cations like Cs⁺ or K⁺ determine higher reaction rates for CO₂ reduction and H₂O reduction (see Nat. Catal. 2021, 4, 654–662. <https://doi.org/10.1038/s41929-021-00655-5>; J. Am. Chem. Soc. 2022, 144, 1589–1602. <https://doi.org/10.1021/JACS.1C10171>.) and those reaction rates further increase for an increased concentration of cation at the surface. Higher reduction rates definitely lead to increased concentration of relevant reaction

intermediates/products (OH⁻, CO) at the surface. Which is the main role of alkali cations in the reconstruction process then? Do they stabilize the anionic Cu species suggested or is the increased corrosion simply due to a higher CO concentration at the surface due to cation-promoted CO₂ activation? By carrying out fixed current density studies (to fix the amount of CO/OH⁻ produced) or using inorganic cations (which promote CO₂, yet do not interact with intermediates, see *J. Am. Chem. Soc.* 2023, 145, 16787–16795. <https://doi.org/10.1021/jacs.3c04769>.) the authors may be able to determine whether there is a direct promotion of corrosion by cations or simply an indirect one due to higher production of CO or OH⁻.

(4) Do the authors have any suggestion on the specific form of these Cu anionic complexes formed during reconstruction? Buonsanti et al. did report Cu⁺ species during Cu dissolution instead (*ChemRxiv. Prepr.* 2022, DOI: 10.26434/chemrxiv-2022-3cr9k.) Could the authors motivate such difference in oxidation state between the proposed intermediates? Note that in Buonsanti et al.'s work, the overall complexes are generally anions, yet the Cu center is positively charge.

(5) Could the authors repeat the study for a Cu system without any pristine oxidic phase? Raaijman et al. suggested that, without initial oxidation, no reconstruction occurs (*ACS Appl. Mater. Interfaces* 2021, DOI: 10.1021/acsmi.1c13989. <https://doi.org/10.1021/acsmi.1c13989>.) and I wonder whether this holds also under these conditions.

(6) "This suggested that H₂-related mechanisms are unlikely to be the primary contributors to Cu reconstruction". By reading this statement, I think the computational model employed to benchmark the experimental system is completely at odds with experimental results. If H₂-related mechanism are not the primary contributors, why should cathodic corrosion occur via Cu-H_x species as indicated in Figure 5? Magnussen et al. proposed that Cu reconstruction is mediated by CO adsorption (see *Nat. Catal.* 2023, DOI: 10.1038/s41929-023-01009-z. <https://doi.org/10.1038/s41929-023-01009-z>), while Buonsanti et al. investigated several different intermediates (*ChemRxiv. Prepr.* 2022, DOI: 10.26434/chemrxiv-2022-3cr9k). Thus, different intermediates besides H should be tested either computationally or through selected reduction studies (e.g. CO reduction at high overpotential) to further clarify the main responsible for cathodic corrosion.

MINOR POINTS

(1) the word "unreasonable" at page 12 should be deleted.

(2) In Figure 6, panel d, there is a typo. The potential in the graph should be -1.1 V vs RHE instead of -0.33 V vs RHE.

(3) Page 23, 4th line. In "cm⁻²" -2 should be set as superscript.

(4) Page 23, 14th line. In "And iquid products", "iquid" must be replaced with "liquid".

(5) In addition to reporting experimental data on a Zenodo repository, the authors should upload their DFT simulations on the ioChem-BD repository, among the suggested platforms by Nature itself (<https://www.nature.com/sdata/policies/repositories>).

Point by point response. New texts or figures in the paper are highlighted.

Reviewer #1 (Remarks to the Author):

Liu *et al* describe in their manuscript Cu nanoparticle morphological transformations at two different values of cathodic potential that should undergo CO₂ reduction reaction (CO₂RR). This system has been studied extensively in the literature and it receives much attention. Their experimental design includes identical location TEM and operando XAFS while the interpretation is assisted with AIMD simulations. The authors call the transformations cathodic corrosion at the CO₂RR cathodic potential, they argue that this has not been reported in the past, and they suggest to not use Cu nano cubes for CO₂RR and use them instead for CORR. This is indeed puzzling; this comparison takes place for the two reactions in dissimilar potentials, dissimilar pH values, and dissimilar ageing conditions. I would suggest the authors to revise their manuscript and focus on their main findings. Unfortunately, the novelty of this work is difficult to comprehend and the results do not support the conclusions. In detail:

Response 1.1: We appreciate the referee's acknowledgement of this topic's significance. To clarify our work's novelty and address the concerns in general, we think it might be helpful to do a briefly recap. The paper contains the three interconnected parts:

Part 1: Our research reveals that Cu catalysts undergo reconstruction under CO₂RR conditions in a manner unexplained by known mechanisms, suggesting a new mechanism at play. This conclusion is drawn from experimental observations (Table R1) obtained by examining of Cu-nanocubes (a well-defined starting point), mainly using identical location TEM for high-resolution, before-and-after particle imaging, supplemented by cyclovoltammetry for broader statistical validation.

Table R1 – Key observations indicating the existence of an additional mechanism for Cu catalyst reconstruction.

Previously reported major mechanisms	i) oxide reduction/dissolution and redeposition (e.g., Jung et al. , JACS, 2019)	ii) H ₂ -induced reconstruction mechanism (e.g., Matsushima et al. , JACS, 2009)	ii) CO-induced reconstruction mechanism (e.g., Magnussen and colleagues , Nature Catalysis. 6 , 837–846 (2023))
Our main observations	a. Reconstruction extent in Cu-nanocubes surpasses surface oxide thickness. b. Cu catalysts keep reconstructing even after the reduction of surface oxide completes.	c. Much lesser reconstruction in Cu observed with H ⁺ replacing K ⁺ in the electrolyte, despite increased HER reactions. d. Significant reconstruction in Cu catalysts upon reintroducing K ⁺ .	e. Cu reconstruction takes place regardless of CO presence in the reaction.

Part 2: Our further experiments and simulations identified two distinct characteristics in the reconstruction of Cu catalysts, including (i) the requirement of having alkali cations in the electrolyte and (ii) the existence of a threshold potential. These findings align with the so-called “cathodic corrosion”, a phenomenon where metal unexpectedly corrodes under cathodic conditions. This phenomenon was recently reviewed by Hersbach and Koper

(*Cathodic corrosion: 21st century insights into a 19th century phenomenon. Current Opinion in Electrochemistry*. **26**, 100653 (2021)).

Although the reconstruction of Cu catalysts has been observed and reported before, no one has specifically linked it to the “cathodic corrosion” that was better known for other metals such as Au, Pt etc. (See Response 2.1, Table R2 for more details) Recently, Koper and colleagues reported a study on possible cathodic corrosion of Cu with relatively low spatial resolutions and claimed that it does not happen in Cu. (Raaijman *et al.*, *ACS Appl Mater Interfaces* **13**, 48730-48744, doi:10.1021/acsami.1c13989 (2021)). Our novel approach has enabled us to demonstrate the contrary, linking Cu catalyst reconstruction to alkali cation induced “cathodic corrosion” for the first time.

Part 3: We went on to explore the effects of "cathodic corrosion" on the catalytic performance of Cu catalysts. We argued that in a typical CO₂RR reaction (e.g., -1.1 V_{RHE} and the catalyst directly in contact with KHCO₃ electrolyte), cathodic corrosion on Cu is inevitable. However, its impact on catalyst stability varies with the catalyst's initial state. For instance, such a reconstruction will mean less for catalysts with inherently complex, atomically rough surfaces, suggesting a stable performance can still be observed. Conversely, catalysts with regular morphologies, like cubic structures with (100) facets, reconstruction caused by cathodic corrosion will make a more noticeable change to the catalyst, impacting their long-term selectivity and activity.

To demonstrate this, we conducted long-term (10-hour) CO₂RR tests at -1.1 V_{RHE}, comparing Cu nanocubes with spherical Cu particles. Our results showed that although Cu nanocubes initially favored C₂H₄ selectivity, this advantage diminished to match that of Cu spheres after 10 hours. We then carried out CORR experiments at -0.37 V_{RHE}, which is below our estimated threshold potential of -0.4 V_{RHE} for cathodic corrosion to take place. We wish to show that cathodic corrosion can be switched off and the shape-induced selectivity can be harnessed in a long term. And that is exactly what we found (new F10 hours data added to **Figure 6**, more on this in **Response 1.5**). We chose to do CORR at -0.37 V_{RHE} because it can produce a wide range of products unlike CO₂RR at such potential. This isn't a recommendation for specific CORR potentials or one should do CORR over CO₂RR, but rather providing another piece of evidence of cathodic corrosion's occurrence and controllability.

Therefore, we hope the referee would now agree that recognizing alkali cation induced cathodic corrosion in Cu catalysis indeed brings important new insights that will lead to future development of potential strategies to enhance CO₂RR catalysts (e.g., methods to mitigate or leverage on this behavior). We hope this explanation clarifies our paper's novelty. We've also revised the paper for clearer communication, as we'll continue to detail in the subsequent Q&As.

(Reviewer 1) The authors call the morphological change “cathodic corrosion” and they back their argument with metal-based corrosion articles. None of these articles refer to corrosion of metallic Cu. Cu has a very low oxidative potential and Cu surfaces are already oxidized from the beginning of the process, as the authors also show for their nanoparticles. Under CO₂ reduction reaction conditions of Cu, it is known that first the oxide reduces and then the metallic copper participates in the reaction by dissolution and reprecipitation. Hence, it seems that what the authors name corrosion is the reduction of the oxide and the subsequent dissolution of Cu, a reconstruction process that is well known, so there is no “puzzle” here. Cathodic corrosion is a wide term that refers to electrochemical degradation of materials that interact with their environment; it does not bring insights into the mechanisms of degradation.

Response 1.2: As detailed in **Response 1.1** and **Table R1**, we implemented many precautions in our experiments to ensure our findings were not just due to the reduction or dissolution of surface oxides. For instance, in our cyclic voltammetry (CV) experiments (Figure 2), we compared Cu samples subjected to different reaction times, such as 20 and 65 minutes. This allowed us to conclude that the additional changes in the 65-minute sample were not attributable to surface oxide reduction, but something else. Following Referee #2's suggestion, we repeated these CV experiments in a nearly anaerobic condition (O₂ < 0.1ppm), yielding same results. (more on this see Response 2.5 and Figure R4)

The term “cathodic corrosion”, discussed in Hersbach and Koper, *Cathodic corrosion: 21st century insights into a 19th century phenomenon. Current Opinion in Electrochemistry*. **26**, 100653 (2021), refers to the unexpected corrosion of metal under cathodic conditions. Indeed all previous report on cathodic corrosion was for other metals, and it has not been reported for Cu before – exactly why this work is novel.

We have now also changed the title of the paper to “**Alkali Cation-Induced Cathodic Corrosion** in Cu Catalysts - a Missing Puzzle Piece for Understanding Their Performance in the Electrochemical Reduction of CO₂”, so that this work is better differentiated from other possible mechanisms of cathodic corrosion of Cu, like CO-induced mechanism appeared in a recent paper (Magnussen *et al. Nature Catalysis* 2023, <https://doi.org/10.1038/s41929-023-01009-z>), as well as the ChemRxiv paper by Buonasanti and colleagues (Vavra, J. *et al. Cu+ transient species mediate Cu catalyst reconstruction during CO₂ electroreduction. ChemRxiv*, doi:<https://doi.org/10.26434/chemrxiv-2022-3cr9k> (2022)). It is highly likely that different pathways exist and the alkali-induced mechanism discussed in this work is just one of them. Please note that in that Nature Catalysis paper by Magnussen and colleagues, no control experiment without introducing CO was carried out. It is likely that the alkali-induced mechanism may also be relevant in their case.

(Reviewer 1) The authors refer to an “onset” potential for the “cathodic corrosion”, i.e., the reconstruction. However, there are only two potential values, two data points. Data for -0.6 and -0.8 V vs RHE should be provided to argue towards this “onset” value (which is relevant for the CORR and the selectivity as well).

Response 1.3: A key feature of “cathodic corrosion” is its onset potential (e.g., Hersbach *et al.*, Nature Communications, 7, 12653 (2016)). In the original submission, we have reported -0.6V_{RHE} data in what is now **Figure S14**, to which we have also incorporated additional data for -0.82V_{RHE} and -1.29V_{RHE} as requested. Clear changes can be observed from the CVs of catalysts being treated for 65 mins under -0.82V_{RHE} or -1.29V_{RHE}, compared to their 20mins treated counterparts, as expected. These findings are in line with our expectations and uphold our estimated threshold value of around -0.4 V_{RHE}. The additional data do not alter the overall narrative of the paper.

Updated Figure S14 with added data at -0.82V_{RHE} and -1.29V_{RHE} can be found below.

The main text at the middle of page 10 now reads as

“**Supplementary Figure S14** show that the changes in CV features associated with {100} facets and defects become suppressed as the electrode potential was adjusted to more positive values of -0.82V_{RHE}, -0.62V_{RHE} and -0.42V_{RHE}.”

(Reviewer 1) In general, there is a huge confusion across the manuscript concerning the degradation pathways of the Cu nanoparticles. For example, in Page 4 and also Page 15, the authors refer to the fragmentation pathway. This is completely different from dissolution and redeposition of Cu species. Fragmented particles are composed of primary particle fragments, while redeposited particles are nucleated from dissolved species. Thus, the statement that “etching out/ redeposition leads to the fragmentation” is erroneous. There are no in-situ data in the manuscript and therefore such statements are fully unsupported, and wrong in this case.

Response 1.4: We apologize for the confusion. The word “fragmentation” was meant to describe the formation of small particles. This term is used as this in some publications such as H. Jung, *et al.* *JACS*. 141, 4624–4633 (2019). But we agree with the reviewer that it causes confusions as people can very well thought we were talking about a direct breakdown of the particle. Hence we have made the necessary changes in the paper, removing the term fragmentation entirely from the paper. The changes include:

Pg 5

Old text –“ Significant changes in reaction selectivity among gas-phase products were observed (**Supplementary Figure S5**), as were the **fragmentation** and morphological changes of the Cu catalysts.”

New text –“ Significant changes in reaction selectivity among gas-phase products were observed (**Supplementary Figure S5**), as were **the morphological changes** of the Cu catalysts.”

Pg 4

Old text - “Our experiments and modelling suggest that Cu can be etched out by forming transient complexes with alkali cations, which can then be redeposited as metallic Cu under cathodic conditions, leading to experimentally observed **fragmentation** and reconstruction during the CO₂RR reaction.”

New text - “Our experiments and modelling suggest that Cu can be etched out by forming transient complexes with alkali cations, which can then be redeposited as metallic Cu under cathodic conditions, leading to experimentally observed **reconstruction of the Cu catalyst and the formation of smaller Cu particles** during the CO₂RR reaction.”

Pg 16

Old text - “One possible explanation is that the current increases because of the **fragmentation** of Cu due to the cathodic corrosion.”

New text - “One possible explanation is that the current increases because of **the redeposition of small Cu nanoparticles formed** due to the cathodic corrosion.”

(Reviewer 1) A major issue is that each of the methods applied for the study (*i.e.*, IL TEM, XAFS, and computational simulations) were used at dissimilar conditions. For example, the XAFS (Fig.4) was performed for 7 nm Cu nanoparticles – most likely nanospheres –, and this is somehow compared to IL-TEM of Cu nanocubes of 40 nm in size. The molecular dynamic simulations (Fig.5) were performed at different potentials than the measurements, with a big discrepancy at the very negative potentials. The authors' use of -1.31 V vs RHE for their simulations is particularly problematic. Even the selectivity results between CO₂RR and CORR was performed for different ageing conditions (10 hours in the first case vs 7 hours in the second case). Please show the results for 10 hours for CORR. Hence, all of these results cannot be used to support each other.

Response 1.5: We thank referee for this comment. In this work, we didn't observe a significant particle size effect, at least not within the range between 7-40 nm. We think this could be due to the fact that the catalysts usually undergo considerable reconstruction after long-term standard CO₂RR reactions because of the cathodic corrosion, making the initial particle size less relevant (see the **updated Figure S5** in the Response 1.9).

For our XAFS studies, we selected 7 nm particles to ensure optimal data quality, which gave important hints towards identifying unknown corrosion intermediates. We've noted in the paper: *"To ensure that XAS captures the maximum amount of information about the sample surface, we employed the smallest attainable particle size of 7 nm. This size represents the lower limit of controllable synthesis in our laboratory while maintaining a sufficiently large specific surface area."*

In the original Part 3, where we aimed to showcase selectivity differences between spherical and cubic particles under varying cathodic corrosion conditions, we continued using 7 nm particles. To address concerns and demonstrate the consistency of our findings, we've now repeated the long-term stability tests (10 hours as requested) multiple times with Cu nanocubes and commercial 25 nm Cu spheres. The **updated Figure 6** (see **Response 1.10**) shows that at -1.1 V_{RHE}, Cu nanocubes initially exhibit higher selectivity towards C₂H₄, which gradually aligns with that of spherical particles over time. In contrast, during the 10-hour CORR test, the selectivity towards C₂H₄ and the distribution of liquid products remained distinct between Cu nanocubes and spheres. These findings are in line with previous results (previously Figure 6, now put into **Figure S24**) obtained with 7 nm particles.

For the question about why we used -1.3V instead of -1.1V for AIMD, we had explained it in the method section as follows: *"Simulation of negative voltage to match the experimental electrode potential is realized by adding extra electrons into the cell and determined from electrostatic potential of implicit water and system's Fermi level."* Since we can only add or remove an integer number of electrons to the system, we do not have the freedom to fine tune the potential and exactly match the experiment. To ease the concern, we have now added additional experiment data to match the simulation at -1.3V instead, which is shown in **Figure S14** and was discussed in **Response 1.3**. 10 hour CORR experiments are also added and are shown in the new **Figure 6** discussed above as requested.

We have added the following sentences to the manuscript on page 13.

*"Due to limitations in our modeling that prevent the precise adjustment of potential to align with the original experimental conditions at -1.1 V_{RHE}, we also conducted additional cyclic voltammetry (CV) experiments at approximately -1.3 V_{RHE} to better match with the simulation parameters. The results, presented in **Supplementary Figures S14(d)**, reveal that the catalyst undergoes significant reconstruction at -1.3 V_{RHE}, mirroring the behavior observed at -1.1 V_{RHE}. This consistency validates the choice of parameters used in our modelling."*

Additional data show consistent results and do not alter the narrative of the paper.

(Reviewer 1) The effect of alkali cations on the processes is not adequately supported in the manuscript. The CVs do not show distinct differences from metal cation containing electrolytes. There have been reports on Cu dissolution under acidic environment at cathodic potentials (<https://doi.org/10.1023/B:PROM.0000013107.65745.b0>). Thus, when using only proton containing electrolyte it is expected to result in different CV features than K⁺ containing electrolyte. However, the CV for the 0.05 M H₂SO₄ and 0.3M K₂SO₄ do not show much of a difference in terms of features related to the surface defects in Cu. Please provide IL TEM (or ex situ TEM) images of particles before/after treatment in acidic electrolyte to prove the hypothesis.

Response 1.6: To address the reference to I. V. Kreizer et al.'s paper ("Partial Reactions of Copper Dissolution under Cathodic Polarization in Acidic Media," Prot. Met., 2004), we would like to firstly clarify that their findings on Cu dissolution pertain to scenarios involving oxygen. In the paper they stated: "*An anomalous dissolution of copper cathodically polarized in acidic media is observed only in the presence of dissolved oxygen, but no anomalous copper dissolution in deaerated media.*" Our experiments were conducted with thoroughly deaerated electrolyte solutions, so the corrosion we observed cannot be attributed to the dissolution of Cu oxides. More discussions about why our observed Cu reconstruction is not just due to dissolution of oxides can be found in earlier Responses 1.1 and 1.2.

Regarding referee's comment that the CVs do not show distinct differences with metal cation-containing electrolytes, we suspect this may be due to unclear labelling in the original Figure 2f, which could have led to misinterpretation. We've therefore remade this figure with clearer labels. As illustrated in Figures 2e and the updated 2f, we compared CVs from catalysts treated in different electrolytes for 20 and 65 minutes. This comparison helps us discern that any additional changes beyond 20 minutes are not due to the initial reduction of surface oxides, which occurs rapidly. It's apparent that for electrolytes containing K⁺, the differences between the two CVs are significant in the areas corresponding to Cu (100) and surface defects. In contrast, when using electrolytes containing only H⁺, the CVs (20min and 65min) appear quite similar. However, reintroducing K⁺ into the electrolyte after the initial 20 minutes results in noticeable differences between the two sets of CV curves (purple and green).

Figures 2e and updated 2f were used to demonstrate that just having H⁺ without alkali cations in the solution do not cause significant reconstruction.

Furthermore, we have updated Figure S10, adding panels c and d, to demonstrate the effect of deliberately introducing O₂ (air) into the system. This resulted in severe dissolution, akin to what was described in the paper mentioned by the referee. This experiment confirms that our observation is distinct from oxide dissolution. The dissolution mechanism due to the presence of O₂, while noteworthy, is less relevant to our study because oxygen should be absent in CO₂RR conditions. If O₂ were present, the oxygen reduction reaction (ORR) would dominate instead of CO₂RR.

Updated Figure S10 Representative IL-TEM bright field images of Cu nanocubes before and after 30 mins of reaction at about $-1.1V_{RHE}$ in Ar saturated $0.05M H_2SO_4$ (a), $0.05M H_2SO_4$ & $0.3M K_2SO_4$ (b) and Air saturated $0.05M H_2SO_4$ (c), showing that dissolution of copper was more evident with presence of K^+ and air. (d) Voltammetric profiles of the Cu nanocubes recorded after 60 mins reactions at about $-1.05V_{RHE}$ in Ar saturated $0.05M H_2SO_4$ (black curve) and Air saturated $0.05M H_2SO_4$ (red curve), indicating that presence of oxygen promotes the generation of defects feature under cathodic potential.

To make this point clearer, we have modified the maintext as follows and have cited the paper mentioned by the referee.

Pg 7-8, Interestingly, significant changes in voltammogram that correspond to the reconstruction of NCs could be observed again once $0.3M K_2SO_4$ was added into the electrolyte, indicating that the presence of alkali cations in the electrolyte likely promoted the catalyst reconstruction. IL-TEM results of the catalysts being treated at $-1.1V_{RHE}$ in acidic electrolyte with and without K^+ are shown in (Supplementary Figures S10(a, b)), which are consistent with the CV observations.

Pg 8, adding text "It's important to emphasize that in our experiments, the electrolyte was thoroughly de-aerated to eliminate oxygen, which, if present, could lead to further dissolution of Cu, as reported in previous studies.⁵¹ To experimentally demonstrate this, we intentionally reintroduced oxygen into the H_2SO_4 electrolyte by blowing air, which then resulted in significant copper dissolution observable through both IL-TEM and CV methods (Supplementary Figures S10(c,d)). These findings indicate that the further reconstruction of Cu catalysts, occurring after the initial surface oxides are removed, is not attributable to additional reduction or dissolution of oxides."

Reference 51. I. V. Kreizer, I. K. M., N. M. Tutukina & I. D. Zartsyn Partial Reactions of Copper Dissolution under Cathodic Polarization in Acidic Media. *Protection of Metals* **40**, 23-25 (2004).

(Reviewer 1) Importantly, the current value is not normalized on the CV data and thus comparison of the features cannot be performed. Even if the loading is the same (which should be reported), the electrochemically active surface area (ECSA) of the catalysts will fluctuate from the initial value upon degradation of the particles.

Response 1.7: We thank referee for the comments. The loading information is now included in the method section and we apologise for its omission in the original submission. The revised method section for Cyclic voltammetry now has the following part: “For the cathodization, the working electrode was prepared by dropping cast ethanol dispersion of 20-50 ug copper nanocubes onto the 0.2cm² glassy carbon on rotation disk electrode. The loading was kept identical for the same batch of samples that were used for CV tests.” For the cathodization, the working electrode was prepared by dropping cast ethanol dispersion of 20-50ug copper nanocubes onto the 0.2cm² glassy carbon on rotation disk electrode. The catalyst loadings were kept identical across the samples in the batch used for any individual set of cyclic voltammetry (CV) comparisons.

The CV data reported were indeed normalized to the capacitance current (double layer region), based on the method used in Raaijman, *et al.*, *Journal of The Electrochemical Society*, 2021 168 096510. This is equivalent of a normalization to the ECSA as they are proportional. We acknowledge the referee's point that the catalyst's surface area changes with structural evolution, which is indeed reflected in the double layer capacitance variation. As shown in Figure R1, which is the raw CV data of the ones shown in Figure 2f without normalization. This data demonstrates that after adding K⁺, the two samples exhibit significantly different double layer capacitances, indicative of altered surface areas due to catalyst reconstruction. The normalization process used the current values from the potential range that showed apparent capacitance behaviour, highlighted by red squares. These details were clarified in the main text, and the relevant reference was added.

Text was added to the Method section, pg 22: “The voltammetries were normalized to yield identical double layer (DL) thicknesses within a potential region where the voltammetries exhibited capacitive behavior ($+0.2\text{ V} < E < +0.28\text{ V}$, $+0.02\text{ V} < E < +0.08\text{ V}$).⁷³”

73. Raaijman, S. J., Arulmozhi, N., da Silva, A. H. M. & Koper, M. T. M. Clean and Reproducible Voltammetry of Copper Single Crystals with Prominent Facet-Specific Features Using Induction Annealing. *Journal of The Electrochemical Society* **168**, doi:10.1149/1945-7111/ac24b9 (2021).

Figure R1 – Cyclic voltammetric profiles without normalization of the Cu nanocubes recorded after 20 mins in 0.05M H₂SO₄ (green curve) and after additional 45min treatment in 0.05M H₂SO₄ & 0.3M K₂SO₄ (purple line). The red squares mark the capacitance current region used for normalization.

(Reviewer 1) Also, in the methodology, the authors report that for the preparation of the catalysts for the CV studies, Ar plasma was used to remove the ligands (something that does not appear to have been performed for all other experimental studies. Is there any effect of the Ar plasma treatment on the intactness of Cu NCs and their degradation behavior with respect to the Cu NCs not treated with plasma? Previous reports have indicated faster degradation process as opposed to ligand capped Cu NCs.

Response 1.8: We thank referee for the comments. We have added TEM of the Ar plasma treated Cu NCs to Updated version of Fig S27. To show that no obvious changes in Cu NC morphology. One sentence was added in the method section in pg 22: “Before electrochemical test, working electrode was treated by Ar plasma at 30W for 60s to remove surface ligands on copper surface. **No obvious changes in the morphology of Cu NCs induced plasma treatment (Supplementary Figure S28).**”

New Figure S28 Cu nanocubes after Ar plasma at 30W for 60s.

By “previous reports”, we assumed that the referee was referring to the paper by Buonsanti and colleagues (J. Huang et al., Nat. Commun. 9, 3117 (2018)). In that work, authors claimed that the ligand removal leads to faster performance degradation. We carried out an 10h CO₂RR test at -1.1 V_{RHE} for Cu NCs that was pretreated by Ar plasma. The result is shown in Figure R2. As compared to the stability of Cu NCs without plasma treatment, the stability behavior remained almost identical. Hence we do not consider Ar plasma would alter the narrative of the paper.

Figure R2 – CO₂RR performance over 10h at -1.1 V_{RHE} for the Cu nanocubes with (a) and without (b) Ar plasma treatment. (same as Figure S5)

(Reviewer 1) In Supplementary Figure S5, the trend for the partial current density of HER and CO₂RR products differs from the trend of Faradaic efficiency (directly proportional to partial current density) reported by other researchers on similar catalysts of the same size and at the same potential (<https://doi.org/10.1038/s41467-018-05544-3>). Please elaborate on the possible reason of this deviation from the results earlier published.

Response 1.9: We repeated the CO₂RR stability test (updated **Figure S5**) and added error bars from three independent measurements. This demonstrates the robustness of our results. We suspect that the different performance could be due to the different catalyst loading (50 ug/cm² vs 10 ug/cm² used in the work J. Huang *et al.*, *Nat. Commun.* **9**, 3117 (2018)). These could lead to different catalyst structure after reconstruction or different reaction microenvironment, which might be responsible for the observed difference in catalytic performance. This work focused on the variations of the catalytic properties due to catalyst reconstructions, which were clearly demonstrated.

Updated Figure S5 Electrocatalytic performance over 10h of the Cu nanocubes at -1.1 V_{RHE}. The figure show gas phase products only. The error bars represent standard error of mean from three independent measurements. Inset: Representative TEM bright field images of Cu nanocubes at different stages of the CO₂RR time. Scale bars represent 20nm.

(Reviewer 1) Faradaic efficiency of more than 100 % is unphysical (Fig.6b, initial Cu nanospheres). In the same Fig.6b, what does the “other” refer to and how come there is no “liquid” as in Fig.6a)?

Response 1.10: We thank referee for this comment. Accurately calculating Faraday Efficiencies (FE) requires precise knowledge of the quantities of gas and liquid products, typically measured via different methods (like gas chromatography and NMR), each carrying inherent measurement errors. Thus, slight deviations in reported FE from 100% are not unusual. To validate our findings' robustness, we have repeated multiple 10-hour stability tests, the results of which are presented in the updated Figure 6. For CO₂RR experiments at -1.1V_{RHE}, and the FE is approximately 100% within the margin of error. As depicted in Figure 6, the initial selectivity advantage of Cu nanocubes towards C₂H₄ over Cu spheres diminishes after 10 hours. The liquid product distribution from CO₂RR, predominantly formic acid, has been added to the new Figure S21 as requested. The main benefit of using Cu cubes (or (100) planes) in CO₂RR lies in generating C₂₊ products, primarily C₂H₄, in the gas phase.

For long-term CORR stability tests at around -0.37V_{RHE}, the relatively low current (< 10 mA) introduces additional measurement challenges, making the experiments prone to systematic errors typically negligible in higher current settings. The updated Figure 6b and new Figure S24 illustrate that while initial total FEs are 100% (within error), they gradually decrease over time to 80-90% level after 10 hours. This decline is likely due to the loss of liquid product crossing over to the anode side of the cell, a known issue in the field (e.g., N. Wang, R. K. Miao, G. Lee, A. Vomiero, D. Sinton, A. H. Ip, H. Liang, E. H. Sargent, Suppressing the liquid product crossover in electrochemical CO₂ reduction. *SmartMat.* **2**, 12–16 (2021)). Nonetheless, as Figures 6c and d indicate, the product distributions in the liquid phase and the FE for C₂H₄ remain relatively stable for both Cu cubes and spheres. This consistency suggests that by operating below the threshold potential for cathodic corrosion, we can effectively maintain the selectivity advantage of Cu nanocubes over the 25 nm Cu spheres, contrasting sharply with the CO₂RR scenario where cathodic corrosion is active (Figure S5, Response 1.9). These are also consistent with the original figure produced with 7nm Cu spheres (now **Figure S25**)

Following changes were made to the manuscript (pg 16-17) “We subsequently conducted 10-hour CORR experiments at -0.37 V_{RHE} using both Cu nanocubes (NCs) and 25 nm spherical Cu nanoparticles. The Faraday Efficiencies (FE) of products from these reactions, at both the start and end of the 10-hour period, are illustrated in **Figure 6b**. Ethylene (C₂H₄) was the sole gas phase product detected via gas chromatography, with FE remaining relatively stable for both Cu NCs (28%-29%) and spherical Cu nanoparticles (26%-25%). An observed increase in hydrogen evolution reaction (HER) for both catalysts is attributed to gradual electrode flooding rather than to changes in catalyst morphology, as confirmed by TEM images before and after the reaction (insets in **Figures 6c & 6d**). Liquid products were quantified by extracting about 0.6 ml of electrolyte every hour from the initial 15 ml catholyte and analyzing it via NMR (refer to **Methods**). A progressive decrease in total FE was noted (**Supplementary information Figures S24**), likely due to some liquid products crossing over through the ion exchange membrane.⁶⁸ The low current and extended duration of the CORR experiments make them prone to such systematic errors. However, the total FE remained acceptably around or above 90%. Importantly, when focusing on the fractions of different liquid products, which were accurately determined via NMR, a consistent distribution was observed for both catalysts. Notably, Cu NCs produced a significantly higher fraction of propanol (about 60%) among all liquid products during the 10-hour test (**Figure 6c**), a result not replicated when spherical Cu nanoparticles were used (**Figure 6d**). Thus, Cu NCs maintained a morphology-induced selectivity advantage in the CORR experiment at -0.37 V_{RHE}, in contrast to the earlier CO₂RR results at -1.1V_{RHE}. The CORR experiments were also carried out with 7 nm Cu spheres and the findings are basically identical(**Supplementary information Figures S25**). This suggests that the corrosion behavior could indeed be switched off at relatively positive potentials, corroborating our hypothesis and indicating a solid understanding of such behaviors.”

New Figure 6. 10hour stability test of (a) CO₂RR at -1.1V_{RHE} and (b) CORR test at -0.37V_{RHE} for both Cu nanocubes and 25 nm Cu spheres. Distribution of liquid products in CORR as a function of time for (c) Cu nanocubes and (d) spherical Cu nanoparticles at -0.37V_{RHE} in 1 M KOH.

New Figure S24 CORR Faraday efficiency as a function of time for (a) Cu nanocubes and (b) 25 nm Cu spheres

(original Figure 6, now Figure S25) Stability tests for (a) 10 hour CO₂RR at -1.1V_{RHE} and (b) 7 hour CORR test at -0.33V_{RHE} for Cu nanocubes and 7nm Cu spheres.

Replotted distribution of liquid products in CORR as a function of time for (c) Cu nanocubes and (d) spherical Cu nanoparticles at -0.37V_{RHE} in 1 M KOH.

(Reviewer 1) The last paragraph of the introduction would account to (unsupported) conclusions and therefore it is misplaced and should be removed.

1.11 Response: We are grateful for the referee's suggestion. To address their concern, we have revised the last paragraph of the introduction. We've removed sentences that previously gave the impression of conclusions. The remaining content has been modified to serve as a preview of the paper, aiming to pique readers' interest and encourage them to continue exploring the rest of the document.

Removed the following from the last paragraph of the introduction

“This finding enhances our understanding of the behaviour of technically important Cu catalysts under electrocatalytic reaction conditions. We show that alkali-cation cathodic corrosion can occur on Cu catalysts, which, to our best knowledge, has not been considered in previous mechanistic discussions for Cu catalysts.”

The last paragraph of the introduction now reads as:

“Since CO₂RR usually requires highly cathodic conditions and having alkali cations in the electrolyte,⁴⁰⁻⁴² Cu catalysts will inevitably undergo surface reconstructions that lead to dynamic morphologies. We argue that having this dynamic morphologies does not necessarily preclude stable operation using Cu catalysts as the catalyst morphology may potentially reach an equilibrium. However, because of the alkali cation induced cathodic corrosion, many approaches towards engineering the Cu catalyst morphology will unlikely to bring long-term selectivity/activity benefits.⁴³⁻⁴⁵ Alternatively, this observation suggests that strategies can be designed to mitigate selectivity changes by reducing the extent to which the cathodic corrosion of Cu can occur. To this end, by operating Cu catalysts at less negative potentials than the threshold (e.g., -0.4 V_{RHE}) in the CO electrochemical reduction reaction (CORR),⁴⁶⁻⁴⁸ we will demonstrate that the cathodic corrosion can be switched off, allowing Cu nanocubes can maintain a stable selectivity advantage over spherical Cu nanoparticles.”

Reviewer #2 (Remarks to the Author):

I eagerly read the work by Liu et al. as, during my research career, I've been myself dealing with the same research question very often. Besides, I found it pretty reassuring that the proposed insights overlap significantly with very recent reports (ChemRxiv. Prepr. 2022, DOI: 10.26434/chemrxiv-2022-3cr9k; Nat. Catal. 2023, DOI: 10.1038/s41929-023-01009-z. <https://doi.org/10.1038/s41929-023-01009-z>.) Such overlap definitely confirms that there is indeed new science here!

Specifically, Liu et al. employed identical location TEM, cyclic voltammetry, in situ XAFS and finally DFT/AIMD to show that, beyond a certain potential threshold (quantified as -0.4 V vs RHE), Cu reconstructs through a mechanism very much aligned with cathodic corrosion. They proved such phenomenon to be independent from the presence of oxidic phases and that it occurs preferentially in presence of alkali cations. Remarkably, via X-ray absorption fine structure, the authors proposed such reconstruction to take place through the formation of Cu anionic species. Such proposed mechanism was validated through density functional theory ab initio molecular dynamic simulations on a protonated Cu surface with a neighboring water cation and a > 0.1 nm solvation layer consisting of explicit water molecules.

Although I find the work very valuable and robust, I think the title and the structure is a bit misleading. Both the previously mentioned papers did observe Cu dissolution (or continuous Cu oxidation-reduction) phenomena which, to the best of my understanding, is conceptually very similar to what the authors here define cathodic corrosion. Thus, in my opinion their work does not really unveil cathodic corrosion, instead they rather reconcile previous results by proposing a unique mechanism, here called cathodic corrosion. Therefore, I personally suggest the authors to change the scope of their manuscript and my feedback below points toward this direction, i.e. to further benchmark the proposed mechanism of an already observed phenomenon.

(1) The recent work by Magnussen et al. (Nat. Catal. 2023, DOI: 10.1038/s41929-023-01009-z. <https://doi.org/10.1038/s41929-023-01009-z>.) is not mentioned anywhere. Although it is pretty recent so the authors may not have read it during the preparation of the manuscript, it should now be considered. Besides, I personally think the work by Buonsanti et al. also needs additional attention, rather than just a small note close in the conclusion.

Response 2.1: We really appreciate referee for the encouraging and constructive comments. Thanks also for pointing out that we missed out that Nature Catalysis paper coming out when we were preparing the paper. We read about the ChemRxiv paper at the late stage of our paper preparation and it was indeed a reassuring feeling.

When we initially used the term “cathodic corrosion” we were referring to the term to an alkali metal cation based mechanism according to the large body of work by Koper as well as the theoretical work by Alexandrov (*i.e.*, I. Evazzade, A. Zagalskaya, V. Alexandrov, Revealing Elusive Intermediates of Platinum Cathodic Corrosion through DFT Simulations. *J. Phys. Chem. Lett.* **13**, 3047–3052 (2022)). But we agree with the referee that cathodic corrosion should be a more general term, that other mechanisms could/should exist and they can also be regarded as corrosion under cathodic potentials. To avoid confusion as the referee suggested, we now have removed the word “unveiling” from the title, and changed it to “Alkali cation Induced Cathodic Corrosion in Cu Catalysts - a Missing Puzzle Piece for Understanding Their Performance in the Electrochemical Reduction of CO₂”. Hopefully this can clearly differentiate this work from other possible pathways for Cu reconstruction.

We do wish to also highlight that in literature there was little attention on linking the reconstruction of Cu under CO₂RR related reaction conditions to the “cathodic corrosion”, possibly because cathodic corrosion itself is not a very well understood phenomenon. To our best knowledge, there were only three papers in open literature about Cu catalysts that have

specifically mentioned cathodic corrosion (Table R2), and hopefully will trigger new ideas for mitigating or taking advantage of the phenomenon.

The Nature Catalysis paper by Magnussen and colleagues has now been cited in the paper. And more discussion was added comparing the current mechanism to the CO-mediated mechanism discussed in the Magnussen's paper, as well as the ChemRxiv paper by Buonsanti. More on this can be found in the Response 2.6.

Table R2, Only three papers that we know of have specifically mentioned “cathodic corrosion” – the term normally used to describe the unexpected corrosion behaviour of other noble metals.

Papers that mentioned specifically about possible cathodic corrosion in Cu	Relevant texts mentioning cathodic corrosion and cited relevant papers
G. H. Simon, C. S. Kley, B. Roldan Cuenya, Potential-Dependent Morphology of Copper Catalysts During CO₂ Electroreduction Revealed by In Situ Atomic Force Microscopy. Angew. Chem. Int. Ed Engl. 60, 2561–2568 (2021).	Reduction at $-1.1 V_{\text{RHE}}$ produces a morphology similar to that at $-1.0 V_{\text{RHE}}$ with the surface exposing Cu(100) terraces. However, the observed rectangular structures appear smaller in size, which might already be an expression of cathodic corrosion.^[33] At this negative potential, gas evolution has
J. Vavra, G. Ramona, F. Dattila, A. Kormányos, T. Priamushk, P. Albertini, A. Loiudice, S. Cherevko, N. Lopéz, R. Buonsanti, Cu+ transient species mediate Cu catalyst reconstruction during CO₂ electroreduction. ChemRxiv (2023), doi:10.26434/chemrxiv-2022-3cr9k-v2.	In previous studies, the occurrence of the catalytic reaction itself, the electrolyte, and the effect of applied potential have been all recognized to play a role in the Cu reconstruction.^{6–14} Strong binding of oxalates and carbonates has been suggested to poison the surface and influence Cu dissolution.^{44,45} Hydride-mediated cathodic corrosion was proposed for noble metals.^{46,47} Anomalous dissolution of Cu in acidic media under cathodic polarization was previously connected to the oxygen reduction reaction (ORR).^{48–50} In addition to reaction intermediates, the cathodic bias itself was found to induce surface changes of Cu in the presence of H and CO adsorbates.¹⁷
S. J. Raaijman, N. Arulmozhi, M. T. M. Koper, Morphological Stability of Copper Surfaces under Reducing Conditions. ACS Appl. Mater. Interfaces. 13, 48730–48744 (2021).	Basically said that cathodic corrosion does not happen in Cu. Although the experiment was very well designed, the imaging resolution used in this study is relatively low. We think it may have missed some of the subtle changes of the catalyst. Their CV comparison is done after 1 min of reaction which may not be long enough. In CV comparison after longer treatment, they started with polycrystalline Cu so that the signals are dominated by 111 planes. They did observe changes after long-term reactions but was attributed to contamination instead. More discussions on this in Response 2.5, where we added an experiment with oxygen-free environment as the referee suggested.

(2) Few different electrolytes have been included and, consequently, different anions have been considered. Could the authors investigate further the role of anions in driving the reconstruction? Besides H and CO species, also bicarbonate, carbonate, hydroxide, and SO₄ can bind to the surface, so these species may actually be co-promoter (or main responsible) of corrosion phenomena.

Response 2.2: We appreciate the referee's query. Our initial tests using anions other than HCO₃⁻ aimed to demonstrate that similar corrosion behaviours occur even when CO is not involved in the reaction. We observed no significant change when substituting KHCO₃ with K₂SO₄. However, when we switched to H₂SO₄, the corrosion was much less pronounced, and reintroducing K⁺ led to significant corrosion again. These findings underscore the critical role of alkali metal cations, aligning with past research on cathodic corrosion in other metals.

Per the referee's request, we conducted additional tests with potassium bicarbonate (Now **Figure S9**) and potassium hydroxide (Now **Figure S18**) at around -1.1 V_{RHE} and observed similar corrosion behaviours. Actually, we do not anticipate being able to detect a promoting effect of anions if there is one. This is firstly because our primary analysis methods (microscopy and CV) only offer qualitative or semi-quantitative insights into catalyst reconstruction at best. Given that significant reconstruction was noted in all cases, any minor promoting effects of anions might go undetected. Furthermore, cations are known to accumulate at the cathode, particularly at highly cathodic potentials. Magnussen's recent study in Nature Catalysis highlighted this through in situ Raman spectra at varying potentials, showing near-total disappearance of bicarbonate species on the surface at about -0.58V (**Figure R3**). We hypothesize that at more negative potentials, anions interact more weakly with the catalyst, a phenomenon they couldn't fully explore probably due to the intense bubbling from HER. In summary, we do not think anions played a significant role in cathodic corrosion. To thoroughly investigate any potential promoting effects, a completely new experimental design would be required, which, we hope the referee would agree, extends beyond the scope of the current paper.

Updated version of Figure S9 IL-TEM and the corresponding *ex situ* TEM results before and after 20min reaction at -1.1V_{RHE} in Ar saturated **a, b** 0.1M KHCO₃ and **c, d** 0.1M K₂SO₄. **e, f** 0.1M K₂CO₃

NEW Figure S18 The voltammetric profiles of Cu nanocubes after 25 mins (black curve) and 60 mins (red curve) of reactions at around $-0.36V_{\text{RHE}}$ in 1M KOH are presented. The purple curve depicts the profile after an initial 25-min reaction at around $-0.36V_{\text{RHE}}$, followed by a 30-seconds reaction at roughly $-1.01 V_{\text{RHE}}$.

Figure R3 In situ SERS spectra of Cu(100) recorded during stepwise decrease of the potential from 0.25 to -0.58 V and subsequent increase back to 0.25 V (* denotes molecules adsorbed on the Cu surface.) adopted from Figure 2d in Magnussen *et al. Nature Catalysis* 2023, <https://doi.org/10.1038/s41929-023-01009-z>

(3) In Fig. S11 the authors show that larger size cations lead to an increased extent of reconstruction. Besides, Fig. S10 reports that higher K^+ concentration leads to higher corrosion rates. Larger size cations like Cs^+ or K^+ determine higher reaction rates for CO_2 reduction and H_2O reduction (see Nat. Catal. 2021, 4, 654–662. <https://doi.org/10.1038/s41929-021-00655-5>; J. Am. Chem. Soc. 2022, 144, 1589–1602. <https://doi.org/10.1021/JACS.1C10171.1>.) and those reaction rates further increase for an increased concentration of cation at the surface. Higher reduction rates definitely lead to increased concentration of relevant reaction intermediates/products (OH^- , CO) at the surface. Which is the main role of alkali cations in the reconstruction process then? Do they stabilize the anionic Cu species suggested or is the increased corrosion simply due to a higher CO concentration at the surface due to cation-promoted CO_2 activation? By carrying out fixed current density studies (to fix the amount of CO/OH^- produced) or using inorganic cations (which promote CO_2 , yet do not interact with intermediates, see J. Am. Chem. Soc. 2023, 145, 16787–16795. <https://doi.org/10.1021/jacs.3c04769.1>.) the authors may be able to determine whether there is a direct promotion of corrosion by cations or simply an indirect one due to higher production of CO or OH^- .

Response 2.3: We thank the referee for these questions. When we initially did the experiments with other alkali cations, we were looking for evidence of whether other alkali cations will do the same, which is the case for cathodic corrosion on other metals (e.g., T. J. P. Hersbach, I. T. McCrum, D. Anastasiadou, R. Wever, F. Calle-Vallejo, M. T. M. Koper, Alkali Metal Cation Effects in Structuring Pt, Rh, and Au Surfaces through Cathodic Corrosion. ACS Appl. Mater. Interfaces. **10**, 39363–39379 (2018)). And that is exactly what we found. Please note that we did not claim that having larger cations will result a more significant corrosion. In the original text we said “*Similar catalyst reconstruction was observed when replacing K^+ with other alkali cations, including Li^+ , Na^+ , and Cs^+* ”. The reason was the same as we explained in Response 2.2 – we felt our methods are largely qualitative so can’t really make such a claim.

We carried out the experiment with constant current as the referee suggested and the result is shown in the updated version of **Figure S13**. It appears that more apparent reconstruction of Cu was observed after treated in electrolyte with presence of alkali cations (without CO gas) at constant current conditions. But it might still be too early to claim a definite correlation between the nature of the alkali cations to the extent of corrosion. Just like the case for anions, this question about the choice of alkali cation will need to be answered in future works with other experiment designs.

We modified the text in pg 8 and incorporated the Figure S13 as follows

Pg 8, Significant reconstruction was observed when the catalysts were treated in electrolyte with other alkali cations including Li^+ , Na^+ , and Cs^+ both at constant voltage (**Supplementary Figure S12**) and at constant current (**Supplementary Figure S13**). This observation is consistent with findings that alkali cations induced cathodic corrosions in other metals.⁵²

New Figure S13 a
 Voltammetric profiles of the Cu cubes recorded after 60min at constant current of $-8.5mA$ in $0.2M Li_2SO_4$ ($-1.27V_{RHE}$) and Cs_2SO_4 ($-1.1V_{RHE}$)
b Voltammetric profiles of the Cu cubes recorded after 60min at constant current of $-7.5mA$ in $0.05M H_2SO_4$ ($-0.92V_{RHE}$) and $0.05M H_2SO_4$ & $0.3M K_2SO_4$ ($-1.5V_{RHE}$) showing that the reconstruction of Cu depends on the alkaline cations in electrolyte, rather than the reaction rate.

0.05M H_2SO_4 & 0.3M K_2SO_4 ($-1.5V_{RHE}$) showing that the reconstruction of Cu depends on the alkaline cations in electrolyte, rather than the reaction rate.

(4) Do the authors have any suggestion on the specific form of these Cu anionic complexes formed during reconstruction? Buonsanti et al. did report Cu⁺ species during Cu dissolution instead (ChemRxiv. Prepr. 2022, DOI: 10.26434/chemrxiv-2022-3cr9k.) Could the authors motivate such difference in oxidation state between the proposed intermediates? Note that in Buonsanti et al.'s work, the overall complexes are generally anions, yet the Cu center is positively charge.

2.4 Response: We were very excited when reading about the *ChemRxiv* work by Buonsanti which we believe is the first paper to ever catch the intermediate of the what it seems to be a CO-mediated process.

In this work, we showed an alkali cation induced cathodic corrosion process without CO. The involvement of Pt anions were previously suggested by Koper and Alexandrov. Hence the hints of having anionic Cu during the in situ XAFS experiment gave yet another consistent picture. Although we couldn't identify the nature of the corrosion intermediates, it is highly likely possible that these are two different processes that involve different types of intermediates. (more on this on Response 2.6)

Our DFT and AIMD work shows that the corrosion process is possible but we can't definitely say those will be the actual intermediates involved in the reaction. Hopefully this paper will trigger more interests in the experimental hunt of the corrosion intermediates.

(5) Could the authors repeat the study for a Cu system without any pristine oxidic phase? Raaijman et al. suggested that, without initial oxidation, no reconstruction occurs (ACS Appl. Mater. Interfaces 2021, DOI: 10.1021/acsami.1c13989. <https://doi.org/10.1021/acsami.1c13989>.) and I wonder whether this holds also under these conditions.

2.5 Response: Due to the necessity of bulky equipment such as a powerful centrifuge for synthesizing and separating Cu nanocubes, it wasn't feasible to conduct the entire workflow in an oxygen-free glovebox. Instead, we placed our electrochemical cells in a glovebox with O₂ levels below 0.1 ppm. We prepared the Cu nanocubes as usual and then electrochemically reduced the surface oxides within the glovebox before proceeding with aging and CV tests. This approach mirrors the method used by Koper and colleagues (ACS Appl. Mater. Interfaces 2021), where oxide-free Cu was prepared using H₂-gas reduction in the glovebox before being anaerobically transferred to the electrochemical cell.

Figure R4 shows our CV results, consistently indicating that cathodic corrosion still occurs, suggesting that the presence of initial oxides does not significantly impact the process. The question then arises: why didn't Koper and colleagues observe this in their experiment? We propose several potential reasons:

1. They employed relatively low-resolution imaging, which might not capture subtle structural changes.
2. Their CV comparisons were made after just 1 minute of reaction, possibly too short to exhibit significant differences.
3. In CV comparison after longer treatment, they began with polycrystalline Cu, which primarily exhibits low-energy (111) plane features. These features could be easily replicated by redeposited Cu, making CV changes less noticeable, unlike our case with (100) planes.
4. They did note changes after long-term reactions but attributed these to contamination, not cathodic corrosion.

Figure R4 Voltammetric profiles of the Cu nanocubes recorded inside the glovebox (O₂<0.1ppm). The black curve is the CV baseline acquired after treated in 0.05M H₂SO₄ at 0.05V_{RHE} for 30min to etch away the surface oxides, showing evident feature related to 100 facet. After following 15 mins at -1.02V_{RHE} in 0.1M KHCO₃ (red curve), the (100) related features suppressed and, (111) defects related feature enhanced, indicating a significant reconstruction of Cu under cathodic potential even with no surface oxides. After another following 25 mins treatment, the signal from (100) further decreased and (111) increased, confirming an ongoing corrosion of Cu.

(6) "This suggested that H₂-related mechanisms are unlikely to be the primary contributors to Cu reconstruction". By reading this statement, I think the computational model employed to benchmark the experimental system is completely at odds with experimental results. If H₂-related mechanism are not the primary contributors, why should cathodic corrosion occur via Cu-H_x species as indicated in Figure 5? Magnussen et al. proposed that Cu reconstruction is mediated by CO adsorption (see Nat. Catal. 2023, DOI: 10.1038/s41929-023-01009-z. <https://doi.org/10.1038/s41929-023-01009-z>), while Buonsanti et al. investigated several different intermediates (ChemRxiv. Prepr. 2022, DOI: 10.26434/chemrxiv-2022-3cr9k). Thus, different intermediates besides H should be tested either computationally or through selected reduction studies (e.g. CO reduction at high overpotential) to further clarify the main responsible for cathodic corrosion.

2.6 Response: We thank the referee for this question and we apologize for the confusion. What we meant to say was that having HER alone without alkali metal cations will not cause similar level of reconstruction. That sentence at the bottom of page 7 is now changed to "*This indicates that in scenarios where only the HER occurs without the presence of alkali cations, the catalyst reconstruction is considerably less pronounced.*"

As requested, we have also studied CO reduction at higher overpotentials and added ex situ TEM characterization of the catalyst after being used for CORR at more negative potentials (New Figure S23). and show that the catalysts indeed undergo significant reconstruction as what was observed in CO₂RR.

Pg 16, adding text "This explanation is supported by the *ex situ* TEM results, via which we observed significant reconstruction of Cu NCs that underwent CORR at more negative potential than -0.4V_{RHE} (*i.e.*, -0.7V_{RHE}, -0.9V_{RHE}) (**Supplementary Figure S23**). The results of these preliminary experiments indicate that the corrosion behavior of Cu catalysts during the CORR closely aligns with our earlier observations in CO₂RR. More importantly, they suggest that this corrosion can be switched off by operating the CORR at a potential more positive than approximately -0.4 V_{RHE} under our experimental conditions."

"New Figure S23" Morphology of the CuNCs after 75min CORR at -0.7V_{RHE} and 40min CORR at -0.9V_{RHE}. The shape of Cu NCs were observed to degrade after the reaction, which is quite different from the stabilized morphology observed on Cu NCs after CORR at -0.37V_{RHE}.

Thanks to the excellent suggestion by the referee, we have carried out additional AIMD investigation (**Figure S26** and relevant text in the **Supplementary information**) and added the following parts into the final section of the paper to discuss the difference between the alkali cation induced cathodic corrosion process with other possible mechanisms, especially the CO-mediated process recently investigated by Magnussen and Buonsanti. Basically we

believe that multiple types of corrosion mechanisms could exist. Given the low surface coverage of CO on the catalysts (e.g., X. Chang, J. Li, H. Xiong, H. Zhang, Y. Xu, H. Xiao, Q. Lu, B. Xu, C-C Coupling Is Unlikely to Be the Rate-Determining Step in the Formation of C₂+ Products in the Copper-Catalyzed Electrochemical Reduction of CO. *Angew. Chem. Int. Ed Engl.* **61**, e202111167 (2022); J. Hou, X. Chang, J. Li, B. Xu, Q. Lu, Correlating CO Coverage and CO Electroreduction on Cu via High-Pressure in Situ Spectroscopic and Reactivity Investigations. *J. Am. Chem. Soc.* **144**, 22202–22211 (2022).) and the low-solubility of CO compared to K⁺ in water, the alkali cation induced mechanism might still dominate the cathodic corrosion process, which could explain why in our experiments, we observed consistent threshold potentials (i.e., -0.4V_{RHE}) with or without CO present in the system. Please note that the work by Magnussen did not include a controlled experiment without CO, hence their evidence couldn't exclude a CO-free pathway as well.

Added text to the conclusion part of the paper:

“The occurrence of alkali cation-induced cathodic corrosion in Cu does not negate the possibility of other mechanisms contributing to catalyst reconstruction. During the preparation and peer-review of our paper, Magnussen and colleagues³³ reported strong experimental evidence towards a CO-induced mechanism. Buonsanti and colleagues⁷⁰ argued that the CO-mediated corrosion involves a cationic Cu intermediate. To understand the role of CO in our proposed mechanism of alkali cation-induced cathodic corrosion, we conducted additional AIMD and structural optimization calculations. These examined the process of Cu extraction at the Cu-water interface with both K⁺ and CO present. **Supplementary Figure S27** reveals that CO tends to outcompete K⁺ for adsorption sites, and therefore likely inhibiting the corrosion mechanism outlined in **Figure 5**. Nevertheless, our simulations also suggest that when CO molecules are distant, K⁺ can approach the surface to initiate the corrosion process. Given the significant difference in solubilities of K⁺ and CO in water, the impeding effect of CO should be limited. Recent works by Xu and colleagues^{71, 72} indicate that CO coverage during CORR reactions is quite low and will decrease further as the electrode potential becomes more negative (e.g., approximately 0.05 monolayer at -0.6V and 0.005 monolayer at -0.9V). Moreover, our observations that the threshold potential for the alkali cation induced cathodic corrosion to take place sit consistently at around -0.4V_{RHE} with (**Figure 6**) or without CO present in the reaction suggest that variations in CO concentration or coverage do not significantly influence the process. Therefore, it is plausible that multiple corrosion pathways coexist, with the alkali cation-induced cathodic corrosion likely being the dominant process, particularly given the relatively low CO coverage on the Cu catalyst.

”

New Figure S26. K and CO configurations at the Cu electrode under cathodic conditions from first-principles structure optimization and molecular dynamics (AIMD). (a, b) Optimized adsorption structures for K with the presence of CO, one with K far away from CO (a), one with K close to CO (b). (c) One K and seven CO at the Cu-water interface, temperature ramped to 300 K in AIMD. (d) One K and two CO at the Cu-water interface, temperature ramped to 300 K in AIMD. The K-Cu distance indicating adsorption and (partial) desorption are marked in blue and orange, respectively. For clarity, all explicit water molecules are concealed in (a-d).

Referee 2 MINOR POINTS

Response 2.7 Appreciate the detailed editing from the referee

(1) the word "unreasonable" at page 12 should be deleted.

Response 2.7 (1) Done, now (at page 13) it reads like "Notably, the structure at 2.5 \AA from prior slow-growth AIMD (shown in the inset of Figure 5a on the right) becomes unstable and cannot be converged"

(2) In Figure 6, panel d, there is a typo. The potential in the graph should be -1.1 V vs RHE instead of -0.33 V vs RHE .

Response 2.7 (2) Apologies this was due to our mistake in the Figure 6 caption. This has been updated.

(3) Page 23, 4th line. In "cm⁻²" -2 should be set as superscript.

Response 2.7 (3) done

(4) Page 23, 14th line. In "And iquid products", "iquid" must be replaced with "liquid".

Response 2.7 (4) the method section has been updated

(5) In addition to reporting experimental data on a Zenodo repository, the authors should upload their DFT simulations on the ioChem-BD repository, among the suggested platforms by Nature itself (<https://www.nature.com/sdata/policies/repositories>).

Resonse 2.7 (5) We have uploaded the files, the links

The experimental data were uploaded to zenodo

<https://doi.org/10.5281/zenodo.10259206>

The DFT was uploaded to zenodo as well <https://doi.org/10.5281/zenodo.8059745>

We are still waiting for the icChem-BD repository permission to be granted, and we will make sure the link is ready for the final draft. The content will be exactly the same as the zenodo repository.

Reviewers' comments:

Reviewer #1 (Remarks to the Author):

The authors' response and modifications on the manuscript are appreciated. In this revised version, the authors claim that the reported "corrosion", i.e., the reconstruction after the oxide dissolution, is alkali cation induced in their work that features CO₂ and CO electroreduction of Cu nanocatalysts in neutral electrolyte. Again, cathodic corrosion is a wide term and does not provide mechanistic insights. Since the authors believe that the alkali-cation-induced reconstruction is the main novelty in their manuscript, more evidence is needed to confirm this effect, particularly with respect to its influence on the electric field distribution of the surfaces at different crystallographic orientation of the catalysts used in this work (spheres and cubes). Currently, unfortunately, the main issues remain.

1. Reconstruction is known and it is widely studied (it is known that it occurs even after the reduction of the oxide layer), therefore there is really no puzzle that is resolved in this study. When the authors refer to other studies for the effects of alkali ions, these are either in near-neutral solutions (like the current study) where researchers have argued on the effect of cations on enhancing the activity and selectivity for CO₂ electroreduction or are done in acidic media where the cation distribution in the double layer suppresses mobility of ions and stabilizes key intermediates. In this study, there is no clear evidence/explanation of the influence of the alkali cations either on the reconstruction of Cu catalysts (as these reconstruction mechanisms have already been reported) or on the activity and their selectivity (considering different shaped particles were used in this study). Therefore, elimination of this word from the title would be appreciated.

2. It is very true that reconstruction still continues after the oxide reduction is completed (this has been shown before). However, it is worth to consider that further restructuring will be affected by the preceding oxide reduction as it will increase the roughness of catalyst surface. In addition, the presence of the defective residual oxide on the subsurface of the catalyst can partake in the restructuring process. In other words, it is not clear why the cation would "attack" or "corrode" the surface of the metallic catalyst? What is the role of the cations on the double layer on the oxide surface vs the metallic surface in neutral media? In their calculations the authors consider the metal surface solely.

3. Importantly, in the study of Raaijman et al. which the authors refer to, similar experiments were performed to assess the presence of cathodic corrosion of Cu. One of the distinct differences in the experiments is that Raaijman et al. have used a controlled-atmosphere experimental set up to avoid oxidation of the Cu surfaces, while the authors of this manuscript report that the tested Cu NCs had about 2 nm thick surface oxide. This is a significant discrepancy as the string material and therefore its dynamic behavior is totally different. Studies in controlled conditions and model samples are recommended to understand these effects.

4. Please elaborate on the effect of other alkali cations on the onset potential of "cathodic" corrosion. Cations with more negative cathodic corrosion onset would be more promising than running the reaction at low cathodic potentials since a desirable product distribution (and Faradaic efficiency) is only obtainable at high cathodic potentials, in particular for facet selective catalyst systems such as nanocubes in neutral media. Experiments with different ions at the same pH values are required to argue on the impact of K⁺, which is the basic scientific argument of this manuscript. To prove the effect of K⁺, model samples and different electrolytes at the same pH are needed. Please include post mortem images of the catalyst after experiments with different cations (Figures S12-13).

5. Why did the authors choose to perform the first CV after 20 min of cathodic treatment? It is recommended to acquire a reference CV for samples before the cathodic potential treatment.

6. The defined potential of -0.4 V vs RHE for 0.1 M KHCO₃ electrolyte is still not clear. Is there any dependence of restructuring initiation time at different potentials above the onset potential?

7. In Figure 3b, images before and after 150 min at -0.36 V vs RHE indicate a slight degradation of particles (top-left corner). Could you elaborate to what process this degradation is related? It does not seem to be oxide reduction as the oxide shell (if present) will be obvious in TEM mode. In addition, the current corresponding to defects decreases after 150 min instead of increasing as one would expect due to degradation observed.

8. In Page 16 first paragraph, it is stated that initial 10 min at cathodic potential is more negative than -0.4 V vs RHE corresponds to the condition favorable for reduction of Cu oxide (previously

mentioned on page 7) and it will be followed by redeposition of small particles. Please revise the statement.

9. In Figure 6, although the faradaic efficiency of methane remains constant at -0.37 V vs RHE, it is evident that the selectivity towards hydrogen and liquid products changes. This indicates that there is a redistribution of active sites (i.e., restructuring). Do you have larger area images of both cubic and spherical particles after 10 h CORR tests to statistically prove absence of restructuring? Also, in graphs c and d please indicate the unit of the y-axis value or add information in the figure caption.

10. How many cyclic voltammetry (CV) cycle were performed during the test after hold at cathodic potential? Have you tried to run multiple cycles of CV without potential hold and observe if there are any changes in OH⁻ adsorption peak and defect-related peak?

11. Please indicate the necessary operation conditions for STEM observations (i.e., high voltage, probe current, convergence and collection angles etc.), to exclude e-beam related damage. It is known that catalysts are particularly sensitive when electron imaged post mortem.

12. In Figure S5, the current density profile is represented with relative wide range of values so it is difficult to assess any fluctuations. Did the authors observe any difference in current density profile stability between experiments when there was "cathodic corrosion" with respect to when it was absent?

Reviewer #2 (Remarks to the Author):

I sincerely thanks the authors for having carefully assessed all my points. I find the current version of the work more robust, self-consistent, and clear in its limitations.

As a very final validation (and extension) of the study, I would suggest the authors to investigate the extent of cation-induced cathodic corrosion in presence of bi-valent and tri-valent cations in the electrolyte, instead of alkali ones (see here for some multivalent cations already employed in the field of electrochemical CO₂ reduction, J. Am. Chem. Soc. 2022, 144, 1589–1602.

<https://doi.org/10.1021/JACS.1C10171>.) Although the authors highlight that their methodology only enables qualitative insights (i.e., it is unable to assess anion effects or difference along the alkali cations' group), they may still observe significant differences with 2+ and 3+ species, given their high acidity.

Color code: Comment from this round.

Comments from the previous round

Changes made to the manuscript this round

Changes made to the manuscript in the previous round

Reviewer 1 (Remarks to the Author):

The authors' response and modifications on the manuscript are appreciated. In this revised version, the authors claim that the reported "corrosion", *i.e.*, the reconstruction after the oxide dissolution, is alkali cation induced in their work that features CO₂ and CO electroreduction of Cu nanocatalysts in neutral electrolyte. Again, cathodic corrosion is a wide term and does not provide mechanistic insights. Since the authors believe that the alkali-cation-induced reconstruction is the main novelty in their manuscript, more evidence is needed to confirm this effect, particularly with respect to its influence on the electric field distribution of the surfaces at different crystallographic orientation of the catalysts used in this work (spheres and cubes). Currently, unfortunately, the main issues remain.

a. Reconstruction is known and it is widely studied (it is known that it occurs even after the reduction of the oxide layer), therefore there is really no puzzle that is resolved in this study. When the authors refer to other studies for the effects of alkali ions, these are either in near-neutral solutions (like the current study) where researchers have argued on the effect of cations on enhancing the activity and selectivity for CO₂ electroreduction or are done in acidic media where the cation distribution in the double layer suppresses mobility of ions and stabilizes key intermediates. In this study, there is no clear evidence/explanation of the influence of the alkali cations either on the reconstruction of Cu catalysts (as these reconstruction mechanisms have already been reported) or on the activity and their selectivity (considering different shaped particles were used in this study). Therefore, elimination of this word from the title would be appreciated.

Response 1-(a): We would like to thank the referee 1 for the engagement and the constructive comments provided throughout the review process. We have carefully considered the feedback and noted the recurrence of certain inquiries previously addressed. While we find the rationale behind the continued queries somewhat unclear, especially regarding the request for additional evidence without specific guidance on the perceived deficiencies, we are committed to clarifying these points to the best of our ability.

The comments about "cathodic corrosion is a wide term" and "reconstruction is known" were discussed in detail in our **Original responses 1.1** and **1.2** (original responses are attached at the end of this letter). In the previous round, the referee said: "*Under CO₂ reduction reaction conditions of Cu, it is known that first the oxide reduces and then the metallic copper participates in the reaction by dissolution and reprecipitation. Hence, it seems that what the authors name corrosion is the reduction of the oxide and the subsequent dissolution of Cu, a reconstruction process that is well known, so there is no "puzzle" here.*" While here the referee says: "*Reconstruction is known and it is widely studied (it is known that it occurs even after the reduction of the oxide layer), therefore there is really no puzzle that is resolved in this study.*" It appears to us that the referee originally thought what we observed was merely an oxide reduction, but now the referee is convinced that it is not the case. We agree the Cu reconstruction is indeed "widely studied", that is also why it is an important subject. But it is also a phenomenon that remains poorly understood. During the review process of this paper, two Nature Catalysis paper (*i.e.*, Amirbeigi et al., *Nature Catalysis*, 6, 837-846 (2023); Vavra, J. et al., *Nature Catalysis* 7, 89-97 (2024).) comes out on this topic. We show Cu reconstruct via a new pathway that has not been considered before. Our evidence showed that such a behaviour is consistent with the phenomenon called "cathodic corrosion" discussed in Hersbach and Koper, *Cathodic corrosion: 21st century insights into a 19th century phenomenon. Current Opinion in Electrochemistry*. **26**, 100653 (2021). Cathodic corrosion has not been reported for Cu before and can have profound effect on Cu catalysts for CO₂RR. We showed that having alkali-cations and negative-enough potentials are the two required conditions for the cathodic corrosion, which is important mechanistic insight for the process. We also showed that by the reconstruction can be avoided if one of those two conditions were mitigated (**Original response 1.3**)

Regarding the comment “... *particularly with respect to its influence on the electric field distribution of the surfaces at different crystallographic orientation of the catalysts used in this work (spheres and cubes).*” In the paper we mostly used Cu nanocubes so that we had a well-defined starting point to assess the morphology change more quantitatively. For instance in **Figure 1c**, significant reconstructions were found on round particles that can be occasionally found in the sample. We said in the paper (page 5) “*Evidently, almost all Cu particles within this view exhibit clear signs of morphological changes after the treatment, irrespective of their size and shape.*” Our XAS studies also showed that the cathodic corrosion take place on round particles as well.

Having nanocubes also enabled us to assess {100} facets related CV features to monitor the morphology change. If the field-concentration is in play, we would expect the changes to be predominantly happening on the corners of the nanocubes. However, from both TEM images and CVs we clearly saw the changes happening on the {100} facets. Hence the possible field concentration is at least not a decisive factor. We have added the following sentence to page 7 to further clarify this point.

“It is important to highlight that morphological changes occur on {100} facets and not just on the particle corners. As shown in Figure 1c, we also show significant morphological transformations occurring on spherical particles. These findings suggest that the shape of the particles and the potential field concentration are at least not the determining factors influencing the reconstruction process.”

We agree with the referee that different Cu facets may have somewhat different cathodic-corrosion behaviours. We chose to study {100} facets because they were believed to exhibit high C₂₊ selectivity in CO₂RR, hence their stability usually brings the most interests. As we explained in the **Original Response 2.2**, our methods only offer qualitative or semi-quantitative results. The most important finding is that we confirmed that cathodic corrosion is in play. But it will be difficult for us to comment which facet corrodes faster than the other. Such a study will require entirely new sets of samples (e.g., {111} facets Cu particles) and will also require more quantitative methods. We hope the referee would agree these are beyond the current scope and should be a subject for follow up studies.

Furthermore, we are very much puzzled when reading the comment: “*In this study, there is no clear evidence/explanation of the influence of the alkali cations either on the reconstruction of Cu catalysts (as these reconstruction mechanisms have already been reported) or on the activity and their selectivity (considering different shaped particles were used in this study).*” We feel that this stands in contrast to the comprehensive evidence and detailed explanations provided in our manuscript and the previous response letter. This question also appealed in the previous review and we have answered in the **Original Response 1.6**.

We respectfully disagree with the referee's remark “*as these reconstruction mechanisms have already been reported*”. The absence of specific references to prior studies by the referee leaves us uncertain about the exact literature being compared to our findings. It is important to note that if the mechanisms of Cu reconstruction were as well-understood as suggested, it would be unlikely for two recent **Nature Catalysis** publications to emerge on this topic, indicating a continuing evolution in our understanding of these processes. The cathodic corrosion pathway described in this work is notably absent from the discussions in the literature including the above-mentioned Nature Catalysis articles. This underscores the originality and significance of our findings, contributing a unique perspective to the ongoing scholarly dialogue surrounding Cu reconstruction mechanisms.

It is unclear to us which term the referee suggests removing from our title, as their comment was: “*Therefore, elimination of this word from the title would be appreciated.*” We infer the referee might be referencing the term “puzzle,” following his/her remark, “*there is no puzzle.*” Our title, after the latest revision, reads “Alkali Cation Induced Cathodic Corrosion in Cu Catalysts - A Missing Puzzle Piece for Understanding Their Performance in Electrochemical CO₂ Reduction.” Our intention was not to imply that the phenomenon of Cu reconstruction has never been observed. Rather, our aim was to

Point by point response R2 _ 03 Feb 2024

highlight that the aspect of cathodic corrosion has been overlooked in the context of understanding and enhancing Cu electrocatalysts. Thus, we employed a metaphor, suggesting that cathodic corrosion represents a missing puzzle piece in a larger jigsaw puzzle of comprehending copper electrocatalysts. We understand this metaphor has caused unnecessary confusions and we apologize for it. Hence we have now changed the title to a simpler version: **Alkali Cation Induced Cathodic Corrosion in Cu Electrocatalysts**. Hope this is better and clearer.

Reviewer 1 b. It is very true that reconstruction still continues after the oxide reduction is completed (this has been shown before). However, it is worth to consider that further restructuring will be affected by the preceding oxide reduction as it will increase the roughness of catalyst surface. In addition, the presence of the defective residual oxide on the subsurface of the catalyst can partake in the restructuring process.

In other words, it is not clear why the cation would “attack” or “corrode” the surface of the metallic catalyst? What is the role of the cations on the double layer on the oxide surface vs the metallic surface in neutral media? In their calculations the authors consider the metal surface solely.

Response 1-(b) We appreciate that the referee recognize that “It is very true that reconstruction still continues after the oxide reduction is completed (this has been shown before)”. This is indeed what we observed. And because of the absence of oxides, we know that there exist additional reconstruction mechanisms. That is also why the two recent Nature Catalysis papers (*i.e.*, Amirbeigi et al., *Nature Catalysis*, 6, 837-846 (2023); Vavra, J. et al., *Nature Catalysis* 7, 89-97 (2024)) got published because they show the reconstruction of Cu beyond an oxide-reduction process.

Here we reported an alkali-cation induced cathodic corrosion, which is a reconstruction pathway that has not been considered before for Cu. This could be an important missing information when we try to understand Cu electrocatalysts and improve the CO₂RR process. We also had to do a lot of control experiments to be sure that this is not an oxide-reduction process. Our experimental evidence did not indicate any residual oxides exist in the catalyst during our tests. For instance, our *in situ* XAFS results show that the Cu catalysts are fully reduced. In addition, in the last round of review, we also added new experimental results done in a glovebox to minimize air-exposure of Cu catalysts during the sample handling process and we made nearly identical observations (**Original Response 2.5**). Regarding the question: “What is the role of the cations on the double layer on the oxide surface vs the metallic surface in neutral media?” we do not consider oxide surface will bring new insights as the oxide reduction is completed, plus copper oxides are thermodynamically unstable under cathodic conditions.

We are aware that in the community there has been debates over the presence of so-called sub-surface oxygen (or residual oxides) under CO₂RR relevant potential. However, our results agree with what we think are the prevailing view that the sub-surface oxides are absent during the CO₂RR reaction." Many recent *in situ*/operando studies supported this view that the Cu is indeed fully reduced during the initial CO₂RR. Examples include ***in situ* Raman studies** (*e.g.*, Magalí Lingenfelder et al., *Nano Lett*, 2021, 21, 2059–2065; Haotian Wang et al., *PNAS*, 2021, 118; Hongyan Liang et al., *J. Am. Chem. Soc.*, 2022, 144, 14005–14011), ***in situ* (GI)XAS studies** (*e.g.*, Thomas F. Jaramillo et al., *J. Am. Chem. Soc.*, 2021, 143, 588–592; Yang Peidong et al., *Nature*, 2023, 614, 267), and **quasi *in situ* XPS studies** (*e.g.*, Beatriz Roldan Cuenya et al., *Nat Energy*, 2020, 5, 317–325).

Although there are still some report claiming residual oxides in Cu (*e.g.*, Yongji Gong et al., *Nat Com*, 2022, 13, 1877; Anders Nilsson et al., *J. Phys. Chem. Lett*, 2017, 8, 285–290; Beatriz Roldan Cuenya et al, *Nat Comms*, 2016, 7, 12123), we wish to highlight that the Cu catalysts in those works usually undergo some special reaction conditions such as the presence of Cl⁻ ions in the electrolyte, or polarizing the Cu (re-oxidizing the Cu), none of which applies to the present work.

To conclude this part, we do not think residual oxides are in play in the observed cathodic corrosion of Cu catalysts.

Regarding the comment: “In other words, it is not clear why the cation would “attack” or “corrode” the surface of the metallic catalyst?” Yes, cathodic corrosion of metal is indeed a somewhat counter-intuitive phenomenon, but it is real. People nowadays even use the principle of cathodic corrosion to make nanomaterials, which are nicely summarized in the review “A. I. Yanson, et al., *Cathodic corrosion: a quick, clean, and versatile method for the synthesis of metallic nanoparticles*. *Angew. Chem. Int. Ed Engl.* 50, 6346–6350 (2011).” However, cathodic corrosion of metal is still not well-understood, despite it has been discovered for more than a century. To the best of our knowledge, there

is only one paper that delves into the kinetics of cathodic corrosion of platinum at the molecular level through Density Functional Theory (DFT) simulations, marks the forefront of research in this area (Alexandrov, V. et al., "Revealing Elusive Intermediates of Platinum Cathodic Corrosion through DFT Simulations," *J. Phys. Chem. Lett.*, 2022, 13, 3047-3052).

Meanwhile, regarding the role of cations, researchers have extensively investigated the modulation of selectivity and activity in the CO₂RR through alkali cations within the double layer. For instance, Xile Hu *et al.* reported that hydrated alkali cations adsorbed on the cathode modify the distribution of electric field in the double layer, impeding hydrogen evolution by suppression of migration of hydronium ions while at the same time promoting CO₂RR (Xile Hu *et al.*, *Nature Catalysis*, 2022, 5, 268–276). Ab initio molecular dynamics simulations reveal that alkali cations on Cu surface can repel protons from the interface and decrease CO₂ activation barrier (Jia Li *et al.*, *Nature Communications*, 2024, 15, 612).

Our modelling is based on all above, that we have a metallic copper surface, with alkali cations present in close proximity. We employed the same methods as in the work of Alexandrov *et al.*, and we observed a similar trend, where a metastable corrosion intermediates can exist. Note that in the work by Vavra, J. *et al.*, *Nature Catalysis* 7, 89-97 (2024), a transient corrosion intermediates were captured by online mass spec, although only with the help of an organic complex.

In summary, our experiments show that an alkali-cation induced cathodic corrosion is happening on metallic Cu electrocatalysts. Our modeling demonstrates that this seemingly counter-intuitive process aligns with cutting-edge mechanistic understanding within the broader scope of research on cathodic corrosion on metals.

Reviewer 1 c. Importantly, in the study of Raaijman et al. which the authors refer to, similar experiments were performed to assess the presence of cathodic corrosion of Cu. One of the distinct differences in the experiments is that Raaijman et al. have used a controlled-atmosphere experimental set up to avoid oxidation of the Cu surfaces, while the authors of this manuscript report that the tested Cu NCs had about 2 nm thick surface oxide. This is a significant discrepancy as the string material and therefore its dynamic behavior is totally different. Studies in controlled conditions and model samples are recommended to understand these effects.

Response 1-(c) This question has been asked by Referee 2 in the previous round. We repeated our experiments in a glovebox with inert atmosphere and residual oxygen level less than 1ppm. These experiments lead to identical conclusions and we have answered in detail in the **Original Response 2.5**. The additional data and discussion seemed to satisfy Referee 2's concerns.

Reviewer 1 d. Please elaborate on the effect of other alkali cations on the onset potential of “cathodic” corrosion. Cations with more negative cathodic corrosion onset would be more promising than running the reaction at low cathodic potentials since a desirable product distribution (and Faradaic efficiency) is only obtainable at high cathodic potentials, in particular for facet selective catalyst systems such as nanocubes in neutral media. Experiments with different ions at the same pH values are required to argue on the impact of K^+ , which is the basic scientific argument of this manuscript. To prove the effect of K^+ , model samples and different electrolytes at the same pH are needed. Please include post mortem images of the catalyst after experiments with different cations (Figures S12-13).

Response 1-(d) The experiments requested by the referee were available in **Figures S12, S13** in the last round revision, where we studied the effect of different alkali cations (Li^+ , Na^+ , K^+ , Cs^+) in both neutral and acidic environments. We showed that other alkali cations indeed will lead to similar corrosion behaviours. In the **Original Response 2.2**, we discussed these in detail. We also explained that our methods are qualitative or at-best semi-quantitative, and they fall short of enabling a precise comparison of the corrosion rates quantitatively or precisely pin-down the onset potential for every experimental conditions. In the paper, we focused on K^+ as it is the alkali cation used most in CO_2RR and usually gave the best CO_2RR performance. **The main point is that cathodic corrosion happens on Cu and it has an onset potential, which is robustly supported by the evidence we’ve presented.**

The referee is correct that working at high cathodic potentials is beneficial for the reaction rates. Our demonstration of CO_2R at more positive potentials was not meant to suggest a superior application but rather to illustrate that cathodic corrosion can be effectively mitigated under these conditions. This supports our hypothesis regarding the process and sheds light on why long-term shape-induced selectivity in Cu catalysts under CO_2RR might not be sustainable at highly cathodic conditions. Acknowledging and understanding this phenomenon is crucial for developing strategies to either mitigate or leverage such effects for improved catalyst design and performance.

As per requested, the additional TEM images of the Cu nanocubes after treated in 0.2M $Li_2SO_4/Na_2SO_4/Cs_2SO_4$ electrolyte for 60min were shown in **Figure R2-1**. The Cu nanocubes undergo significant reconstruction in all electrolytes, which is consistent with the CV results in **Fig S12**.

Figure R2-1 Morphology of the CuNCs after 60min reaction at around $-1.1V_{RHE}$ in Ar saturated 0.2M $Li_2SO_4/Na_2SO_4/Cs_2SO_4$ (from left to right) imaged with ex situ TEM.

Reviewer 1 e. Why did the authors choose to perform the first CV after 20 min of cathodic treatment? It is recommended to acquire a reference CV for samples before the cathodic potential treatment.

Response 1-(e) The reason to perform the first cyclic voltammetry (CV) after a 20-minute cathodic treatment was driven by practical considerations related to the handling of Cu nanoparticles. Given the challenge of preventing air contact during sample preparation, the as-synthesized Cu samples typically possess a thin oxide layer on their surface as we shown in the paper. This oxide layer does not significantly interact with OH⁻ adsorption during CV, and therefore do not permit useful morphological information to be extracted from the CV technique. To ensure a meaningful baseline for comparison, we initially subject the catalyst to a 20-minute cathodic treatment. This procedure is aimed at reducing the native oxide layer, thereby ensuring that subsequent changes observed in CV are attributable to the catalyst's behavior rather than surface oxides. We explained this in the original text, page 7: *“Figures 2e show two voltammograms, recorded in 0.1M NaOH after the catalysts were treated at about -1.05V_{RHE} in Ar-saturated 0.1M KHCO₃ for 20 mins and 65 mins, respectively. This two-stage comparison helps to exclude the influence of surface oxide crusts, which should be reduced quickly (e.g., within 10 mins for 40 nm thick Cu₂O¹¹) under such a highly cathodic condition. It is therefore reasonable to assume the surface oxides are completely reduced within 20 mins under our experimental conditions.”* In the in situ XAS experiment, a quick reduction of surface oxides were observed that is consistent with our experiment design.

To further address concerns raised during the first round of revisions, we conducted additional experiments within a glovebox to acquire a reference CV from an oxide-free Cu sample. This involved removing the surface oxide using sulfuric acid and performing the CV under an air-free environment, as described in our **Original Response 2.5**. The findings from these glovebox experiments, which observed reconstruction following cathodic treatment of the oxide-free Cu, align with the results and interpretations derived from the two-stage CV approach depicted in **Figure 2e**. This consistency across different experimental conditions reinforces the validity of our methodology and the conclusions drawn from our study.

Fig 2e Voltammetric profiles of the Cu nanocubes recorded after 20 mins (black curve) and 65 mins (red curve) of reactions at about -1.05V_{RHE} in 0.1M KHCO₃.

Reviewer 1 f. The defined potential of -0.4 V vs RHE for 0.1 M KHCO₃ electrolyte is still not clear. Is there any dependence of restructuring initiation time at different potentials above the onset potential?

Response 1-(f) We thank referee for this comment. Onset potentials were found in cathodic corrosion in other noble metal, as we mentioned in page 10 “Another important characteristic of cathodic corrosion is that it occurs only when the electrode potential is more negative than a certain onset potential⁵⁴”. That is why we searched for it in the case of Cu catalysts. We found that in our experimental conditions, such a threshold does exist around -0.4 V_{RHE}. We found that similarly prepared Cu electrodes will have a very different trend of current vs time above or below about -0.4 V_{RHE} (**Figure S22**). Later CV (**Figure 3**), IL-TEM (**Figure S14**), in situ XAFS (**Figure 4**) and AIMD results (**Figure 5**) also gave consistent conclusions. More importantly, we showed that long-term (10hour) CORR at -0.33V_{RHE} does not show signs of cathodic corrosion compared to the case for more negative potentials (e.g., -1 V_{RHE}), and the Cu nanocubes can maintain a selectivity advantage over Cu spherical particles, unlike the case of CO₂RR at -1.1V_{RHE}

Regarding the "initiation time" of restructuring at different potentials, our experimental methods lacked the temporal resolution necessary to precisely identify such a moment, if there is one. We cannot exclude the possibility that it is because the corrosion has very long “initiation time”. However, distinguishing between a prolonged initiation time and the absence of corrosion initiation is challenging and, from a practical perspective, may not significantly impact the overall understanding of the process. For instance, no cathodic corrosion was observed after 10 hours at -0.33 V_{RHE}, whereas at -1 V_{RHE} corrosion occurred within a few minutes (**Figure 3**). This observation aligns with the principle that catalysis is primarily concerned with kinetics rather than thermodynamics; significantly slowing down a process is practically equivalent to stopping it. In literature researchers often do not make such a distinction. For example, in the 2009 science paper (J. K. Edwards, B. Solsona, E. N. N, A. F. Carley, A. A. Herzing, C. J. Kiely, G. J. Hutchings, *Switching off hydrogen peroxide hydrogenation in the direct synthesis process. Science. 323*, 1037–1041 (2009)), the authors in the title said the hydrogenation of H₂O₂ to H₂O is “switched off”. But obviously H₂O₂ hydrogenation to water is a highly thermodynamically favourable process, it just became kinetically sluggish with the right choice of catalysts, so it is effectively “switched off”. In our case, we found an on-set potential that at least effectively switched off the cathodic corrosion. To rigorously study the rate of corrosion and its dependence on reaction conditions will be an interesting topic for future studies. But it is beyond the scope and it doesn’t affect the merit of this work.

Reviewer 1 g. In Figure 3b, images before and after 150 min at -0.36 V vs RHE indicate a slight degradation of particles (top-left corner). Could you elaborate to what process this degradation is related? It does not seem to be oxide reduction as the oxide shell (if present) will be obvious in TEM mode. In addition, the current corresponding to defects decreases after 150 min instead of increasing as one would expect due to degradation observed.

Response 1-(g) As we explained in **Response 1-(e)** and also shown in the paper, the as synthesized Cu nanocubes always have a thin layer of oxides. So there is always going to be some changes due to the reduction/dissolution of oxides. The important thing is that the amount of deformation due to oxide reduction/dissolution was much less (**Figure 3b**) compared to the case of cathodic corrosion (**Figures 1 and 2**).

The TEM bright field images shown in Figure 3b is mainly dominated by mass-thickness and diffraction contrast and the magnification is relatively low. Since the oxide layer is very thin (*e.g.*, ~ 2nm), it does not necessarily give a clear contrast. Higher resolution STEM-HAADF images and STEM-EELS maps (**Figures S2-S4**) are more suitable to visualize these thin oxide layers.

Regarding the referee's point about the anticipated increase in defects-related features within the CV due to observed TEM degradation, it's crucial to consider that the CV results depicted in **Figure 3a** reflect samples post 30-minute treatment at -0.36V or higher, where oxide layers are reduced. The difference in surface defects between samples treated for 30 minutes and those for 150 minutes may be random, given that the initial surface-oxidation of the Cu nanocubes and the following electrochemical reduction process are not within our control. However, the key observation is the dramatic contrast in structural changes and surface defects induced within just 3 minutes of treatment at $-1V_{RHE}$, compared to an additional 150 minutes at -0.36V. This difference in treatment conditions markedly affects not only areas associated with surface defects but also those corresponding to {100} facets, with the total charge transfer being nearly identical in both scenarios. This serves as compelling evidence that upon activation of cathodic corrosion, the catalyst undergoes significant structural alterations.

To better clarify this point, we added the following text in page 10 **“Note that the images of the Cu nanocubes before the reaction shown in Figure 3b will have a thin-layer of surface oxides, making them differ slightly than those generating the signal of reference CV (30 mins) in Figure 3a, since the 30 mins of treatment under $-0.36V_{RHE}$ or more will remove the surface oxides.”**

Reviewer 1 h. In Page 16 first paragraph, it is stated that initial 10 min at cathodic potential is more negative than -0.4 V vs RHE corresponds to the condition favorable for reduction of Cu oxide (previously mentioned on page 7) and it will be followed by redeposition of small particles. Please revise the statement.

Response 1-(h) We appreciate the referee's feedback and apologize for the confusion caused by our initial explanation. In response, we have revised the text for better clarity. It's important to note that our experiments depicted in the original **Figure S22**, the same electrode containing Cu nanocubes was tested in a flow cell at different potentials with 1M KOH as the electrolyte. The current was recorded when we subsequently ramping to more negative potentials. Specifically, the electrode underwent sequential 10-minute treatments at $-0.27\text{ V}_{\text{RHE}}$, $-0.36\text{ V}_{\text{RHE}}$, $-0.42\text{ V}_{\text{RHE}}$, and followed by $-0.46\text{ V}_{\text{RHE}}$, respectively. Interestingly, we firstly observed that the current decreases or stays roughly the same with time when the electrode potentials were more positive than about $-0.4\text{ V}_{\text{RHE}}$. A sharp contrast was observed when the electrode potential becomes more negative than about $-0.4\text{ V}_{\text{RHE}}$, where an obvious increase of current with time was observed in each case. The current increase suggest the roughening of Cu surfaces and the formation of Cu nanoparticles. Because the catalyst has already been treated with negative potentials close to $-0.4\text{ V}_{\text{RHE}}$ for more than 20 mins, there should be no oxides left in the catalysts, hence the surface roughening and the formation of nanoparticles indicate the cathodic corrosion process. To further exclude the possible interference of residual oxides, additional experiments were carried out over prolonged duration (**New Figure S22 e, f**). The observed decrease in current over a 10-hour reaction at $-0.37\text{ V}_{\text{RHE}}$ and the increase in current over an 80-minute reaction at $-0.58\text{ V}_{\text{RHE}}$ are consistent with the previous findings. These strongly suggest that the cathodic corrosion is happening and an onset potential is around $-0.4\text{ V}_{\text{RHE}}$ in our experiment conditions.

We have revised the relevant text on page 16 as following:

*“CORR experiments were carried out using an electrolyser flow cell and a gas diffusion electrode (GDE) (see **Methods**), with 1M KOH being used as the electrolyte. **Supplementary Figure S22 (a-d)** shows the cell current as a function of time on a Cu nanocube-containing electrode at various CORR potentials. The electrode was treated sequentially for 10 minutes each at $-0.27\text{ V}_{\text{RHE}}$, $-0.36\text{ V}_{\text{RHE}}$, $-0.42\text{ V}_{\text{RHE}}$, and then $-0.46\text{ V}_{\text{RHE}}$. We initially observed stable or decreasing currents at potentials more positive than $-0.4\text{ V}_{\text{RHE}}$, but then noted a significant current increase at more negative potentials, indicating Cu surface roughening and Cu nanoparticle formation. Since the catalyst has already been treated with negative potentials close to $-0.4\text{ V}_{\text{RHE}}$ for 20 mins, there should be no oxides left in the catalysts. Further experiments, including a 10-hour test at $-0.37\text{ V}_{\text{RHE}}$ and a separate 80-minute test at $-0.58\text{ V}_{\text{RHE}}$ (**Supplementary Figure S22 (e, f)**), show consistent results. These findings strongly suggest that the cathodic corrosion is in play when the electrode potential is more negative than an onset potential around $-0.4\text{ V}_{\text{RHE}}$ under our experimental conditions.”*

Updated Figure S22 Online current of CORR using Cu nanocubes at different cathodic potentials. The CORR was carried out using the same electrode loaded with Cu nanocubes, with **(a)** 10 mins at $-0.27V_{RHE}$, and then **(b)** 10 mins at $-0.36V_{RHE}$, followed by **(c)** 10 mins at $-0.42V_{RHE}$ and finally **(d)** 10 mins at $-0.46V_{RHE}$. Current of CORR using two other electrodes of Cu nanocubes at $-0.37V$ over 10h **(e)** and at $-0.58V$ over 80min **(f)**, showing consistent trend.

Reviewer 1 i. In Figure 6, although the faradaic efficiency of methane remains constant at -0.37 V vs RHE, it is evident that the selectivity towards hydrogen and liquid products changes. This indicates that there is a redistribution of active sites (i.e., restructuring). Do you have larger area images of both cubic and spherical particles after 10 h CORR tests to statistically prove absence of restructuring? Also, in graphs c and d please indicate the unit of the y-axis value or add information in the figure caption.

Response 1-(i) We appreciate the referee's inquiry. As per additional TEM images of the Cu nanocubes and Cu spheres after 10 hour of CORR tests at -0.37 V_{RHE} are shown in **Figure R2-3 a and b**, respectively. The results are consistent with the ones shown in **Figure 6**. The shape of the Cu nanocubes are largely retained. Some minor structural changes are likely due to the reduction of surface oxides, which are much less obvious compared to the case if the electrode potential is more negative than -0.4 V_{RHE} (**Figure R2-3 c**). These observations are consistent with the rest of the paper.

Figure R2-3 TEM bright field images of the (a) Cu nanocubes and (b) Cu spheres after 10h CORR at -0.37 V_{RHE}. TEM bright field images of the Cu nanocubes after 1h CORR at -0.7 V_{RHE}.

Having a stable catalyst morphology is only the first step to achieve a stable catalytic performance, and many reactor-related factors can have significant impact on the final results. In particular, long-term CORR stability measured at a gas diffusion layer electrode (GDE) in the flow cell and at relatively positive potential and low current levels can be challenging, as factors such as product cross-over

The observed increase in the hydrogen evolution reaction (HER) for both cubic and spherical catalysts (**Figure 6b**) indicate a gradual flooding of the GDE, which impede the transport of CO supply to the electrode-electrolyte interface, thereby promoting the reduction of hydrogen over the course of the reaction time. We have explained this in the previous round of revision on pg 16: *“An observed increase in hydrogen evolution reaction (HER) for both catalysts is attributed to gradual electrode flooding rather than to changes in catalyst morphology, as confirmed by TEM images before and after the reaction (insets in Figures 6c & 6d).”*

We also want to highlight that in CORR around -0.4 V_{RHE}, having Cu{100} facets favours producing propanol over ethanol and acetic acid among liquid products, as reported by Kang et al., *Nature Catalysis*, 2019, 2, 423–430. This is exactly what we observed that the CORR using Cu nanocubes produces significantly more propanol compared to the counterpart using Cu spheres, and the selectivity advantage is stable during across a 10-hour CORR test (**Figure 6c**). This is in sharp contrast compared to the case of CO₂RR operated at -1.1 V_{RHE} where we showed that the Cu nanocube morphology cannot be maintained under such conditions, because of the cathodic corrosion.

During the first round of review, we repeated the long-term CO₂RR and CORR experiments multiple times and the results can be fully reproduced. We also noticed that the total amount of liquid products measured in CORR decrease overtime consistently (**Supplementary information Figures S24**), leading to the decreasing FE towards all liquid products. This was highly likely due to the fact we only collected liquid product from the cathode side, and some liquid products got crossover through the ion exchange membrane and get oxidized. This is also a known effect and can usually be neglected when

doing high-current experiments. Nonetheless, using TEM and CORR experiments, we show that if we can switch off cathodic corrosion (in this case by using more positive potentials), we can preserve the catalyst morphology and harness the shape-induced selectivity advantage in Cu nanocubes.

In the previous round of revision in the paper (pg 17): “A progressive decrease in total FE was noted (Supplementary information Figures S24), likely due to some liquid products crossing over through the ion exchange membrane.⁶⁹ The low current and extended duration of the CORR experiments make them prone to such systematic errors. However, the total FE remained acceptably around or above 90%. Importantly, when focusing on the fractions of different liquid products, which were accurately determined via NMR, a consistent distribution was observed for both catalysts. Notably, Cu NCs produced a significantly higher fraction of propanol (about 60%) among all liquid products during the 10-hour test (Figure 6c), a result not replicated when spherical Cu nanoparticles were used (Figure 6d). Thus, Cu NCs maintained a morphology-induced selectivity advantage in the CORR experiment at $-0.37 V_{RHE}$, in contrast to the earlier CO₂RR results at $-1.1 V_{RHE}$.”

The unit of the y-axis value has been added in the updated Figure 6.

Updated Figure 6. The comparison of Cu nanocubes and 25 nm spherical Cu nanoparticles when used for 10-hour tests of (a) CO₂RR at $-1.1 V_{RHE}$ and (b) CORR test at $-0.37 V_{RHE}$. Liquid product distributions of the CORR experiments as a function of time for (c) Cu nanocubes and (d) 25 nm spherical Cu nanoparticles, tested at $-0.37 V_{RHE}$ in 1 M KOH. The error bars show standard error of three independent measurements. Insets show *ex situ* TEM bright field images of the corresponding sample in each case before and after the CORR experiments.

Reviewer 1 j. How many cyclic voltammetry (CV) cycles were performed during the test after hold at cathodic potential? Have you tried to run multiple cycles of CV without potential hold and observe if there are any changes in OH- adsorption peak and defect-related peak?

Response 1-(j) We thank the referee for this question. It was reported that consecutive cycles of CV can introduce some changes to the catalysts (T. M. Koper *et al.*, *ACS Appl. Mater. Interfaces* 2021, 13, 48730–48744). Hence in our experiments, we normally only take the cyclic voltammetry (CV) data from the first cycle, after the sample being treated at a cathodic potential for a certain period of time.

Multiple cycles were performed as requested by the referee. In the case of CV acquired after treatment in 0.1 M KHCO_3 for 20 mins (**Figure R2-4a**), it was found that consecutive CVs did introduce small changes in the Cu nanocube relevant and an overall loss of Cu surface area. This observation is consistent with literature and can be attributed to the partially irreversible reconstruction of Cu (100) caused by OH- adsorption (T. M. Koper *et al.*, *ACS Appl. Mater. Interfaces* 2021, 13, 48730–48744). In the paper, we focused on comparing the 1st CV cycle of materials being treated at different conditions, which allowed us to observe the cathodic corrosion. Note that the extend of feature changes of consecutive CVs (**Figure R2-4a**) were much less pronounced than that induced by cathodic corrosion.

A potential hold or a cathodic treatment is necessary, because the thin oxide layer that inevitably present in our sample would impede the OH- adsorption signal, and thus, no meaningful information regarding Cu morphology could be acquired from CV (**Figure R4b**). We repeated the experiments inside the glovebox after we use 0V_{RHE} in sulfuric acid to remove the surface oxides. The multiple cycles of CV results are shown in **Figure R2-4c**. A similar partially irreversible reconstruction of Cu (100) with successive cycling was also observed in this condition as well.

In summary, we are aware of possible changes due to multiple cycles of CV, hence we only used the 1st cycle. And the changes due to cathodic corrosion were much more pronounced than the possible changes due to multiple CV cycles. Hence this does not affect the integrity of the current conclusions.

Figure R2-4 (a) Voltametric cycling recorded in 0.1M NaOH of the Cu nanocubes treated in 0.1M KHCO_3 for 20min. (b) Voltametric profiles of the Cu nanocubes with no cathodic treatment. (c) Voltametric cycling of the oxide free Cu nanocubes inside the glovebox.

Reviewer 1 k. Please indicate the necessary operation conditions for STEM observations (i.e., high voltage, probe current, convergence and collection angles etc.), to exclude e-beam related damage. It is known that catalysts are particularly sensitive when electron imaged post mortem.

Response 1-(k) We thanks the referee for the comment. The STEM characterization is done using an aberration-corrected JEOL ARM200CF microscope equipped with a cold field emission source, being operated at 200kV. The convergence angle is about 30 mrad. For STEM imaging, a typical probe current will be roughly 10 pA. For STEM-EELS analysis, larger probe current is needed, around 100 pA. Please note that the microscope do not have the ability to simultaneously measure probe current while taking the data. The probe current was estimated from the data previously measured using a Faraday cup.

The referee is correct and electron-beam damage is a general problem when studying nanomaterials (or probably any materials) using electron microscopy techniques. In our case of Cu nanocubes, we found the e-beam effect is negligible when we used STEM imaging and EELS to evaluate the surface oxide layer on Cu. As shown in **Figure R2-5**, the surface oxide layer appears the same in two consecutive STEM-EELS acquisition, and the results obtained from EELS are consistent from that obtained from STEM-HAADF and BF images, which require much less dose. Hence the e-beam effect is negligible. These STEM results enabled us to realize the Cu catalysts continue to reconstruct long after the surface oxides were reduced, leading to the identification of cathodic corrosion on Cu.

Figure R2-5 (a) and (b) are STEM high angle annular dark field (HAADF) images and (c) is a STEM bright-field (BF) image. The oxide shell can be visible from these STEM images. (d) is a STEM-ADF image with rotated scan, getting ready for the STEM-EELS mapping. The first attempt of STEM-EELS mapping was stopped mid-way due to sample drift. The Oxygen, Cu^0 and Cu^{x+} maps and their RGB composite image of part of the particle that had been scanned are shown (d-g). After resting the sample for a few mins, a second STEM-EELS map was taken, and the results were shown (h-k). There is no obvious difference between the oxide thickness between the two attempts. From the results of the 2nd STEM-EELS mapping attempt, there is no obvious difference between the part of the particles that have been previously exposed to long exposures during the 1st attempt and the rest part of the particle. There is also no obvious difference in the oxide layer thickness measured using STEM-EELS (higher dose) and STEM HAADF and BF images (much lower dose).

Reviewer 1 L. In Figure S5, the current density profile is represented with relative wide range of values so it is difficult to assess any fluctuations. Did the authors observe any difference in current density profile stability between experiments when there was “cathodic corrosion” with respect to when it was absent?

Response 1-(L) We thank the referee for this comment. The current density profile has now been represented in a smaller range to facilitate visual assessment and error bars from three independent measurements have been added. (**updated Figure S5**). In previous stability tests, we found that the measured current during CO₂RR can be affected not only by the catalysts but also by the amount of electrolyte. One usually needs to add just right amount to barely submerge the catalyst in order to get the optimized performance. However this makes it very sensitive to the bubble accumulation and electrolyte evaporation during the reaction, which are very practical consideration regarding the cell configuration. As shown in the **updated Figure S5**, the current trend exhibits a fluctuated step-wise feature due to the repeatable accumulation and release of gas bubbles around the membrane. The evaporation of electrolyte would result in less electrode area within the electrolyte and contribute to a declining current over long-term reaction.

We went on repeating the experiment of 10-hour CO₂RR and this time using more electrolyte and a wet CO₂ flow (**Figure R2-6**). Immersing more electrolyte to an additional area of blank glassy carbon made the current less sensitive to the electrolyte height. In this case we observed a clear increase in current at least during the initial 40-minute reaction, followed by a consistently declining current over 10 hours of CORR were observed (**Figure R2-6**). This is in sharp contrast compared to the case of CORR operated at -0.37 V_{RHE} in a flow-cell. This increase in current over such an extended period of time is consistent with the picture of dissolution and redeposition of metal. Eventually catalyst deactivate possibly because of particle detachment or agglomeration, which are common degradation pathways for nanoparticle catalysts. But the total current drop was much less in this case (25% over 10hours) where we used more electrolyte and wet CO₂, compared to the original experiment (43% over 10 hours) shown in the **updated Figure S5**. Note that in our experiments, since Cu nanocubes were used, cathodic corrosion always lead to surface roughening and the formation of smaller nanoparticles in the beginning, hence can be observed as an increase in current densities in the early stage of the reactions. This is also consistent with our observations in CORR, discussed above in **Response 1-(h)**.

Updated Figure S5 The current density and gas phase products of the Cu nanocubes at $-1.1 V_{\text{RHE}}$ over 10h CO_2RR . Shaded areas of the current profile show standard deviations from three independent measurements. The error bars of the faradic efficiency represent standard error of mean from three independent measurements. Inset: Representative TEM bright field images of Cu nanocubes at different stages of the CO_2RR time. Scale bars represent 20nm.

Figure R2-6 The current density at $-1.1 V_{\text{RHE}}$ over 10h CO_2RR using more electrolyte and wet CO_2 flow (top) and current density at $-0.37 V_{\text{RHE}}$ over 10h CORR (bottom) replotted from Fig 6c, showing different trend over reaction time.

Reviewer 2 (Remarks to the Author):

I sincerely thanks the authors for having carefully assessed all my points. I find the current version of the work more robust, self-consistent, and clear in its limitations. As a very final validation (and extension) of the study, I would suggest the authors to investigate the extent of cation-induced cathodic corrosion in presence of bi-valent and tri-valent cations in the electrolyte, instead of alkali ones (see here for some multivalent cations already employed in the field of electrochemical CO₂ reduction, J. Am. Chem. Soc. 2022, 144, 1589–1602. <https://ddec1-0-enctp.trendmicro.com:443/wis/clicktime/v1/query?url=https%3a%2f%2fdoi.org%2f10.1021%2fJACS.1C10171&umid=f41f88c2-787a-49c3-bdbf-f61339fab54&auth=8d3ccd473d52f326e51c0f75cb32c9541898e5d5-2e0273b41fbaaadd518d262fb6e59c60ebaa8cb5>.)

Although the authors highlight that their methodology only enables qualitative insights (i.e., it is unable to assess anion effects or difference along the alkali cations' group), they may still observe significant differences with 2+ and 3+ species, given their high acidity.

Response 2-(a) We would like to thank both referees for making this paper more robust, self-consistent and clearer in its limitations.

In response to this final query, we conducted additional experiments using multivalent cations. Cu nanocubes were subjected to a 30-minute treatment at about -1 V_{RHE}, using 0.05 M K₂SO₄, 0.1 M MgSO₄, and 0.05 M Al₂(SO₄)₃ solutions, respectively, before undergoing cyclic voltammetry and TEM characterizations. The electrolyte's pH was adjusted to around 3 by the addition of sulfuric acid. Notably, after treatment in the 0.1 M Mg²⁺ and Al³⁺ electrolyte, signals from Cu defects were significantly suppressed (**New Figure S27a**). Treatment in Al³⁺ electrolyte resulted in a more pronounced suppression of the OH- adsorption signal for both Cu (100) and defects compared to that in K⁺ and Mg²⁺ electrolytes.

In addition, the emergence of anodic peaks at -0.02 V to +0.08 V, along with their asymmetrical nature, does not align with the OH- adsorption on Cu sites. This discrepancy can be attributed to the contamination of the Cu surface most likely by the hydroxide precipitates when Mg²⁺ or Al³⁺ present in the electrolyte, a consequence of the highly cathodic potential fostering a strong alkaline environment near the electrode, despite starting under acidic conditions. These observations align with the reports by Mariana C.O. Monteiro *et al.* in *Electrochimica Acta*, 2019, 325, 134915, and Saket S. Bhargava *et al.* in *Electrochimica Acta*, 2021, 394, 139055.

Our TEM Cu nanocubes after 30min treatment at about -1 V_{RHE} in 0.05M Al₂(SO₄)₃ exhibit well-preserved cubic shape, potentially due to the protection of the Cu surface by Al(OH)₃ precipitates or the absence of cathodic corrosion in Al³⁺ electrolyte (**new Figure S27b**). Consequently, we are unable to conclusively determine the role of multivalent cations in Cu reconstruction under cathodic potential, acknowledging the limitations of our experimental approaches.

New Fig S27 (a) Voltammetric profiles of the Cu nanocubes treated in 0.05M K_2SO_4 (black), 0.1M $MgSO_4$ (red) and 0.05M $Al_2(SO_4)_3$ (purple) at $-1.02V_{RHE}$ for 30min. With a higher valence of the cations, Cu related signal in CV was more suppressed. **(b)** TEM images of the Cu nanocubes treated in 0.05M $Al_2(SO_4)_3$ electrolyte at $-1.02V_{RHE}$ for 30min. The cubic shape was well preserved due to the protection of Cu surface by $Al(OH)_3$ precipitates or the absence of cathodic corrosion in Al^{3+} electrolyte. The regions marked by red circles were possible precipitated Al species covering the Cu nanocubes.

Added text to the conclusion part of the paper, pg 19:

“It was predicted⁷² that multivalent cations can have significant effect on CO_2RR performance and some experimental exploration was reported.⁷³ Hence we went on investing whether multivalent cations (i.e., Mg^{2+} and Al^{3+}) can also lead to cathodic corrosion of Cu. The results (Supplementary Figure S27a) show that, despite we deliberately using an acidic environment (adding H_2SO_4 to until the pH reaches 3), the local basic environment near the Cu electrode likely still result in the precipitation of hydroxides^{74,75} which contaminated the Cu electrode. As a result, the CV results can not be used to detect Cu surface reconstruction or to judge whether the cathodic corrosion occurs. No significant structural changes were observed using TEM (Supplementary Figure S27b) after the reaction with multivalent cations. Hence it remains unclear whether multivalent cations can lead to or help to prevent cathodic corrosion.”

REVIEWER COMMENTS

Reviewer #1 (Remarks to the Author):

The authors' modifications and omissions (changes in the title for example) are highly appreciated. The manuscript is certainly more coherent now. A few comments to be addressed in a future revision are as follows.

- Taking into account <https://pubs.acs.org/doi/10.1021/acs.jpcllett.1c04187> and <https://pubs.acs.org/doi/10.1021/acsami.8b13883> and coming back to the role of cation concentration, could Cu form intermediate ternary hydride with adsorbed alkali cation? In addition, it would be more accurate to mention that alkali cation induced cathodic corrosion of Cu is presumably happening along with other dissolution/ redeposition mechanisms involving different transient species (<https://www.nature.com/articles/s41929-023-01070-8>).
- It is clear that initial cathodic treatment would reduce the oxide layer. However, if the purpose is solely to remove oxide layer, then the time of cathodic treatment definitely has to be shorter (at least for -1.1 V vs RHE condition). This conclusion is drawn from the images in Figure 1 c-f and Figure 2 a-d. The images after 20 min cathodic treatment show that Cu is dissolved more than the initial thickness of the oxide (as the authors discuss in the main text) which means that cathodic corrosion has happened and thus the first CV after 20min is the one of corroded Cu oxide/metal. In other words, it still remains unclear where is the threshold (in time) between oxide reduction and cathodic corrosion.
- The statement "The difference in surface defects between samples treated for 30 minutes and those for 150 minutes may be random, given that the initial surface-oxidation of the Cu nanocubes and the following electrochemical reduction process are not within our control" is elusive. Indeed, since these two CVs are two separate experiments there will be some discrepancy, but how comparable can be the CVs after 20 min and 65 min of cathodic treatment in this case? (Figure 2e, the comparison is done by authors in page 7 of main text). The relevance of the chosen method of comparison and justification of its effect is unclear.
- Also, two current density profiles in Figure R2-6 show that the overall current density value (not considering the trend) remains essentially the same at -1.1 VRHE and at -0.37 VRHE. However, applying 3 times lower potential should have caused substantial decrease in the current density value provided the electrochemical systems were similar. Please provide an explanation for this.
- Please include the information about the EELS measurement parameters to the methods part of the main text.

Reviewer #3 (Remarks to the Author):

The manuscript describes the IL-TEM and cyclic voltammetry studies of the degradation of Cu cubes during electrochemical carbon dioxide reduction and proposes that the process is driven by cathodic corrosion. I have examined the revised manuscript and the comments by the other reviewers. In general, I believe the authors have tried their best to address the comments by the reviewers, but at this point, the paper still needs a bit more work.

Here are my comments:

1. The manuscript is now a little confusing to read and I suggest that the authors do the following,
 - a. Be more precise in the nomenclature when they are referring to the pre-catalyst (e.g. page 4, last paragraph of introduction, many approaches towards engineering the Cu catalyst morphology) versus the actual working state of the catalyst.
 - b. Have a table summarizing the different conditions they studied for the reactions (especially in KHCO_3 , K_2SO_4 and H_2SO_4) in terms of methods used and whether they observe cathodic corrosion or not
2. I share the concern of reviewer 1 whether there is an impact of pH on their results (comment 1d). I suggest the authors address this point more thoroughly in their manuscript.
3. Can the authors quantify better the results of their IL-TEM experiments? For example, perhaps the average change in the projected area of several or all the imaged cubes between

measurements. The authors had only selected examples to show in their Figures. I think it will be clearer to the reader if the authors can show in a plot the changes between different conditions with their associated spread.

4. The authors did not provide any references or supporting evidence that shows that the red shift in Figure 4a can be associated with the formation of anionic copper. I am not convinced that the shift is not an experimental artifact. The authors need to provide literature support for the claim and demonstrate the reproducibility of the shift.

5. Lastly, the authors can consider revising the general tone of the paper and adopt the use of more speculative language in their discussions. I believe doing so would not compromise the novelty of their findings.

Reviewer 1 (Remarks to the Author):

The authors' modifications and omissions (changes in the title for example) are highly appreciated. The manuscript is certainly more coherent now. A few comments to be addressed in a future revision are as follows.

Response: We thank the referee for the encouraging comment. We have carefully addressed the additional comments from referee as follows:

Reviewer 1-(aa). Taking into account <https://pubs.acs.org/doi/10.1021/acs.jpcelett.1c04187> and <https://pubs.acs.org/doi/10.1021/acsami.8b13883> and coming back to the role of cation concentration, could Cu form **intermediate ternary hydride** with adsorbed alkali cation?

Response 1-(aa) We thank the referee for this insightful question. Cathodic corrosion, in general, remains a phenomenon that lacks comprehensive understanding. While it is acknowledged to occur, the precise nature of the corrosion intermediates, not only on copper but also on other metals, remains elusive. Recent paper by Raffaella's team (Vavra, J. et al., Nature Catalysis 7, 89-97 (2024)) achieved important success in identifying such intermediates experimentally for copper. They utilized a strong coordinating agent to stabilize the intermediates, enabling observation through online mass spectrometry. However, it's worth noting that certain experimental conditions were not thoroughly examined in their work, such as the involvement of CO.

The intermediate ternary hydride was hypothesized by Koper's team (<https://pubs.acs.org/doi/10.1021/acsami>.) and was adopted by the theoretical investigation by Alexandrov (<https://pubs.acs.org/doi/10.1021/acs.jpcelett.1c04187>). In our work, we took Alexandrov's theoretical framework (and we thanked him for his advice and help in the acknowledgement) and therefore, we naturally adopted this hypothesis of intermediate ternary hydride. Our experimental results are also consistent with this hypothesis of intermediate ternary hydride, although we cannot completely rule out other intermediate forms. It's important to recognize that various corrosion mechanisms and intermediates might coexist. Our discussions primarily focus on intermediates other than the CO-related ones highlighted in recent publications in Nature Catalysis by the teams of Raffaella's and of Magnussen's, given our experimental evidence suggesting that CO is not a necessary conditions for Cu to experience significant reconstruction beyond the reduction of surface oxides.

In light of this comprehensive consideration and to avoid introducing unnecessary terminology into the field, we have revised our manuscript to include the term "intermediate ternary hydride", complete with appropriate references, as outlined above.

Pg 4 original text: *"Our experiments and modelling suggest that Cu can be etched out by forming transient complexes with alkali cations, which can then be redeposited as metallic Cu under cathodic conditions, leading to experimentally observed reconstruction of the Cu catalyst and the formation of smaller Cu particles during the CO₂RR reaction."*

New text: *"Our experiments and modelling suggest that Cu can be etched out by **possibly forming intermediate ternary hydride^{40,41}** with alkali cations, which can then be redeposited as metallic Cu under cathodic conditions, leading to experimentally observed reconstruction of the Cu catalyst and the formation of smaller Cu particles during the CO₂RR reaction."*

Pg 10 original text: “In literature,^{53,54} it was believed that alkali cations can lead to the formation of transient species with the cathode metal.”

New text: “Koper and colleagues^{40,55} proposed that the presence of alkali cations can lead to the formation of intermediate ternary hydride with the cathode metal.”

Pg 15 original text: “From these results, we can infer a plausible mechanism⁵⁹ of the cathodic corrosion of Cu: at an electrode potential more negative than the onset potential (i.e., about -0.4 V_{RHE} at our conditions) and with the presence of alkali cations in the solution, Cu forms soluble hydride complexes.”

New text: From these results, we can infer a plausible mechanism⁵⁹ of the cathodic corrosion of Cu: at an electrode potential more negative than the onset potential (i.e., about -0.4 V_{RHE} at our conditions) and with the presence of alkali cations in the solution, Cu forms soluble intermediate ternary hydride species.

Reviewer 1-(ab) In addition, it would be more accurate to mention that alkali cation induced cathodic corrosion of Cu is presumably happening along with other dissolution/ redeposition mechanisms involving different transient species (<https://www.nature.com/articles/s41929-023-01070-8>).

Response 1-(ab) We agree with the referee. Please note that this has been included in the original manuscript already. In the first round of review, review 2 has raised a relevant question and we have addressed it in Response 2.6 and added Figure S26. We have made some modifications to the last part of the discussion on pages 20-21 (also to answer **Reviewer 3-2**). The text read like “The occurrence of alkali cation-induced cathodic corrosion in Cu does not negate the possibility of other mechanisms contributing to catalyst reconstruction...” and “...Consequently, it is reasonable to infer that several corrosion mechanisms might coexist, with alkali-cation induced cathodic corrosion possibly being the dominating process, especially considering the relatively low coverage of CO on the Cu catalyst in the reaction.”

Reviewer 1-(ac). It is clear that initial cathodic treatment would reduce the oxide layer. However, if the purpose is solely to remove oxide layer, then the time of cathodic treatment definitely has to be shorter (at least for -1.1 V vs RHE condition). This conclusion is drawn from the images in Figure 1 c-f and Figure 2 a-d. The images after 20 min cathodic treatment show that Cu is dissolved more than the initial thickness of the oxide (as the authors discuss in the main text) which means that cathodic corrosion has happened and thus the first CV after 20min is the one of corroded Cu oxide/metal. In other words, it still remains unclear where is the threshold (in time) between oxide reduction and cathodic corrosion.

Response 1-(ac) We thank the referee for the insightful question. We appreciate the opportunity to further clarify this point, which was also touched upon in a previous round of reviews (**Reviewer 1f:** “Is there any dependence of restructuring initiation time at different potentials above the onset potential?”). We would like to reiterate that the primary focus of our study is to ascertain the occurrence of alkali-induced cathodic corrosion on Cu electrocatalysts, rather than to delineate the timing or rate of this phenomenon. Unfortunately, our main analytical approaches (i.e., IL-TEM and CV) do not possess the temporal resolution required to conclusively determine the timing or rate of the corrosion process.

It should be reasonable to speculate that this type of corrosion would commence as soon as the metallic copper surface is exposed after the removal of surface oxides. However, the variability in surface oxide thickness across different particles might result in differing observed initiation times. For example, a particle with a 2 nm oxide layer might exhibit a shorter initiation time than one with a 2.5 nm layer. Consequently, the term "initiation-time" may not be well-defined in the context of our experiment.

Nevertheless, our in situ XAFS results (**Figure 4**) suggest that oxide layers are typically removed within 15 minutes under standard reaction conditions, revealing a metallic Cu surface. This observation aligns with literature findings, such as those by Zhu *et al.* in the Proceedings of the National Academy of Sciences, which reported complete reduction within 10 minutes for a 40 nm Cu₂O layer. Furthermore, a very recent study in *Nature Catalysis* by Núria López and colleagues indicated that at weak reduction potentials (*i.e.*, 0 V_{RHE}), it takes approximately 208 seconds for an oxygen atom to diffuse through a 5 nm copper oxide layer, which corroborates our experimental findings.

Therefore, the benchmark cyclic voltammetry was performed on the Cu nanoparticles after a 20-minute treatment in our experiments to ensure complete removal of surface oxides. Any morphological changes observed beyond this point are not related to the reduction of surface oxides but to other processes. This approach allows us to clearly differentiate between changes due to oxide dissolution/redeposition and those arising from mechanisms such as alkali-cation-induced cathodic corrosion. Indeed, the cathodic corrosion might have already commenced within this 20-minute window, which is a critical aspect given that a metallic surface is essential for the occurrence of cathodic corrosion.

In conclusion, we hope the reviewer would agree that detecting the ill-defined "initiation-time" is not a goal of this paper. Our main objective is to demonstrate the occurrence of alkali cation-induced cathodic corrosion in Cu catalysts, a phenomenon not previously reported. The successful removal of surface oxides in the benchmark CV experiments serves as a crucial control in substantiating our findings.

We have included further discussion on this topic at the end of the manuscript on page 21.

*“The identification of the alkali cation-induced cathodic corrosion in Cu catalysts marks a crucial first step towards devising effective mitigation strategies. In this study, we focused on identifying the occurrence of alkali cation-induced cathodic corrosion. However, our current methodologies, such as IL-TEM and CV, lack the temporal resolution necessary to analyze the kinetics of this corrosion process or to pinpoint its initiation-time. It is reasonable to speculate that this type of corrosion will start as soon as the metallic copper is exposed to the electrolyte following the removal of surface oxides, though the oxide thickness and hence the timing may vary slightly from particle to particle. Our in situ XAFS results (**Figure 4**) suggested that oxide layers should be removed within 15 minutes under typical reaction conditions, consistent with literature¹¹ that reports a completion of reduction within 10 minutes for a 40 nm Cu₂O. Furthermore, Núria López and colleagues recently reported⁸⁰ that, at weak reduction potentials (*i.e.*, 0 V_{RHE}), it takes approximately 208 seconds for an oxygen atom to diffuse through a 5 nm copper oxide layer. This simulation result largely agrees with our experimental observations.”*

Reviewer 1-(ad). The statement “The difference in surface defects between samples treated for 30 minutes and those for 150 minutes may be random, given that the initial surface-oxidation of the Cu nanocubes and the following electrochemical reduction process are not within our control” is elusive. Indeed, since these two CVs are two separate experiments there will be some discrepancy, but how comparable can be the CVs after 20 min and 65 min of cathodic treatment in this case? (Figure 2e, the comparison is done by authors in page 7 of main text). The relevance of the chosen method of comparison and justification of its effect is unclear.

Response 1-(ad) We apologize for any confusion stemming from our previous explanation and appreciate the opportunity to provide further clarification and correction. The question raised by the referee, along with the question labelled “**Reviewer 1g**” from the previous round, pertains to Figure 3a in our study, where we illustrate the effect of potential. In this experiment, we conducted a comparison among the following:

- (i) CV of Cu nanocubes treated at -0.36V for 30 mins, serving as a benchmark.
- (ii) CV of Cu nanocubes treated with a total of 150 mins at -0.36V,
- (iii) CV of Cu nanocubes treated at -0.36V for 30 mins followed by approximately -1V for 3 mins

The key discovery from this analysis is that an additional 3-minute treatment at -1V in the case (iii) resulted in significantly more pronounced changes in catalyst morphologies, as observed from the CV, compared to an extra 120-minute treatment at -0.36V in the case (ii), despite the total charge transfer being approximately the same when we integrate the current. This observation strongly suggests the involvement of a different mechanism altering the catalyst morphology at more negative potentials, consistent with the alkali-cation-induced cathodic potential hypothesis.

We've also observed a slight disparity between the CVs from cases (i) and (ii), albeit much smaller than that between cases (ii) and (iii) (left panel of Figure R3-1). While this minor distinction does not impact our primary conclusion, we endeavored to investigate it further. However, despite our efforts, we were unable to identify a method to eliminate this slight difference, hence attributing it to "random" causes, which may have led to confusion.

Several potential sources could contribute to such a slight variance. **Firstly**, there may be subtle differences in the samples between the two electrodes, even when utilizing the same batch of Cu nanocubes, possibly stemming from random experimental errors such as the slight differences in the thickness of the oxide shell. **Secondly**, the longer duration in experiment case (ii) might render the particles more susceptible to coalescence or agglomeration, common degradation pathways for nanoparticle catalysts (Raffaella Buonsanti et al., *Angew Chem. Int. Ed.* 2020, 59, 14736-14746; Peidong Yang et al., *Nature*. 2023, 614). **Thirdly**, while minor, we cannot exclude other potential degradation pathways such as the H₂-induced pathway (e.g., Matsushima, H., et al., *J Am Chem Soc* 131, 10362-10363 (2009)).

Although our current methodologies do not provide a definitive explanation for these slight differences, it's important to underscore that our primary focus lies on the more substantial disparity observed in the CV of case (iii) compared to others. To validate the robustness of our findings, we repeated the experiment (right panel of **Figure R3-1**), and the conclusion remains consistent.

Fig R3-1. Initial (left) and Repeated (right) Voltammograms of Cu nanocubes after 30 mins (black curve) and 150 mins (red curve) of reactions at around $-0.36V_{RHE}$ in $0.1M KHCO_3$, in contrast of the profile after an initial 30-min reaction at around $-0.36V_{RHE}$, followed by a 3-min reaction at approximately $-1.01 V_{RHE}$.

We also acknowledge the referee's concern regarding the relatively minor differences observed in the two CVs between the Cu nanocubes treated at $-1.1V$ for 20 mins and 65 mins, as depicted in Figure 2e. In order to underscore the robustness of this result, we conducted three separate CV experiments using three electrodes of Cu nanocubes prepared at the same batch and treated under identical conditions: at $-1.1V$ for 20 minutes in $0.1M KHCO_3$ (**Fig R3-2**). Considering the limited availability of rotating disk electrodes during a single batch of sample preparation in our laboratory, , we focused on obtaining multiple attempts on the 20-minute benchmark CV.

We plotted the standard variations of the CVs of Cu nanocubes treated for 20 mins using three repeated experiments. It's evident that the results exhibit high consistency, and the CV of the 65-minute treated sample diverges significantly from the benchmark, therefore agreeing with the observations in Figure 2e. In summary, these "random" errors is a minor point which does not affect the main conclusion of the paper.

Fig R3-2. Voltammograms profiles of the Cu nanocubes recorded after 20 mins with standard errors from three independent measurements (blue shaded area) and 65 mins (red curve) of reactions at about $-1.05V_{RHE}$ in $0.1M KHCO_3$.

Reviewer 1-(ae). Also, two current density profiles in Figure R2-6 show that the overall current density value (not considering the trend) remains essentially the same at $-1.1 V_{RHE}$ and at $-0.37 V_{RHE}$. However, applying 3 times lower potential should have caused substantial decrease in the current density value provided the electrochemical systems were similar. Please provide an explanation for this.

Response 1-(ae) We thank referee for this comment. We wish to highlight that the 10-hour CO_2RR was conducted in CO_2 -saturated 0.1M $KHCO_3$ solution in a two-compartment H-cell. The current observed on Cu catalysts is predominantly constrained by the transportation of CO_2 from the electrolyte to the electrode surface, owing to the notably limited solubility of CO_2 in aqueous solutions.

In contrast, the 10-hour CORR experiment in a flow cell, utilizing a gas diffusion layer, facilitates direct diffusion of gaseous CO to the interface between the electrolyte and catalysts, thereby ensuring significantly improved mass transport conditions. Consequently, the resulting current is more closely correlated with the surface area of the catalysts. At similar overpotential, the current achieved on Cu catalysts operating within a Flow cell configuration will significantly exceed that which is limited by reactant transport in an H-cell.

It is worth noting that compared to CORR in a flow cell, the restricted mass transport condition in the H-cell setup leads to lower CO coverage on the Cu surface at equivalent overpotentials, subsequently diminishing the reaction rate towards multi-carbon products (Calle-Vallejo, F et al., *Angew. Chem. Int. Ed.* 2013, 52, 7282-7285; Lei Wang et al., *ACS Catal.* 2018, 8, 7445-7454). Consequently, a higher overpotential is necessitated for CO_2RR in an H-cell compared to CORR in a flow cell, in order to achieve comparable activity levels towards achieving deeper reduction products.

So in summary, applying 3 times lower potential does not necessarily cause substantial difference in current density because of the different cells were used.

Reviewer 1-(af). Please include the information about the EELS measurement parameters to the methods part of the main text.

Response 1-(af): The following description about the EELS measurement has now be included in the method section after introducing the STEM. We thank the referee for spotting this.

“In a typical STEM-EELS measurement, a probe current of around 100pA was utilized, with a dwell time of approximately 20 milliseconds per pixel. To generate the elemental map, background removal, signal integration, and multiple linear least square fitting were performed using Gatan Digitalmicrograph software.”

Reviewer #3 (Remarks to the Author):

The manuscript describes the IL-TEM and cyclic voltammetry studies of the degradation of Cu cubes during electrochemical carbon dioxide reduction and proposes that the process is driven by cathodic corrosion. I have examined the revised manuscript and the comments by the other reviewers. In general, I believe the authors have tried their best to address the comments by the reviewers, but at this point, the paper still needs a bit more work.

Here are my comments:

Reviewer 3-1. The manuscript is now a little confusing to read and I suggest that the authors do the following,

a. Be more precise in the nomenclature when they are referring to the pre-catalyst (e.g. page 4, last paragraph of introduction, many approaches towards engineering the Cu catalyst morphology) versus the actual working state of the catalyst.

Response 3-1 a. We appreciate the referee's encouraging comments and many constructive suggestions. We have made necessary changes to the text to differentiate “pre-catalysts”, as follows:

Pg 4:

“However, because of the alkali cation-induced cathodic corrosion, many approaches towards engineering the morphology of Cu pre-catalyst may unlikely bring long-term selectivity/activity benefits.”⁴⁴⁻⁴⁶

Pg 16:

“Therefore, if the alkali cation-induced cathodic corrosion is in play, Cu catalysts will inevitably undergo reconstructions under a typical CO₂RR condition and the selectivity advantages brought by morphology control in the pre-catalyst stage will unlikely to be stable.”

Pg 19:

*“As schematically summarized in **Figure 7**, we illustrate that this corrosion behavior in Cu does not necessarily preclude stable operation of CO₂RR, but will limit long-term selectivity and activity enhancement from morphology-controlled Cu pre-catalysts.”*

b. Have a table summarizing the different conditions they studied for the reactions (especially in KHCO₃, K₂SO₄ and H₂SO₄) in terms of methods used and whether they observe cathodic corrosion or not.

Response 3-1 b. We appreciate the referee's excellent suggestion, which will surely aid readers in keeping track of important experimental details. We have now included a new Table S1 in the supplementary information section, summarizing experimental conditions. Moreover, we have ensured thorough cross-referencing between the main text and supplementary information. Each entry in the table now denotes the corresponding figure where the experiment condition is utilized, and vice versa. We are confident that this implementation will enhance readers' comprehension and navigation through our work. For the next question about pH, we also summarized a separate Table S2 just to highlight all the pH conditions we have tested.

An example of the text referencing Table S1 on page 5: “A pair of representative IL-TEM bright field images of the catalyst before and after treating the catalysts at $-1.1V_{RHE}$ in a CO_2 -saturated $0.1M KHCO_3$ electrolyte (**Table S1, entry 2**) for 20 mins were shown in **Figure 1c.**”

Table S1 Summary of the experimental conditions investigated in this study, detailing the methods employed and the observation of alkali cation-induced cathodic corrosion. Instances where significant morphological changes of Cu were not observed are highlighted with 'No' in red in the corresponding columns representing the observation techniques. 'n.a.' indicates that data is not available.

#	Electrochemical Treatment Conditions					IL-TEM	CV†	Ex situ TEM
	Potential	Reactor	Gas	Electrolyte	Note	Significant morphology changes of Cu?		
1	$\sim -1.1 V_{RHE}$	H-Cell	CO_2	$0.1M KHCO_3$ pH=6.8	Initial long-term CO_2RR stability test (10h) with Cu nanocubes. Figures S5, S23, 6	n.a.	n.a.	Yes
2	$\sim -1.1 V_{RHE}$	Single Cell	CO_2	$0.1M KHCO_3$	IL-TEM to initially identify morphological changes beyond the reduction of surface oxides Figures 1, S6, S7	Yes	n.a.	Yes
3	$\sim -1.1 V_{RHE}$	Single Cell	Ar	$0.1M KHCO_3$ pH=8.6	Having CO_2 is not a necessary factor for the observed morphological change Figures 2, S7, S9, S10, S11, S13	Yes	Yes	Yes
4	$\sim 0 V_{RHE}$	Single Cell	Ar	$0.1M KHCO_3$	Additional experiment to show that we did not just observe the reduction of surface oxides Figures S7, S8	No	n.a.	n.a.
5	$\sim -1.1 V_{RHE}$	Single Cell	Ar	$0.1M K_2SO_4$ pH=7	The effect of different anions. Figures S11	Yes	n.a.	Yes
6	$\sim -1.1 V_{RHE}$	Single Cell	Ar	$0.1M K_2CO_3$ pH=11.6		Yes	n.a.	Yes
7	$\sim -1.1 V_{RHE}$	Single Cell	Ar	$0.05M H_2SO_4$ pH=1.16	The effect of K^+ cation Figures 2, S12	No	No	n.a.
8	$\sim -1.1 V_{RHE}$	Single Cell	Ar	$0.05M H_2SO_4$ + $0.3M K_2SO_4$ pH=1.28		Yes	Yes	n.a.
9	$\sim -1.1 V_{RHE}$	Single Cell	Air	$0.05M H_2SO_4$	To purposely reoxidize Cu and confirm that the phenomenon we observed is very different than oxidation/reduction of Cu Figure S12	Yes	Yes	n.a.
10	$\sim -1.1 V_{RHE}$	Single Cell	Ar	$0.5M KHCO_3$	Changing K^+ concentration Figure S13	Yes	Yes	n.a.
11	$\sim -1.1 V_{RHE}$	Single Cell	Ar	$1.5M KHCO_3$		Yes	Yes	n.a.
12	$\sim -1.1 V_{RHE}$	Single Cell	Ar	$0.2M Li_2SO_4$	Constant potential experiments using different alkali cations Figure S14	n.a.	The reference for the group	n.a.
13	$\sim -1.1 V_{RHE}$	Single Cell	Ar	$0.2M Na_2SO_4$		n.a.	Yes	n.a.

14	$\sim -1.1 V_{RHE}$	Single Cell	Ar	0.2M K ₂ SO ₄	Constant current experiments using different alkali cations Figure S15	n.a.	Yes	n.a.
15	$\sim -1.1 V_{RHE}$	Single Cell	Ar	0.2M Cs ₂ SO ₄		n.a.	Yes	n.a.
16	$\sim -1.27 V_{RHE}$	Single Cell	Ar	0.2M Li ₂ SO ₄		n.a.	The reference for the group	n.a.
17	$\sim -1.1 V_{RHE}$	Single Cell	Ar	0.2M Cs ₂ SO ₄		n.a.	Yes	n.a.
18	$\sim -0.92 V_{RHE}$	Single Cell	Ar	0.05M H ₂ SO ₄		n.a.	The reference for the group	n.a.
19	$\sim -1.5 V_{RHE}$	Single Cell	Ar	0.05M H ₂ SO ₄ +0.3M K ₂ SO ₄		n.a.	Yes	n.a.
20	$\sim -0.36 V_{RHE}$	Single Cell	Ar	0.1M KHCO ₃	Looking for the threshold potential Figures 3, S9, S10, S16, S19	No	No	n.a.
21	$\sim -0.4 V_{RHE}$	Single Cell	Ar	0.1M KHCO ₃		n.a.	No	n.a.
22	$\sim -0.6 V_{RHE}$	Single Cell	Ar	0.1M KHCO ₃		n.a.	Yes	n.a.
23	$\sim -0.8 V_{RHE}$	Single Cell	Ar	0.1M KHCO ₃		n.a.	Yes	n.a.
24	$\sim -1.3 V_{RHE}$	Single Cell	Ar	0.1M KHCO ₃		n.a.	Yes	n.a.
25	$\sim -0.36 V_{RHE}$ (30mins + 120mins)	Single Cell	Ar	0.1M KHCO ₃	Additional experiments to confirm the effect of the electrodepotential Figures S9, S10, 3, S17, S18, S19, S20	No	No	No
26	$\sim -0.36 V_{RHE}$ (30 mins) + $\sim -1 V_{RHE}$ (3min)	Single Cell	Ar	0.1M KHCO ₃		Yes	Yes	n.a.
27	$\sim -0.36 V_{RHE}$ (25mins + 35 mins)	Single Cell	Ar	1M KOH		n.a.	No	n.a.
28	$\sim -0.36 V_{RHE}$ (25mins) + $\sim -1 V_{RHE}$ (30 secs)	Single Cell	Ar	1M KOH pH=14		n.a.	Yes	n.a.
29	$\sim -0.4 V_{RHE}$	In situ cell	CO ₂	0.1M KHCO ₃	In situ XAFS experiment Spherical 7nm Cu nanoparticles Figures 4, S21	n.a.	n.a.	n.a.
30	$\sim -1.1 V_{RHE}$	In situ cell	CO ₂	0.1M KHCO ₃		n.a.	n.a.	n.a.
31	$\sim -1.1 V_{RHE}$	H-Cell	CO ₂	0.1M KHCO ₃	10 hour CO ₂ RR with Spherical 25nm Cu nanoparticles Figures 6, S23	n.a.	n.a.	Yes
32	$\sim -0.27 V_{RHE}$	GDE, flow cell	CO	1M KOH	Comparison of variation trend in CORR current using Cu nanocubes between different cathodic potentials Figures S24	n.a.	n.a.	n.a.
33	$\sim -0.37 V_{RHE}$	GDE, flow cell	CO	1M KOH		n.a.	n.a.	n.a.
34	$\sim -0.42 V_{RHE}$	GDE, flow cell	CO	1M KOH		n.a.	n.a.	n.a.
35	$\sim -0.46 V_{RHE}$	GDE, flow cell	CO	1M KOH		n.a.	n.a.	n.a.
36	$\sim -0.37 V_{RHE}$	GDE, flow cell	CO	1M KOH	Long-term stability test (10h) with 40nm Cu nanocubes Figures 6, S24, S26	n.a.	n.a.	No
37	$\sim -0.58 V_{RHE}$	GDE, flow cell	CO	1M KOH	Long-term stability test (80 mins) with 40nm Cu nanocubes Figures S24	n.a.	n.a.	n.a.

Point by point response R3 _ 11 Apr 2024

38	$\sim -0.7 V_{\text{RHE}}$	GDE, flow cell	CO	1M KOH	Post-reaction observation after CORR at more negative potentials Figures S25	n.a.	n.a.	Yes
39	$\sim -0.9 V_{\text{RHE}}$	GDE, flow cell	CO	1M KOH		n.a.	n.a.	Yes
40	$\sim -0.37 V_{\text{RHE}}$	GDE, flow cell	CO	1M KOH	Long-term CORR stability test (10h) with 25 nm Cu spherical nanoparticles Figures 6, S26	n.a.	n.a.	No
41	$\sim -0.33 V_{\text{RHE}}$	GDE, flow cell	CO	1M KOH	Additional long-term stability test (7h) with 40nm Cu nanocubes Figures S27	n.a.	n.a.	No
42	$\sim -0.33 V_{\text{RHE}}$	GDE, flow cell	CO	1M KOH	Additional long-term stability test (7h) with 7nm nanoparticles Figures S27	n.a.	n.a.	Yes (possibly due to agglomeration)
43	$\sim -1 V_{\text{RHE}}$	Single Cell	Ar	0.05M K_2SO_4	To check the effect of group II and III cations, at pH level is 3 Figure S29	n.a.	Possible precipitation of hydroxides exclude meaningful conclusions, possibly due to local basic conditions, despite the overall pH of the electrolyte is about 3.	
44	$\sim -1 V_{\text{RHE}}$	Single Cell	Ar	0.1M MgSO_4		n.a.		
45	$\sim -1 V_{\text{RHE}}$	Single Cell	Ar	0.05M $\text{Al}_2(\text{SO}_4)_3$		n.a.		
46	$\sim -1.1 V_{\text{RHE}}$	Single Cell	Ar	0.1M H_2SO_4 + 0.3M K_2SO_4 pH=0.98	Additional experiment to confirm the pH of the electrolyte does not play major role in whether or not the alkali-cation induced cathodic corrosion will be observed in Cu Figure S32	n.a.	Yes	Yes

† For CV experiment, unless specified otherwise, the results were compared to a baseline established by pre-treating the catalysts at the same voltages for 20-30 mins to make sure the surface oxides were removed.

Reviewer 3-2. I share the concern of reviewer 1 whether there is an impact of pH on their results (comment 1d). I suggest the authors address this point more thoroughly in their manuscript.

Response 3-2 We appreciate the question raised by the reviewer. In our investigations, the electrolyte's nominal pH did not significantly influence the alkali-induced cathodic corrosion. To provide a comprehensive overview, we have included a summary table (**Table S2**) in our response, which draws data from **Table S1**, detailing the pH conditions evaluated for Cu catalysts within our study.

To further clarify whether the lower pH values noted in our experiments was a contributing factor to the alkali-cation induced cathodic corrosion, we conducted an additional experiment. This experiment involved K⁺ ions at a lower pH of approximately 0.98 (**Table S1**, entry 46). Consistent with our expectations, cathodic corrosion was observed again (**Figure S31**). This outcome reinforces our conclusion that the crucial determinant is the presence of alkali cation ions, rather than the electrolyte's pH value, within the explored pH range of 1-14.

We have added the following text into the discussion part on page 20-21 *"The occurrence of alkali cation-induced cathodic corrosion in Cu does not negate the possibility of other mechanisms contributing to catalyst reconstruction... Meanwhile the H₂-induced^{34,35} mechanisms for changing Cu morphologies could also be in play during a typical CO₂RR or CORR experiment. Our findings, as depicted in **Figure 2f**, demonstrate that under identical conditions at -1.1V_{RHE}, the morphological changes in Cu caused solely by the presence of H₂SO₄ in the electrolyte (pH=1.16, **Table S1**, entry 7) were notably less pronounced than those observed when alkali cations were present in the electrolyte (pH=1.28, **Table S1**, entry 8). An additional experiment conducted with K₂SO₄, wherein we adjusted the pH to 0.98 using H₂SO₄ (**Table S1**, entry 46), yielded significant morphological alterations again (**Figure S31**). These findings underscore the pivotal role of alkali cations over the electrolyte's pH in influencing the morphology of Cu. Consequently, it is reasonable to infer that several corrosion mechanisms might coexist, with alkali-cation induced cathodic corrosion possibly being the dominating process, especially considering the relatively low coverage of CO on the Cu catalyst in the reaction."*

Table S2 Selective entries from Table S1 showcasing the pH conditions of electrolytes (from more basic to more acidic conditions) tested for Cu NCs around -1.1V_{RHE}, along with the indication of whether alkali cation-induced cathodic corrosion occurred or not. Entries correspond to their respective entry numbers in **Table S1**.

#	Electrochemical Treatment Conditions					IL-TEM	CV [†]	Ex situ TEM
	Potential	Reactor	Gas	Electrolyte	Note	Significant morphology changes of Cu?		
28	~ -0.36 V _{RHE} (25mins) + ~ -1 V _{RHE} (30 secs)	Single Cell	Ar	1M KOH pH=14	Additional experiments to confirm the effect of the electrodepotential Figures S9, S10, 3, S17, S18, S19, S20	n.a.	Yes	n.a.
6	~ -1.1 V _{RHE}	Single Cell	Ar	0.1M K ₂ CO ₃ pH=11.6	The effect of different anions. Figures S11	Yes	n.a.	Yes
3	~ -1.1 V _{RHE}	Single Cell	Ar	0.1M KHCO ₃ pH=8.6	Having CO ₂ is not a necessary factor for the observed morphological	Yes	Yes	Yes

					change Figures 2, S7, S9, S10, S11, S13			
5	$\sim -1.1 V_{RHE}$	Single Cell	Ar	0.1M K_2SO_4 pH=7	The effect of different anions. Figures S11	Yes	n.a.	Yes
1	$\sim -1.1 V_{RHE}$	H-Cell	CO_2	0.1M $KHCO_3$ pH=6.8	Initial long-term CO_2RR stability test (10h) with Cu nanocubes. Figures S5, S23, 6	n.a.	n.a.	Yes
8	$\sim -1.1 V_{RHE}$	Single Cell	Ar	0.05M H_2SO_4 + 0.3M K_2SO_4 pH=1.28	The effect of K^+ cation Figures 2, S12	Yes	Yes	n.a.
7	$\sim -1.1 V_{RHE}$	Single Cell	Ar	0.05M H_2SO_4 pH=1.16		No	No	n.a.
46	$\sim -1.1 V_{RHE}$	Single Cell	Ar	0.1M H_2SO_4 + 0.3M K_2SO_4 pH=0.98	Additional experiment to confirm the pH of the electrolyte does not play major role in whether or not the alkali-cation induced cathodic corrosion will be observed in Cu Figure S32	n.a.	Yes	Yes

Figure S31 (a) Voltammetric profiles of the Cu nanocubes recorded after 20 mins (green curve) and 65 mins (red curve) of reaction at about $-1.05V_{RHE}$ in Ar saturated 0.05M H_2SO_4 (Table S1, entry 7). The purple curve represents the case where the Cu nanocubes first undergo 20 mins of treatment in Ar saturated 0.05M H_2SO_4 , followed by subsequent 45 mins of reaction at about $-1.05V_{RHE}$ (purple curve) in Ar saturated 0.1M H_2SO_4 + 0.3M K_2SO_4 (Table S1, entry 46) (b) TEM images of Cu NCs after 1h treatment in 0.1M H_2SO_4 + 0.3M K_2SO_4 . The scale bar represents 50nm. After additional 45min cathodic treatment in K^+ containing electrolyte, the defects related signal increases and 100 facet related signal decreases. The scale bars represent 50 nm.

Reviewer 3-3. Can the authors quantify better the results of their IL-TEM experiments? For example, perhaps the average change in the projected area of several or all the imaged cubes between measurements. The authors had only selected examples to show in their Figures. I think it will be clearer to the reader if the authors can show in a plot the changes between different conditions with their associated spread.

Response 3-3 To enhance the quantitative understanding of the IL-TEM results, we undertook additional IL-TEM studies and statistical analysis focused on the maximum depth of the

morphological changes observed in IL-TEM images after the reaction, as illustrated in Figures 1 and 2.

We conducted additional IL-TEM examinations on Cu nanocubes following treatments under two conditions: (A) a 20-minute treatment at approximately -1.1V, and (B) a 150-minute treatment at -0.37V both in an Ar-saturated 0.1M KHCO₃ environment. Representative IL-TEM images are presented in **Figure S9**, with the distribution of the maximum depth of the morphological changes detailed in **Figure S10**. Notably, in condition (A), we observed significant morphological changes extending well beyond the surface oxide layer's estimated thickness (2-5nm), a phenomenon not evident under condition (B). These findings align with our hypothesis that alkali-cation-induced cathodic corrosion occurs exclusively under condition (A) and not under condition (B), which is operated below the threshold potential.

Please also note that not every particle in case (A) show significant morphological adjustments. Typically, these particles were located at the center of a cluster, as depicted in **Figure S9**, potentially obscured by residual surfactants and hence not actively participating in the reaction on the TEM grid during the IL-TEM analysis. However, the sharp contrast between conditions (A) and (B) is evident from the absence of significant morphological alterations in any particles under condition (B). This rigorous quantitative analysis serves to validate our qualitative assessments of Cu catalysts via IL-TEM, reinforcing the reliability and depth of our findings.

We have modified the following text on page 7

“In contrast, the morphological changes observed on nanocubes subjected to electrochemical treatment under $0V_{RHE}$ (Table S1, entry 4) or slightly more negative potential of $-0.36V_{RHE}$ (Table S1, entry 25) in Ar-saturated 0.1M KHCO₃ electrolyte, are primarily confined to less than 5-6 nm (Figures S8-S10), which matches the estimated thickness of oxide layers. When we replace the electrolyte to Ar-saturated 0.1M K₂SO₄ (Table S1, entry 5) or 0.1M K₂CO₃ (Table S1, entry 6) and applied $-1.1V_{RHE}$, significant catalyst reconstruction will again be observed (Figures S11).”

Figure S9 Representative cases showing Cu NCs before and after (a) 20min in Ar-saturated 0.1M KHCO₃ at about -1.1V (Table S1, entry 3) (b) 150min in Ar-saturated 0.1M KHCO₃ at about -0.36V (Table S1, entries 20 and 25) imaged by IL-TEM. The blue profiles represent the outlines acquired from the particles before the reaction. The bold blue lines were drawn to measure the maximum difference between the Cu cubes in the projected images before and after the reaction and represented as the maximum corrosion depth in Fig S10. To mitigate the potential influence of particle orientation on the projected shapes, Cu cubes aligned approximately to the [100] zone axis, exhibiting a cubic shape with averaged lengths in the TEM images, were predominantly selected for the measurements.

Figure S10 Statistical distribution of maximum depth of morphological changes of the samples shown in Figure S9, manually measured (as in the case of Figures 1 and 2) from IL-TEM images of Cu NCs (a) after 20min in 0.1M KHCO₃ at about -1.1V_{RHE}, Ar; (b) after 150min in 0.1M KHCO₃ at about -0.36V_{RHE}, Ar. The sample size is about 40 particles in each case.

Reviewer 3-4. The authors did not provide any references or supporting evidence that shows that the red shift in Figure 4a can be associated with the formation of anionic copper. I am not convinced that the shift is not an experimental artifact. The authors need to provide literature support for the claim and demonstrate the reproducibility of the shift.

Response 3-4 We thanks the referee for pointing this out. To the best of our knowledge, there are no previous reports on XANES data regarding anionic Cu. We acknowledge that our suggestion regarding the existence of such species is speculative and lacks solid evidence. In response to the concerns raised by the referee, we have taken the following steps to address this issue:

Firstly, we have repeated the in situ XAFS experiment to demonstrate the robustness and reproducibility of our observation, as illustrated in **Figure R3-3**. This redshift phenomenon is consistently observed only when the sample is under conditions associated with alkali-cation induced cathodic corrosion.

Furthermore, we would like to highlight relevant studies in the literature that utilize similar red shifts of the absorption edge to identify the electron-rich nature of metals within their respective alloys. Notably, Guihua Yu's work published in JACs demonstrates the electron-rich nature of Pd in a Pd-In alloy through a negatively shifted absorption edge and a lower white line, as depicted in **Figure R3-4**. They state, "For ISAA In-Pdene, the energy of the absorption edge (E_0) and the height of the white line (H_w) are lower than those of Pdene and Pd foil, proving the electron-rich nature of Pd in ISAA In-Pdene." Dingsheng Wang has reported a Pt-Zn intermetallic where Pt is in electron rich environment as revealed by the lower energy of absorption edge than Pt foil in XANES (Dingsheng Wang *et al.*, *Nat Comm.* 2019, 10, 3787). They said: "As seen from the normalized X-ray absorption near-edge structure (XANES) curves at the Pt L3-edge (Fig. 2a), PtZn/HNCNT has lower energy of the absorption edge (E_0) than Pt foil and Pt/HNCNT, suggesting the electron richness of Pt atoms in PtZn/HNCNT." These studies inspired our hypothesis that spectral redshifts may provided hints of the presence of negatively charged Cu atoms in the cathodic corrosion intermediates.

In sum, we agree with the referee that the in situ XANES data only provides hints regarding the nature of the corrosion intermediates. Interestingly, there are proposals suggesting the involvement of metal anions in cathodic corrosion on other metals, such as the intermediate ternary hydride proposed by Koper's team (<https://pubs.acs.org/doi/10.1021/acsami>.) and the theoretical investigation by Alexandrov(<https://pubs.acs.org/doi/10.1021/acs.jpcllett.1c04187>), as we discussed in **Respond 1-(aa)** above. But we agree that we do not have enough evidence to make a definite statement of the existence of anionic Cu.

We have modified the relevant text to include the references and to soften the tone significantly, avoiding mentioning Cu anions entirely, as follows:

Pg 12 original text: "Intriguingly, further cathodization at $-1.05 V_{RHE}$ led to a redshift for the shoulder feature relative to the Cu foil reference, suggesting the emergence of anionic Cu species under reaction conditions (**Figure 4b**). This trend was not observed when catalysts were cathodized only at $-0.4 V_{RHE}$ for 72 minutes (**Figure 4c**)."

New text: "Intriguingly, further cathodization at $-1.05 V_{RHE}$ led to a redshift for the shoulder feature relative to the Cu foil reference (**Figure 4b**). Similar redshifts on Pt L edge⁶⁰ or Pd K edge⁶¹ has been previously reported in alloys where the precious metals becomes electron-rich. Since this trend was not observed when catalysts were cathodized only at $-0.4 V_{RHE}$ for 72 minutes (**Figure 4c**), we suspect that this feature could be related to electron-rich form of Cu

in the corrosion intermediates. More work is certainly needed to further identify the nature of such intermediates.”

Pg 12 original text: “Our in situ XAFS results suggested that an anionic form of Cu is involved in the transient species. This is in line with earlier theoretical proposals^{40,41} about involvement of metal anions in the cathodic corrosion of other metals.”

New text: “Our in situ XAFS results provided some hints on the existence of an electron-rich form of Cu in the corrosion intermediates, which are inline with earlier proposals about the possible involvement of intermediate ternary hydride^{40,41} in the cathodic corrosion of other metals.”

Fig R3-3 Repeated In situ Cu K-edge XANES of spherical Cu nanoparticles about 7 nm in size operated at -1.05 V_{RHE} in CO₂ saturated 0.1M KHCO₃

Fig R3-4 Normalized XANES spectra at the Pd K-edge of Pd foil, PdO, ISAA In-Pdene, and Pdene. Adopted from Figure 2c in Minghao Xie *et al.*, *J. Am. Chem.Soc.* 2023, 145, 13957–13967.

Reviewer 3-5. Lastly, the authors can consider revising the general tone of the paper and adopt the use of more speculative language in their discussions. I believe doing so would not compromise the novelty of their findings.

Response 3-5 We appreciate the advice from the referee. We have gone through the manuscript, and made changes to soften the tone, especially when we meant to speculate certain things. This includes new texts added in **Response 3-2** and **Response 3-4**, which we will not repeat here. We hope the tone of the paper will be more acceptable with these changes. Once again, we want to thank the referee again for supporting this paper.

Pg 4: “Our experiments and modelling suggest that Cu can be etched out by **possibly** forming intermediate ternary hydride^{40,41}”

Pg 4: “However, because of the alkali cation-induced cathodic corrosion, many approaches towards engineering the **morphology of Cu pre-catalyst may** unlikely bring long-term selectivity/activity benefits.⁴⁶⁻⁴⁸”

Pg 7: “These results suggested that CO-related mechanisms³² **is likely not the only possible mechanism responsible** for observed morphological changes **in Cu catalysts.**”

Pg 8: “This **suggests** that in scenarios where only the HER occurs without the presence of alkali cations, the catalyst reconstruction is considerably less pronounced.”

Pg 8: “These findings indicate that the further reconstruction of Cu catalysts, occurring after the initial surface oxides are removed, is **unlikely due** to additional reduction or dissolution of oxides.”

Pg 16: “Therefore, if the alkali cation-induced cathodic corrosion is in play, Cu catalysts will inevitably undergo reconstructions under a typical CO₂RR condition and the selectivity advantages brought by morphology control **in the pre-catalyst stage will unlikely to be** stable.”

REVIEWERS' COMMENTS

Reviewer #1 (Remarks to the Author):

The authors addressed the comments and updated the manuscript accordingly. A clarification of limitations in the temporal resolution of the techniques and a distinct statement of the study focus is appreciated and will prevent misunderstanding by the reader. Please include Figures R3-1 and R3-2 to the SI and, for reproducibility reasons, please add the values of convergence and collection angle for the eels measurements.

Reviewer #3 (Remarks to the Author):

The authors have adequately addressed my concerns. One final suggestion here is that it may be beneficial for the authors to do another round of general editing. The paper is somewhat lengthy for Nature Communications.

Reviewer #1 (Remarks to the Author):

The authors addressed the comments and updated the manuscript accordingly. A clarification of limitations in the temporal resolution of the techniques and a distinct statement of the study focus is appreciated and will prevent misunderstanding by the reader. Please include Figures R3-1 and R3-2 to the SI and, for reproducibility reasons, please add the values of convergence and collection angle for the eels measurements.

Response 4 (1) We would like to thank the referee for making this paper more robust, self-consistent and clearer in its limitations. We have added Figures R3-1 and Figures R3-2 in Supplementary as **new Fig S12** and **new Fig S21**. Related texts are added as below:

New text, Pg 7: "These disparities observed are notably more substantial than those arising from the experimental uncertainties inherent between individual CV experiments (Figure S12)."

New text, Pg 10: "The repeated the CV experiments show consistent results as shown in Figure S20."

Information regarding convergence and collection angle for the STEM-EELS measurements have been added in **Microscopy Characterization, Method Section**, as below:

New text, Pg 26 "The convergence half angle is about 29 mrad and the collection half angle is about 36 mrad for the STEM-EELS measurement."

Reviewer #3 (Remarks to the Author):

The authors have adequately addressed my concerns. One final suggestion here is that it may be beneficial for the authors to do another round of general editing. The paper is somewhat lengthy for Nature Communications.

Response 4 (2) We thank the referee for the comments. We have carefully proofread the manuscript and tried our best to remove any redundant materials.